# Is Out-of-Distribution Detection Learnable?

**Zhen Fang[1], Yixuan Li[2], Jie Lu[1]\*, Jiahua Dong[3,4], Bo Han[5], Feng Liu[1,6]\***

[1]Australian Artificial Intelligence Institute, University of Technology Sydney.
[2]Department of Computer Sciences, University of Wisconsin-Madison.
[3]State Key Laboratory of Robotics, Shenyang Institute of Automation,
Chinese Academy of Sciences. [4]ETH Zurich, Switzerland.
[5]Department of Computer Science, Hong Kong Baptist University.
[6]School of Mathematics and Statistics, University of Melbourne.
{zhen.fang,jie.lu}@uts.edu.au, sharonli@cs.wisc.edu,
dongjiahua1995@gmail.com,bhanml@comp.hkbu.edu.hk,feng.liu1@unimelb.edu.au

## Abstract

Supervised learning aims to train a classifier under the assumption that training and test data are from the same distribution. To ease the above assumption, researchers have studied a more realistic setting: *out-of-distribution* (OOD) detection, where test data may come from classes that are unknown during training (*i.e.*, OOD data). Due to the unavailability and diversity of OOD data, good generalization ability is crucial for effective OOD detection algorithms. To study the generalization of OOD detection, in this paper, we investigate the *probably approximately correct* (PAC) learning theory of OOD detection, which is proposed by researchers as an *open problem*. First, we find a necessary condition for the learnability of OOD detection. Then, using this condition, we prove several impossibility theorems for the learnability of OOD detection under some scenarios. Although the impossibility theorems are frustrating, we find that some conditions of these impossibility theorems may not hold in some practical scenarios. Based on this observation, we next give several necessary and sufficient conditions to characterize the learnability of OOD detection in some practical scenarios. Lastly, we also offer theoretical supports for several representative OOD detection works based on our OOD theory.

## 1 Introduction

The success of supervised learning is established on an implicit assumption that training and test data share a same distribution, *i.e.*, *in-distribution* (ID) [1, 2, 3, 4]. However, test data distribution in many real-world scenarios may violate the assumption and, instead, contain *out-of-distribution* (OOD) data whose labels have not been seen during the training process [5, 6]. To mitigate the risk of OOD data, researchers have considered a more practical learning scenario: OOD detection which determines whether an input is ID/OOD, while classifying the ID data into respective classes. OOD detection has shown great potential to ensure the reliable deployment of machine learning models in the real world. A rich line of algorithms have been developed to empirically address the OOD detection problem [6, 7, 8, 9, 10, 11, 12, 13, 14, 15, 16, 17, 18, 19, 20]. However, very few works study theory of OOD detection, which hinders the rigorous path forward for the field. This paper aims to bridge the gap.

In this paper, we provide a theoretical framework to understand the learnability of the OOD detection problem. We investigate the probably approximately correct (PAC) learning theory of OOD detection, which is posed as an open problem to date. Unlike the classical PAC learning theory in a supervised setting, our problem setting is fundamentally challenging due to the *absence of OOD data* in training.

---

*Corresponding author

36th Conference on Neural Information Processing Systems (NeurIPS 2022).

In many real-world scenarios, OOD data can be diverse and priori-unknown. Given this, we study whether there exists an algorithm that can be used to detect various OOD data instead of merely some specified OOD data. Such is the significance of studying the learning theory for OOD detection [4]. This motivates our question: *is OOD detection PAC learnable? i.e., is there the PAC learning theory to guarantee the generalization ability of OOD detection?*

To investigate the learning theory, we mainly focus on two basic spaces: domain space and hypothesis space. The domain space is a space consisting of some distributions, and the hypothesis space is a space consisting of some classifiers. Existing agnostic PAC theories in supervised learning [21, 22] are distribution-free, *i.e.*, the domain space consists of all domains. Yet, in Theorem 4, we shows that the learning theory of OOD detection is not distribution-free. In fact, we discover that OOD detection is learnable only if the domain space and the hypothesis space satisfy some special conditions, *e.g.*, Conditions 1 and 3. Notably, there are many conditions and theorems in existing learning theories and many OOD detection algorithms in the literature. Thus, it is very difficult to analyze the relation between these theories and algorithms, and explore useful conditions to ensure the learnability of OOD detection, especially when we have to explore them *from the scratch*. Thus, the main aim of our paper is to study these essential conditions. From these essential conditions, we can know *when* OOD detection can be successful in practical scenarios. We restate our question and goal in following:

> *Given hypothesis spaces and several representative domain spaces, what are the conditions to ensure the learnability of OOD detection? If possible, we hope that these conditions are necessary and sufficient in some scenarios.*

**Main Results.** We investigate the learnability of OOD detection starting from the largest space—the total space, and give a necessary condition (Condition 1) for the learnability. However, we find that the overlap between ID and OOD data may result in that the necessary condition does not hold. Therefore, we give an impossibility theorem to demonstrate that OOD detection fails in the total space (Theorem 4). Next, we study OOD detection in the separate space, where there are no overlaps between the ID and OOD data. Unfortunately, there still exists impossibility theorem (Theorem 5), which demonstrates that OOD detection is not learnable in the separate space under some conditions.

Although the impossibility theorems obtained in the separate space are frustrating, we find that some conditions of these impossibility theorems may not hold in some practical scenarios. Based on this observation, we give several necessary and sufficient conditions to characterize the learnability of OOD detection in the separate space (Theorems 6 and 10). Especially, when our model is based on *fully-connected neural network* (FCNN), OOD detection is learnable in the separate space if and only if the feature space is finite. Furthermore, we investigate the learnability of OOD detection in other more practical domain spaces, *e.g.*, the finite-ID-distribution space (Theorem 8) and the density-based space (Theorem 9). By studying the finite-ID-distribution space, we discover a compatibility condition (Condition 3) that is a necessary and sufficient condition for this space. Next, we further investigate the compatibility condition in the density-based space, and find that such condition is also the necessary and sufficient condition in some practical scenarios (Theorem 11).

**Implications and Impacts of Theory.** Our study is not of purely theoretical interest; it has also practical impacts. First, when we design OOD detection algorithms, we normally only have finite ID datasets, corresponding to the finite-ID-distribution space. In this case, Theorem 8 gives the necessary and sufficient condition to the success of OOD detection. Second, our theory provides theoretical support (Theorems 10 and 11) for several representative OOD detection works [7, 8, 23]. Third, our theory shows that OOD detection is learnable in image-based scenarios when ID images have clearly different semantic labels and styles (*far-OOD*) from OOD images. Fourth, we should not expect a universally working algorithm. It is necessary to design different algorithms in different scenarios.

## 2 Learning Setups

We start by introducing the necessary concepts and notations for our theoretical framework. Given a feature space $\mathcal{X} \subset \mathbb{R}^d$ and a label space $\mathcal{Y} := \{1, ..., K\}$, we have an ID joint distribution $D_{X_I Y_I}$ over $\mathcal{X} \times \mathcal{Y}$, where $X_I \in \mathcal{X}$ and $Y_I \in \mathcal{Y}$ are random variables. We also have an OOD joint distribution $D_{X_O Y_O}$, where $X_O$ is a random variable from $\mathcal{X}$, but $Y_O$ is a random variable whose outputs do not belong to $\mathcal{Y}$. During testing, we will meet a mixture of ID and OOD joint distributions: $D_{XY} := (1 - \pi^{\text{out}}) D_{X_I Y_I} + \pi^{\text{out}} D_{X_O Y_O}$, and can only observe the marginal distribution $D_X := (1 - \pi^{\text{out}}) D_{X_I} + \pi^{\text{out}} D_{X_O}$, where the constant $\pi^{\text{out}} \in [0, 1)$ is an unknown class-prior probability.

**Problem 1** (OOD Detection [4])**.** *Given an ID joint distribution $D_{X_IY_I}$ and a training data $S :=$ $\{(\mathbf{x}^1, y^1), ..., (\mathbf{x}^n, y^n)\}$ drawn independent and identically distributed from $D_{X_IY_I}$, the aim of OOD detection is to train a classifier $f$ by using the training data $S$ such that, for any test data $\mathbf{x}$ drawn from the mixed marginal distribution $D_X$: 1) if $\mathbf{x}$ is an observation from $D_{X_I}$, $f$ can classify $\mathbf{x}$ into correct ID classes; and 2) if $\mathbf{x}$ is an observation from $D_{X_O}$, $f$ can detect $\mathbf{x}$ as OOD data.*

According to the survey [4], when $K > 1$, OOD detection is also known as the open-set recognition or open-set learning [24, 25]; and when $K = 1$, OOD detection reduces to one-class novelty detection and semantic anomaly detection [26, 27, 28].

**OOD Label and Domain Space.** Based on Problem 1, we know it is not necessary to classify OOD data into the correct OOD classes. Without loss of generality, let all OOD data be allocated to one big OOD class, *i.e.*, $Y_O = K + 1$ [24, 29]. To investigate the PAC learnability of OOD detection, we define a domain space $\mathscr{D}_{XY}$, which is a set consisting of some joint distributions $D_{XY}$ mixed by some ID joint distributions and some OOD joint distributions. In this paper, the joint distribution $D_{XY}$ mixed by ID joint distribution $D_{X_IY_I}$ and OOD joint distribution $D_{X_OY_O}$ is called ***domain***.

**Hypothesis Spaces and Scoring Function Spaces.** A hypothesis space $\mathcal{H}$ is a subset of function space, *i.e.*, $\mathcal{H} \subset \{h : \mathcal{X} \to \mathcal{Y} \cup \{K + 1\}\}$. We set $\mathcal{H}^{\text{in}} \subset \{h : \mathcal{X} \to \mathcal{Y}\}$ to the ID hypothesis space. We also define $\mathcal{H}^{\text{b}} \subset \{h : \mathcal{X} \to \{1, 2\}\}$ as the hypothesis space for binary classification, where 1 represents the ID data, and 2 represents the OOD data. The function $h$ is called the hypothesis function. A scoring function space is a subset of function space, *i.e.*, $\mathcal{F}_l \subset \{\mathbf{f} : \mathcal{X} \to \mathbb{R}^l\}$, where $l$ is the output's dimension of the vector-valued function $\mathbf{f}$. The function $\mathbf{f}$ is called the scoring function.

**Loss and Risks.** Let $\mathcal{Y}_{\text{all}} = \mathcal{Y} \cup \{K + 1\}$. Given a loss function $\ell^2 : \mathcal{Y}_{\text{all}} \times \mathcal{Y}_{\text{all}} \to \mathbb{R}_{\geq 0}$ satisfying that $\ell(y_1, y_2) = 0$ if and only if $y_1 = y_2$, and any $h \in \mathcal{H}$, then the *risk* with respect to $D_{XY}$ is

$$R_D(h) := \mathbb{E}_{(\mathbf{x},y)\sim D_{XY}}\ell(h(\mathbf{x}), y). \tag{1}$$

The $\alpha$-risk $R_D^{\alpha}(h) := (1 - \alpha)R_D^{\text{in}}(h) + \alpha R_D^{\text{out}}(h), \forall \alpha \in [0, 1]$, where the risks $R_D^{\text{in}}(h)$, $R_D^{\text{out}}(h)$ are

$$R_D^{\text{in}}(h) := \mathbb{E}_{(\mathbf{x},y)\sim D_{X_IY_I}}\ell(h(\mathbf{x}), y), \qquad R_D^{\text{out}}(h) := \mathbb{E}_{\mathbf{x}\sim D_{X_O}}\ell(h(\mathbf{x}), K + 1).$$

**Learnability.** We aim to select a hypothesis function $h \in \mathcal{H}$ with approximately minimal risk, based on finite data. Generally, we expect the approximation to get better, with the increase in sample size. Algorithms achieving this are said to be consistent. Formally, we introduce the following definition:

**Definition 1** (Learnability of OOD Detection)**.** *Given a domain space $\mathscr{D}_{XY}$ and a hypothesis space $\mathcal{H} \subset \{h : \mathcal{X} \to \mathcal{Y}_{\text{all}}\}$, we say OOD detection is **learnable** in $\mathscr{D}_{XY}$ for $\mathcal{H}$, if there exists an algorithm $\mathbf{A}^3 : \cup_{n=1}^{+\infty}(\mathcal{X} \times \mathcal{Y})^n \to \mathcal{H}$ and a monotonically decreasing sequence $\epsilon_{\text{cons}}(n)$, such that $\epsilon_{\text{cons}}(n) \to 0$, as $n \to +\infty$, and for any domain $D_{XY} \in \mathscr{D}_{XY}$,*

$$\mathbb{E}_{S\sim D_{X_IY_I}^n}\big[R_D(\mathbf{A}(S)) - \inf_{h\in\mathcal{H}} R_D(h)\big] \leq \epsilon_{\text{cons}}(n), \tag{2}$$

*An algorithm $\mathbf{A}$ for which this holds is said to be consistent with respect to $\mathscr{D}_{XY}$.*

Definition 1 is a natural extension of agnostic PAC learnability of supervised learning [30]. If for any $D_{XY} \in \mathscr{D}_{XY}$, $\pi^{\text{out}} = 0$, then Definition 2 is the agnostic PAC learnability of supervised learning. Although the expression of Definition 1 is different from the normal definition of agnostic PAC learning in [21], one can easily prove that they are equivalent when $\ell$ is bounded, see Appendix D.3.

Since OOD data are unavailable, it is impossible to obtain information about the class-prior probability $\pi^{\text{out}}$. Furthermore, in the real world, it is possible that $\pi^{\text{out}}$ can be any value in $[0, 1)$. Therefore, the imbalance issue between ID and OOD distributions, and the priori-unknown issue (*i.e.*, $\pi^{\text{out}}$ is unknown) are the core challenges. To ease these challenges, researchers use AUROC, AUPR and FPR95 to estimate the performance of OOD detection [18, 31, 32, 33, 34, 35]. It seems that there is a gap between Definition 1 and existing works. To eliminate this gap, we revise Eq. (2) as follows:

$$\mathbb{E}_{S\sim D_{X_IY_I}^n}\big[R_D^{\alpha}(\mathbf{A}(S)) - \inf_{h\in\mathcal{H}} R_D^{\alpha}(h)\big] \leq \epsilon_{\text{cons}}(n), \ \forall \alpha \in [0, 1]. \tag{3}$$

If an algorithm $\mathbf{A}$ satisfies Eq. (3), then the imbalance issue and the prior-unknown issue disappear. That is, $\mathbf{A}$ can simultaneously classify the ID data and detect the OOD data well. Based on the above discussion, we define the strong learnability of OOD detection as follows:

---

[2]Note that $\mathcal{Y}_{\text{all}} \times \mathcal{Y}_{\text{all}}$ is a finite set, therefore, $\ell$ is bounded.

[3]Similar to [30], in this paper, we regard an algorithm as a mapping from $\cup_{n=1}^{+\infty}(\mathcal{X} \times \mathcal{Y})^n$ to $\mathcal{H}$.

**Definition 2** (Strong Learnability of OOD Detection). *Given a domain space $\mathscr{D}_{XY}$ and a hypothesis space $\mathcal{H} \subset \{h : \mathcal{X} \to \mathcal{Y}_{\text{all}}\}$, we say OOD detection is **strongly learnable** in $\mathscr{D}_{XY}$ for $\mathcal{H}$, if there exists an algorithm $\mathbf{A} : \cup_{n=1}^{+\infty}(\mathcal{X} \times \mathcal{Y})^n \to \mathcal{H}$ and a monotonically decreasing sequence $\epsilon_{\text{cons}}(n)$, such that $\epsilon_{\text{cons}}(n) \to 0$, as $n \to +\infty$, and for any domain $D_{XY} \in \mathscr{D}_{XY}$,*

$$\mathbb{E}_{S \sim D_{X_{\text{I}}Y_{\text{I}}}^n}\big[R_D^\alpha(\mathbf{A}(S)) - \inf_{h \in \mathcal{H}} R_D^\alpha(h)\big] \leq \epsilon_{\text{cons}}(n), \; \forall \alpha \in [0,1].$$

In Theorem 1, we have shown that the strong learnability of OOD detection is equivalent to the learnability of OOD detection, if the domain space $\mathscr{D}_{XY}$ is a *prior-unknown space* (see Definition 3). In this paper, we mainly discuss the learnability in the prior-unknown space. Therefore, *when we mention that OOD detection is learnable, we also mean that OOD detection is strongly learnable.*

**Goal of Theory.** Note that the agnostic PAC learnability of supervised learning is distribution-free, *i.e.*, the domain space $\mathscr{D}_{XY}$ consists of all domains. However, due to the absence of OOD data during the training process [8, 14, 24], it is obvious that the learnability of OOD detection is not distribution-free (*i.e.*, Theorem 4). In fact, we discover that the learnability of OOD detection is deeply correlated with the relationship between the domain space $\mathscr{D}_{XY}$ and the hypothesis space $\mathcal{H}$. That is, OOD detection is learnable only when the domain space $\mathscr{D}_{XY}$ and the hypothesis space $\mathcal{H}$ satisfy some special conditions, *e.g.,* Condition 1 and Condition 3. We present our goal as follows:

> ***Goal:*** *given a hypothesis space $\mathcal{H}$ and several representative domain spaces $\mathscr{D}_{XY}$, what are the **conditions** to ensure the learnability of OOD detection? Furthermore, if possible, we hope that these conditions are **necessary and sufficient** in some scenarios.*

Therefore, compared to the agnostic PAC learnability of supervised learning, our theory doesn't focus on the distribution-free case, but focuses on discovering essential conditions to guarantee the learnability of OOD detection in several representative and practical domain spaces $\mathscr{D}_{XY}$. By these essential conditions, we can know *when* OOD detection can be successful in real applications.

# 3 Learning in Priori-unknown Spaces

We first investigate a special space, called prior-unknown space. In such space, Definition 1 and Definition 2 are equivalent. Furthermore, we also prove that if OOD detection is strongly learnable in a space $\mathscr{D}_{XY}$, then one can discover a larger domain space, which is prior-unknown, to ensure the learnability of OOD detection. These results imply that it is enough to consider our theory in the prior-unknown spaces. The prior-unknown space is introduced as follows:

**Definition 3.** *Given a domain space $\mathscr{D}_{XY}$, we say $\mathscr{D}_{XY}$ is a priori-unknown space, if for any domain $D_{XY} \in \mathscr{D}_{XY}$ and any $\alpha \in [0,1)$, we have $D_{XY}^\alpha := (1-\alpha)D_{X_{\text{I}}Y_{\text{I}}} + \alpha D_{X_{\text{O}}Y_{\text{O}}} \in \mathscr{D}_{XY}$.*

**Theorem 1.** *Given domain spaces $\mathscr{D}_{XY}$ and $\mathscr{D}_{XY}' = \{D_{XY}^\alpha : \forall D_{XY} \in \mathscr{D}_{XY}, \forall \alpha \in [0,1)\}$, then*
*1) $\mathscr{D}_{XY}'$ is a priori-unknown space and $\mathscr{D}_{XY} \subset \mathscr{D}_{XY}'$;*
*2) if $\mathscr{D}_{XY}$ is a priori-unknown space, then Definition 1 and Definition 2 are **equivalent**;*
*3) OOD detection is strongly learnable in $\mathscr{D}_{XY}$ **if and only if** OOD detection is learnable in $\mathscr{D}_{XY}'$.*

The second result of Theorem 1 bridges the learnability and strong learnability, which implies that if an algorithm $\mathbf{A}$ is consistent with respect to a prior-unknown space, then this algorithm $\mathbf{A}$ can address the imbalance issue between ID and OOD distributions, and the priori-unknown issue well. Based on Theorem 1, we focus on our theory in the prior-unknown spaces. Furthermore, to demystify the learnability of OOD detection, we introduce five representative priori-unknown spaces:

• Single-distribution space $\mathscr{D}_{XY}^{D_{XY}}$. For a domain $D_{XY}$, $\mathscr{D}_{XY}^{D_{XY}} := \{D_{XY}^\alpha : \forall \alpha \in [0,1)\}$.

• Total space $\mathscr{D}_{XY}^{\text{all}}$, which consists of all domains.

• Separate space $\mathscr{D}_{XY}^s$, which consists of all domains that satisfy the separate condition, that is for any $D_{XY} \in \mathscr{D}_{XY}^s$, $\text{supp}D_{X_{\text{O}}} \cap \text{supp}D_{X_{\text{I}}} = \emptyset$, where $\text{supp}$ means the support set.

• Finite-ID-distribution space $\mathscr{D}_{XY}^F$, which is a prior-unknown space satisfying that the number of distinct ID joint distributions $D_{X_{\text{I}}Y_{\text{I}}}$ in $\mathscr{D}_{XY}^F$ is finite, *i.e.*, $|\{D_{X_{\text{I}}Y_{\text{I}}} : \forall D_{XY} \in \mathscr{D}_{XY}^F\}| < +\infty$.

• Density-based space $\mathscr{D}_{XY}^{\mu,b}$, which is a prior-unknown space consisting of some domains satisfying that: for any $D_{XY}$, there exists a density function $f$ with $1/b \leq f \leq b$ in $\text{supp}\mu$ and $0.5 * D_{X_{\text{I}}} +$

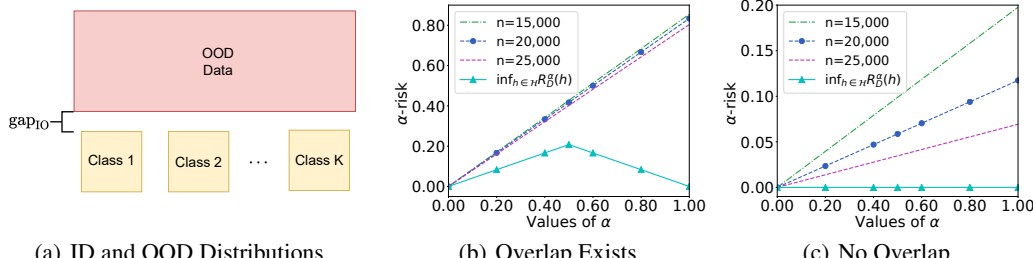

|     (a) ID and OOD Distributions | (b) Overlap Exists | (c) No Overlap |

Figure 1: Illustration of $\inf_{h\in\mathcal{H}} R_D^\alpha(h)$ (solid lines with triangle marks) and the estimated $\mathbb{E}_{S\sim D_{\text{in}}^n} R_D^\alpha(\mathbf{A}(S))$ (dash lines) with $\alpha \in [0,1)$ in different scenarios, where $D_{\text{in}} = D_{X_I Y_I}$ and the algorithm $\mathbf{A}$ is the free-energy OOD detection method [23]. Subfigure (a) shows the ID and OOD distributions. In (a), $\text{gap}_{\text{IO}}$ represents the distance between the support sets of ID and OOD distributions. In (b), since there is an overlap between ID and OOD data, the solid line is a ployline. In (c), since there is no overlap between ID and OOD data, we can check that $\inf_{h\in\mathcal{H}} R_D^\alpha(h)$ forms a straight line (the solid line). However, since dash lines are always straight lines, two observations can be obtained from (b) and (c): 1) dash lines cannot approximate the solid ployline in (b), which implies the unlearnability of OOD detection; and 2) the solid line in (c) is a straight line and may be approximated by the dash lines in (c). The above observations motivate us to propose Condition 1.

$0.5 * D_{X_O} = \int f \mathrm{d}\mu$, where $\mu$ is a measure defined over $\mathcal{X}$. Note that if $\mu$ is discrete, then $D_X$ is a discrete distribution; and if $\mu$ is the Lebesgue measure, then $D_X$ is a continuous distribution.

The above representative spaces widely exist in real applications. For example, 1) if the images from different semantic labels with different styles are clearly different, then those images can form a distribution belonging to a separate space $\mathscr{D}_{XY}^s$; and 2) when designing an algorithm, we only have finite ID datasets, *e.g.*, CIFAR-10, MNIST, SVHN, and ImageNet, to build a model. Then, finite-ID-distribution space $\mathscr{D}_{XY}^F$ can handle this real scenario. Note that the single-distribution space is a special case of the finite-ID-distribution space. In this paper, we mainly discuss these five spaces.

## 4 Impossibility Theorems for OOD Detection

In this section, we first give a necessary condition for the learnability of OOD detection. Then, we show this necessary condition does not hold in the total space $\mathscr{D}_{XY}^{\text{all}}$ and the separate space $\mathscr{D}_{XY}^s$.

**Necessary Condition.** We find a necessary condition for the learnability of OOD detection, *i.e.*, Condition 1, motivated by the experiments in Figure 1. Details of Figure 1 can be found in Appendix C.2.

**Condition 1** (Linear Condition). *For any $D_{XY} \in \mathscr{D}_{XY}$ and any $\alpha \in [0,1)$,*

$$\inf_{h\in\mathcal{H}} R_D^\alpha(h) = (1-\alpha) \inf_{h\in\mathcal{H}} R_D^{\text{in}}(h) + \alpha \inf_{h\in\mathcal{H}} R_D^{\text{out}}(h).$$

To reveal the importance of Condition 1, Theorem 2 shows that Condition 1 is a *necessary and sufficient* condition for the learnability of OOD detection if the $\mathscr{D}_{XY}$ is the single-distribution space.

> **Theorem 2.** *Given a hypothesis space $\mathcal{H}$ and a domain $D_{XY}$, OOD detection is learnable in the single-distribution space $\mathscr{D}_{XY}^{D_{XY}}$ for $\mathcal{H}$ **if and only if** linear condition (i.e., Condition 1) holds.*

Theorem 2 implies that Condition 1 is important for the learnability of OOD detection. Due to the simplicity of single-distribution space, Theorem 2 implies that Condition 1 is the necessary condition for the learnability of OOD detection in the prior-unknown space, see Lemma 1 in Appendix F.

**Impossibility Theorems.** Here, we first study whether Condition 1 holds in the total space $\mathscr{D}_{XY}^{\text{all}}$. If Condition 1 does not hold, then OOD detection is not learnable. Theorem 3 shows that Condition 1 is not always satisfied, especially, when there is an overlap between the ID and OOD distributions:

**Definition 4** (Overlap Between ID and OOD). *We say a domain $D_{XY}$ has overlap between ID and OOD distributions, if there is a $\sigma$-finite measure $\tilde{\mu}$ such that $D_X$ is absolutely continuous with respect to $\tilde{\mu}$, and $\tilde{\mu}(A_{\text{overlap}}) > 0$, where $A_{\text{overlap}} = \{\mathbf{x} \in \mathcal{X} : f_I(\mathbf{x}) > 0 \text{ and } f_O(\mathbf{x}) > 0\}$. Here $f_I$ and $f_O$ are the representers of $D_{X_I}$ and $D_{X_O}$ in Radon–Nikodym Theorem [36],*

$$D_{X_I} = \int f_I \mathrm{d}\tilde{\mu}, \quad D_{X_O} = \int f_O \mathrm{d}\tilde{\mu}.$$

**Theorem 3.** *Given a hypothesis space $\mathcal{H}$ and a prior-unknown space $\mathscr{D}_{XY}$, if there is $D_{XY} \in \mathscr{D}_{XY}$, which has overlap between ID and OOD, and $\inf_{h \in \mathcal{H}} R_D^{\text{in}}(h) = 0$ and $\inf_{h \in \mathcal{H}} R_D^{\text{out}}(h) = 0$, then Condition 1 does not hold. Therefore, OOD detection is not learnable in $\mathscr{D}_{XY}$ for $\mathcal{H}$.*

Theorem 3 clearly shows that under proper conditions, Condition 1 does not hold, if there exists a domain whose ID and OOD distributions have overlap. By Theorem 3, we can obtain that the OOD detection is not learnable in the total space $\mathscr{D}_{XY}^{\text{all}}$ for any non-trivial hypothesis space $\mathcal{H}$.

---

**Theorem 4** (Impossibility Theorem for Total Space)**.** *OOD detection is not learnable in the total space $\mathscr{D}_{XY}^{\text{all}}$ for $\mathcal{H}$, if $|\phi \circ \mathcal{H}| > 1$, where $\phi$ maps ID labels to 1 and maps OOD labels to 2.*

---

Since the overlaps between ID and OOD distributions may cause that Condition 1 does not hold, we then consider studying the learnability of OOD detection in the separate space $\mathscr{D}_{XY}^s$, where there are no overlaps between the ID and OOD distributions. However, Theorem 5 shows that even if we consider the separate space, the OOD detection is still not learnable in some scenarios. Before introducing the impossibility theorem for separate space, *i.e.*, Theorem 5, we need a mild assumption:

**Assumption 1** (Separate Space for OOD)**.** *A hypothesis space $\mathcal{H}$ is separate for OOD data, if for each data point $\mathbf{x} \in \mathcal{X}$, there exists at least one hypothesis function $h_{\mathbf{x}} \in \mathcal{H}$ such that $h_{\mathbf{x}}(\mathbf{x}) = K + 1$.*

Assumption 1 means that every data point $\mathbf{x}$ has the possibility to be detected as OOD data. Assumption 1 is mild and can be satisfied by many hypothesis spaces, *e.g.*, the FCNN-based hypothesis space (Proposition 1 in Appendix K), score-based hypothesis space (Proposition 2 in Appendix K) and universal kernel space. Next, we use *Vapnik–Chervonenkis* (VC) dimension [22] to measure the size of hypothesis space, and study the learnability of OOD detection in $\mathscr{D}_{XY}^s$ based on the VC dimension.

---

**Theorem 5** (Impossibility Theorem for Separate Space)**.** *If Assumption 1 holds, $\text{VCdim}(\phi \circ \mathcal{H}) < +\infty$ and $\sup_{h \in \mathcal{H}} |\{\mathbf{x} \in \mathcal{X} : h(\mathbf{x}) \in \mathcal{Y}\}| = +\infty$, then OOD detection is not learnable in separate space $\mathscr{D}_{XY}^s$ for $\mathcal{H}$, where $\phi$ maps ID labels to 1 and maps OOD labels to 2.*

---

The finite VC dimension normally implies the learnability of supervised learning. However, in our results, the finite VC dimension cannot guarantee the learnability of OOD detection in the separate space, which reveals the difficulty of the OOD detection. Although the above impossibility theorems are frustrating, there is still room to discuss the conditions in Theorem 5, and to find out the proper conditions for ensuring the learnability of OOD detection in the separate space (see Sections 5 and 6).

## 5    When OOD Detection Can Be Successful

Here, we discuss when the OOD detection can be learnable in the separate space $\mathscr{D}_{XY}^s$, finite-ID-distribution space $\mathscr{D}_{XY}^F$ and density-based space $\mathscr{D}_{XY}^{\mu,b}$. We first study the separate space $\mathscr{D}_{XY}^s$.

**OOD Detection in the Separate Space.** Theorem 5 has indicated that $\text{VCdim}(\phi \circ \mathcal{H}) = +\infty$ or $\sup_{h \in \mathcal{H}} |\{\mathbf{x} \in \mathcal{X} : h(\mathbf{x}) \in \mathcal{Y}\}| < +\infty$ is necessary to ensure the learnability of OOD detection in $\mathscr{D}_{XY}^s$ if Assumption 1 holds. However, generally, hypothesis spaces generated by feed-forward neural networks with proper activation functions have finite VC dimension [37, 38]. Therefore, we study the learnability of OOD detection in the case that $|\mathcal{X}| < +\infty$, which implies that $\sup_{h \in \mathcal{H}} |\{\mathbf{x} \in \mathcal{X} : h(\mathbf{x}) \in \mathcal{Y}\}| < +\infty$. Additionally, Theorem 10 also implies that $|\mathcal{X}| < +\infty$ is the necessary and sufficient condition for the learnability of OOD detection in separate space, when the hypothesis space is generated by FCNN. Hence, $|\mathcal{X}| < +\infty$ may be necessary in the space $\mathscr{D}_{XY}^s$.

For simplicity, we first discuss the case that $K = 1$, *i.e.*, the one-class novelty detection. We show the necessary and sufficient condition for the learnability of OOD detection in $\mathscr{D}_{XY}^s$, when $|\mathcal{X}| < +\infty$.

---

**Theorem 6.** *Let $K = 1$ and $|\mathcal{X}| < +\infty$. Suppose that Assumption 1 holds and the constant function $h^{\text{in}} := 1 \in \mathcal{H}$. Then OOD detection is learnable in $\mathscr{D}_{XY}^s$ for $\mathcal{H}$ **if and only if** $\mathcal{H}_{\text{all}} - \{h^{\text{out}}\} \subset \mathcal{H}$, where $\mathcal{H}_{\text{all}}$ is the hypothesis space consisting of all hypothesis functions, and $h^{\text{out}}$ is a constant function that $h^{\text{out}} := 2$, here 1 represents ID data and 2 represents OOD data.*

---

The condition $h^{\text{in}} \in \mathcal{H}$ presented in Theorem 6 is mild. Many practical hypothesis spaces satisfy this condition, *e.g.*, the FCNN-based hypothesis space (Proposition 1 in Appendix K), score-based hypothesis space (Proposition 2 in Appendix K) and universal kernel-based hypothesis space. Theorem 6 implies that if $K = 1$ and OOD detection is learnable in $\mathscr{D}_{XY}^s$ for $\mathcal{H}$, then the hypothesis space $\mathcal{H}$

should contain almost all hypothesis functions, implying that if the OOD detection can be learnable in the distribution-agnostic case, then a large-capacity model is necessary.

Next, we extend Theorem 6 to a general case, *i.e.*, $K > 1$. When $K > 1$, we will first use a binary classifier $h^b$ to classify the ID and OOD data. Then, for the ID data identified by $h^b$, an ID hypothesis function $h^{\text{in}}$ will be used to classify them into corresponding ID classes. We state this strategy as follows: given a hypothesis space $\mathcal{H}^{\text{in}}$ for ID distribution and a binary classification hypothesis space $\mathcal{H}^b$ introduced in Section 2, we use $\mathcal{H}^{\text{in}}$ and $\mathcal{H}^b$ to construct an OOD detection's hypothesis space $\mathcal{H}$, which consists of all hypothesis functions $h$ satisfying the following condition: there exist $h^{\text{in}} \in \mathcal{H}^{\text{in}}$ and $h^b \in \mathcal{H}^b$ such that for any $\mathbf{x} \in \mathcal{X}$,

$$h(\mathbf{x}) = i, \quad \text{if } h^{\text{in}}(\mathbf{x}) = i \text{ and } h^b(\mathbf{x}) = 1; \text{ otherwise, } h(\mathbf{x}) = K + 1. \tag{4}$$

We use $\mathcal{H}^{\text{in}} \bullet \mathcal{H}^b$ to represent a hypothesis space consisting of all $h$ defined in Eq. (4). In addition, we also need an additional condition for the loss function $\ell$. This condition is shown as follows:

**Condition 2.** $\ell(y_2, y_1) \leq \ell(K + 1, y_1)$, *for any in-distribution labels $y_1$ and $y_2 \in \mathcal{Y}$.*

**Theorem 7.** *Let $|\mathcal{X}| < +\infty$ and $\mathcal{H} = \mathcal{H}^{\text{in}} \bullet \mathcal{H}^b$. If $\mathcal{H}_{\text{all}} - \{h^{\text{out}}\} \subset \mathcal{H}^b$ and Condition 2 holds, then OOD detection is learnable in $\mathscr{D}_{XY}^s$ for $\mathcal{H}$, where $\mathcal{H}_{\text{all}}$ and $h^{\text{out}}$ are defined in Theorem 6.*

**OOD Detection in the Finite-ID-Distribution Space.** Since researchers can only collect finite ID datasets as the training data in the process of algorithm design, it is worthy to study the learnability of OOD detection in the finite-ID-distribution space $\mathscr{D}_{XY}^F$. We first show two necessary concepts below.

**Definition 5** (ID Consistency). *Given a domain space $\mathscr{D}_{XY}$, we say any two domains $D_{XY} \in \mathscr{D}_{XY}$ and $D'_{XY} \in \mathscr{D}_{XY}$ are ID consistency, if $D_{X_I Y_I} = D'_{X_I Y_I}$. We use the notation $\sim$ to represent the ID consistency, i.e., $D_{XY} \sim D'_{XY}$ if and only if $D_{XY}$ and $D'_{XY}$ are ID consistency.*

It is easy to check that the ID consistency $\sim$ is an equivalence relation. Therefore, we define the set $[D_{XY}] := \{D'_{XY} \in \mathscr{D}_{XY} : D_{XY} \sim D'_{XY}\}$ as the equivalence class with respect to space $\mathscr{D}_{XY}$.

**Condition 3** (Compatibility). *For any equivalence class $[D'_{XY}]$ with respect to $\mathscr{D}_{XY}$ and any $\epsilon > 0$, there exists a hypothesis function $h_\epsilon \in \mathcal{H}$ such that for any domain $D_{XY} \in [D'_{XY}]$,*

$$h_\epsilon \in \{h' \in \mathcal{H} : R_D^{\text{out}}(h') \leq \inf_{h \in \mathcal{H}} R_D^{\text{out}}(h) + \epsilon\} \cap \{h' \in \mathcal{H} : R_D^{\text{in}}(h') \leq \inf_{h \in \mathcal{H}} R_D^{\text{in}}(h) + \epsilon\}.$$

In Appendix F, Lemma 2 has implied that Condition 3 is a general version of Condition 1. Next, Theorem 8 indicates that Condition 3 is the *necessary and sufficient condition* in the space $\mathscr{D}_{XY}^F$.

> **Theorem 8.** *Suppose that $\mathcal{X}$ is a bounded set. OOD detection is learnable in the finite-ID-distribution space $\mathscr{D}_{XY}^F$ for $\mathcal{H}$ **if and only if** the compatibility condition (i.e., Condition 3) holds. Furthermore, the learning rate $\epsilon_{\text{cons}}(n)$ can attain $O(1/\sqrt{n^{1-\theta}})$, for any $\theta \in (0, 1)$.*

Theorem 8 shows that, in the process of algorithm design, OOD detection cannot be successful without the compatibility condition. Theorem 8 also implies that Condition 3 is essential for the learnability of OOD detection. This motivates us to study whether OOD detection can be successful in more general spaces (*e.g.*, the density-based space), when the compatibility condition holds.

**OOD Detection in the Density-based Space.** To ensure that Condition 3 holds, we consider a basic assumption in learning theory—*Realizability Assumption* (see Appendix D.2), *i.e.*, for any $D_{XY} \in \mathscr{D}_{XY}$, there exists $h^* \in \mathcal{H}$ such that $R_D(h^*) = 0$. We discover that in the density-based space $\mathscr{D}_{XY}^{\mu,b}$, Realizability Assumption can conclude the compatibility condition (*i.e.*, Condition 3). Based on this observation, we can prove the following theorem:

> **Theorem 9.** *Given a density-based space $\mathscr{D}_{XY}^{\mu,b}$, if $\mu(\mathcal{X}) < +\infty$, the Realizability Assumption holds, then when $\mathcal{H}$ has finite Natarajan dimension [21], OOD detection is learnable in $\mathscr{D}_{XY}^{\mu,b}$ for $\mathcal{H}$. Furthermore, the learning rate $\epsilon_{\text{cons}}(n)$ can attain $O(1/\sqrt{n^{1-\theta}})$, for any $\theta \in (0, 1)$.*

To further investigate the importance and necessary of Realizability Assumption, Theorem 11 has indicated that in some practical scenarios, Realizability Assumption is the necessary and sufficient condition for the learnability of OOD detection in the density-based space. Therefore, Realizability Assumption may be indispensable for the learnability of OOD detection in some practical scenarios.

# 6 Connecting Theory to Practice

In Section 5, we have shown the successful scenarios where OOD detection problem can be addressed in theory. In this section, we will discuss how the proposed theory is applied to two representative hypothesis spaces—neural-network-based hypothesis spaces and score-based hypothesis spaces.

**Fully-connected Neural Networks.** Given a sequence $\mathbf{q} = (l_1, l_2, ..., l_g)$, where $l_i$ and $g$ are positive integers and $g > 2$, we use $g$ to represent the ***depth*** of neural network and use $l_i$ to represent the ***width*** of the $i$-th layer. After the activation function $\sigma$ is selected[4], we can obtain the architecture of FCNN according to the sequence $\mathbf{q}$. Let $\mathbf{f}_{\mathbf{w},\mathbf{b}}$ be the function generated by FCNN with weights $\mathbf{w}$ and bias $\mathbf{b}$. An FCNN-based scoring function space is defined as: $\mathcal{F}_{\mathbf{q}}^{\sigma} := \{\mathbf{f}_{\mathbf{w},\mathbf{b}} : \forall \text{ weights } \mathbf{w}, \forall \text{ bias } \mathbf{b}\}$. In addition, for simplicity, given any two sequences $\mathbf{q} = (l_1, ..., l_g)$ and $\mathbf{q}' = (l'_1, ..., l'_{g'})$, we use the notation $\mathbf{q} \lesssim \mathbf{q}'$ to represent the following equations and inequalities:

1) $g \leq g', l_1 = l'_1, l_g = l'_{g'}$;    2) $l_i \leq l'_i, \forall i = 1, ..., g-1$;  and  3) $l_{g-1} \leq l'_i, \forall i = g, ..., g'-1$.

In Appendix L, Lemma 10 shows $\mathbf{q} \lesssim \mathbf{q}' \Rightarrow \mathcal{F}_{\mathbf{q}}^{\sigma} \subset \mathcal{F}_{\mathbf{q}'}^{\sigma}$. We use $\lesssim$ to compare the sizes of FCNNs.

**FCNN-based Hypothesis Space.** Let $l_g = K + 1$. The FCNN-based scoring function space $\mathcal{F}_{\mathbf{q}}^{\sigma}$ can induce an FCNN-based hypothesis space. For any $\mathbf{f}_{\mathbf{w},\mathbf{b}} \in \mathcal{F}_{\mathbf{q}}^{\sigma}$, the induced hypothesis function is:

$$h_{\mathbf{w},\mathbf{b}} := \underset{k \in \{1,...,K+1\}}{\arg\max} f_{\mathbf{w},\mathbf{b}}^k, \text{ where } f_{\mathbf{w},\mathbf{b}}^k \text{ is the } k\text{-th coordinate of } \mathbf{f}_{\mathbf{w},\mathbf{b}}.$$

Then, the FCNN-based hypothesis space is defined as $\mathcal{H}_{\mathbf{q}}^{\sigma} := \{h_{\mathbf{w},\mathbf{b}} : \forall \text{ weights } \mathbf{w}, \forall \text{ bias } \mathbf{b}\}$.

**Score-based Hypothesis Space.** Many OOD detection algorithms detect OOD data by using a score-based strategy. That is, given a threshold $\lambda$, a scoring function space $\mathcal{F}_l \subset \{\mathbf{f} : \mathcal{X} \to \mathbb{R}^l\}$ and a scoring function $E : \mathcal{F}_l \to \mathbb{R}$, then $\mathbf{x}$ is regarded as ID data if and only if $E(\mathbf{f}(\mathbf{x})) \geq \lambda$. We introduce several representative scoring functions $E$ as follows: for any $\mathbf{f} = [f^1, ..., f^l]^\top \in \mathcal{F}_l$,
• softmax-based function [7] and temperature-scaled function [8]: $\lambda \in (\frac{1}{l}, 1)$ and $T > 0$,

$$E(\mathbf{f}) = \max_{k \in \{1,...,l\}} \frac{\exp(f^k)}{\sum_{c=1}^{l} \exp(f^c)}, \qquad E(\mathbf{f}) = \max_{k \in \{1,...,l\}} \frac{\exp(f^k/T)}{\sum_{c=1}^{l} \exp(f^c/T)}; \qquad (5)$$

• energy-based function [23]: $\lambda \in (0, +\infty)$ and $T > 0$,

$$E(\mathbf{f}) = T \log \sum_{c=1}^{l} \exp(f^c/T). \qquad (6)$$

Using $E$, $\lambda$ and $\mathbf{f} \in \mathcal{F}_{\mathbf{q}}^{\sigma}$, we have a classifier: $h_{\mathbf{f},E}^{\lambda}(\mathbf{x}) = 1$, if $E(\mathbf{f}(\mathbf{x})) \geq \lambda$; otherwise, $h_{\mathbf{f},E}^{\lambda}(\mathbf{x}) = 2$, where $1$ represents the ID data and $2$ represents the OOD data. Hence, a binary classification hypothesis space $\mathcal{H}^b$, which consists of all $h_{\mathbf{f},E}^{\lambda}$, is generated. We define $\mathcal{H}_{\mathbf{q},E}^{\sigma,\lambda} := \{h_{\mathbf{f},E}^{\lambda} : \forall \mathbf{f} \in \mathcal{F}_{\mathbf{q}}^{\sigma}\}$.

**Learnability of OOD Detection in Different Hypothesis Spaces.** Next, we present applications of our theory regarding the above two practical and important hypothesis spaces $\mathcal{H}_{\mathbf{q}}^{\sigma}$ and $\mathcal{H}_{\mathbf{q},E}^{\sigma,\lambda}$.

**Theorem 10.** *Suppose that Condition 2 holds and the hypothesis space $\mathcal{H}$ is FCNN-based or score-based, i.e., $\mathcal{H} = \mathcal{H}_{\mathbf{q}}^{\sigma}$ or $\mathcal{H} = \mathcal{H}^{\text{in}} \bullet \mathcal{H}^b$, where $\mathcal{H}^{\text{in}}$ is an ID hypothesis space, $\mathcal{H}^b = \mathcal{H}_{\mathbf{q},E}^{\sigma,\lambda}$ and $\mathcal{H} = \mathcal{H}^{\text{in}} \bullet \mathcal{H}^b$ is introduced below Eq. (4), here $E$ is introduced in Eqs. (5) or (6). Then*

> *There is a sequence $\mathbf{q} = (l_1, ..., l_g)$ such that OOD detection is learnable in the separate space $\mathscr{D}_{XY}^s$ for $\mathcal{H}$ **if and only if** $|\mathcal{X}| < +\infty$.*

*Furthermore, if $|\mathcal{X}| < +\infty$, then there exists a sequence $\mathbf{q} = (l_1, ..., l_g)$ such that for any sequence $\mathbf{q}'$ satisfying that $\mathbf{q} \lesssim \mathbf{q}'$, OOD detection is learnable in $\mathscr{D}_{XY}^s$ for $\mathcal{H}$.*

Theorem 10 states that 1) when the hypothesis space is FCNN-based or score-based, the finite feature space is the necessary and sufficient condition for the learnability of OOD detection in the separate space; and 2) a larger architecture of FCNN has a greater probability to achieve the learnability of

---

[4]We consider the *rectified linear unit* (ReLU) function as the default activation function $\sigma$, which is defined by $\sigma(x) = \max\{x, 0\}, \forall x \in \mathbb{R}$. *We will not repeatedly mention the definition of $\sigma$ in the rest of our paper.*

OOD detection in the separate space. Note that when we select Eqs. (5) or (6) as the scoring function $E$, Theorem 10 also shows that the selected scoring functions $E$ can guarantee the learnability of OOD detection, which is a theoretical support for the representative works [8, 23, 7]. Furthermore, Theorem 11 also offers theoretical supports for these works in the density-based space, when $K = 1$.

**Theorem 11.** *Suppose that each domain $D_{XY}$ in $\mathscr{D}_{XY}^{\mu,b}$ is attainable, i.e., $\arg\min_{h \in \mathcal{H}} R_D(h) \neq \emptyset$ (the finite discrete domains satisfy this). Let $K = 1$ and the hypothesis space $\mathcal{H}$ be score-based ($\mathcal{H} = \mathcal{H}_{\mathbf{q},E}^{\sigma,\lambda}$, where $E$ is in Eqs. (5) or (6)) or FCNN-based ($\mathcal{H} = \mathcal{H}_{\mathbf{q}}^{\sigma}$). If $\mu(\mathcal{X}) < +\infty$, then the following four conditions are **equivalent**:*

> *Learnability in $\mathscr{D}_{XY}^{\mu,b}$ for $\mathcal{H}$ $\iff$ Condition 1 $\iff$ Realizability Assumption $\iff$ Condition 3*

Theorem 11 still holds if the function space $\mathcal{F}_{\mathbf{q}}^{\sigma}$ is generated by Convolutional Neural Network.

**Overlap and Benefits of Multi-class Case.** We investigate when the hypothesis space is FCNN-based or score-based, what will happen if there exists an overlap between the ID and OOD distributions?

**Theorem 12.** *Let $K = 1$ and the hypothesis space $\mathcal{H}$ be score-based ($\mathcal{H} = \mathcal{H}_{\mathbf{q},E}^{\sigma,\lambda}$, where $E$ is in Eqs. (5) or (6)) or FCNN-based ($\mathcal{H} = \mathcal{H}_{\mathbf{q}}^{\sigma}$). Given a prior-unknown space $\mathscr{D}_{XY}$, if there exists a domain $D_{XY} \in \mathscr{D}_{XY}$, which has an overlap between ID and OOD distributions (see Definition 4), then OOD detection is not learnable in the domain space $\mathscr{D}_{XY}$ for $\mathcal{H}$.*

When $K = 1$ and the hypothesis space is FCNN-based or score-based, Theorem 12 shows that overlap between ID and OOD distributions is the sufficient condition for the unlearnability of OOD detection. Theorem 12 takes roots in the conditions $\inf_{h \in \mathcal{H}} R_D^{\text{in}}(h) = 0$ and $\inf_{h \in \mathcal{H}} R_D^{\text{out}}(h) = 0$. However, when $K > 1$, we can ensure $\inf_{h \in \mathcal{H}} R_D^{\text{in}}(h) > 0$ if ID distribution $D_{X_{\mathrm{I}}Y_{\mathrm{I}}}$ has overlap between ID classes. By this observation, we conjecture that when $K > 1$, OOD detection is learnable in some special cases where overlap exists, even if the hypothesis space is FCNN-based or score-based.

## 7 Discussion

**Understanding Far-OOD Detection.** Many existing works [7, 39] study the far-OOD detection issue. Existing benchmarks include 1) MNIST [40] as ID dataset, and Texture [41], CIFAR-10 [42] or Place365 [43] as OOD datasets; and 2) CIFAR-10 [42] as ID dataset, and MNIST [40], or Fashion-MNIST [43] as OOD datasets. In far-OOD case, we find that the ID and OOD datasets have different semantic labels and different styles. From the theoretical view, we can define far-OOD detection tasks as follows: for $\tau > 0$, a domain space $\mathscr{D}_{XY}$ is $\tau$-far-OOD, if for any domain $D_{XY} \in \mathscr{D}_{XY}$,

$$\text{dist}(\text{supp}D_{X_{\mathrm{O}}}, \text{supp}D_{X_{\mathrm{I}}}) > \tau.$$

Theorems 7, 8 and 10 imply that under appropriate hypothesis space, $\tau$-far-OOD detection is learnable. In Theorem 7, the condition $|\mathcal{X}| < +\infty$ is necessary for the separate space. However, one can prove that in the far-OOD case, when $\mathcal{H}^{\text{in}}$ is agnostic PAC learnable for ID distribution, the results in Theorem 7 still holds, if the condition $|\mathcal{X}| < +\infty$ is replaced by a weaker condition that $\mathcal{X}$ is compact. In addition, it is notable that when $\mathcal{H}^{\text{in}}$ is agnostic PAC learnable for ID distribution and $\mathcal{X}$ is compact, the KNN-based OOD detection algorithm [44] is consistent in the $\tau$-far-OOD case.

**Understanding Near-OOD Detection.** When the ID and OOD datasets have similar semantics or styles, OOD detection tasks become more challenging. [45, 46] consider this issue and name it near-OOD detection. Existing benchmarks include 1) MNIST [40] as ID dataset, and Fashion-MNIST [43] or Not-MNIST [47] as OOD datasets; and 2) CIFAR-10 [42] as ID dataset, and CIFAR-100 [48] as OOD dataset. From the theoretical view, some near-OOD tasks may imply the overlap condition, *i.e.* Definition 4. Therefore, Theorems 3 and 12 imply that near-OOD detection may be not learnable. Developing a theory to understand the feasibility of near-OOD detection is still an *open question*.

**Understanding One-class Novelty Detection.** In one-class novelty detection and semantic anomaly detection (*i.e.* $K = 1$), Theorem 6 has revealed that it is necessary to use a large-capacity model to ensure the good generalization in the separate space. Theorem 3 and Theorem 12 suggest that we should try to avoid the overlap between ID and OOD distributions in the one-class case. If the overlap cannot be avoided, we suggest considering the multi-class OOD detection instead of the one-class case. Additionally, in the density-based space, Theorem 11 has shown that it is necessary to select a suitable hypothesis space satisfying the Realizability Assumption to ensure the learnability of OOD

detection in the density-based space. Generally, a large-capacity model can be helpful to guarantee that the Realizability Assumption holds.

## 8 Related Work

We briefly review the related theoretical works below. See Appendix A for detailed related works.

**OOD Detection Theory.** [49] understands the OOD detection via goodness-of-fit tests and typical set hypothesis, and argues that minimal density estimation errors can lead to OOD detection failures without assuming an overlap between ID and OOD distributions. Beyond [49], [50] paves a new avenue to designing provable OOD detection algorithms. Compared to [50, 49], our theory focuses on the PAC learnable theory of OOD detection and identifies several necessary and sufficient conditions for the learnability of OOD detection, opening a door to study OOD detection in theory.

**Open-set Learning Theory.** [51] and [29, 52] propose the agnostic PAC learning bounds for open-set detection and open-set domain adaptation, respectively. Unfortunately, [29, 51, 52] all require that the test data are indispensable during the training process. To investigate open-set learning (OSL) *without accessing the test data* during training, [24] proposes and investigates the *almost* agnostic PAC learnability for OSL. However, the assumptions used in [24] are very strong and unpractical.

**Learning Theory for Classification with Reject Option.** Many works [53, 54] also investigate the *classification with reject option* (CwRO) problem, which is similar to OOD detection in some cases. [55, 56, 57, 58, 59] study the learning theory and propose the PAC learning bounds for CwRO. However, compared to our work regarding OOD detection, existing CwRO theories mainly focus on how the ID risk $R_D^{\text{in}}$ (*i.e.*, the risk that ID data is wrongly classified) is influenced by special rejection rules. Our theory not only focuses on the ID risk, but also pays attention to the OOD risk.

**Robust Statistics.** In the field of robust statistics [60], researchers aim to propose estimators and testers that can mitigate the negative effects of outliers (similar to OOD data). The proposed estimators are supposed to be independent of the potentially high dimensionality of the data [61, 62, 63]. Existing works [64, 65, 66] in the field have identified and resolved the statistical limits of outlier robust statistics by constructing estimators and proving impossibility results. In the future, it is a promising and interesting research direction to study the robustness of OOD detection based on robust statistics.

**PQ Learning Theory.** Under some conditions, PQ learning theory [67, 68] can be regarded as the PAC theory for OOD detection in the semi-supervised or transductive learning cases, *i.e.*, test data are required during training. Besides, [67, 68] aim to give the PAC estimation under Realizability Assumption [21]. Our theory does not only study the PAC estimation in the realization cases, but also studies the other cases, which are more difficult than PAC theory under Realizability Assumption.

## 9 Conclusions and Future Works

Detecting OOD data has shown its significance in improving the reliability of machine learning. However, very few works discuss OOD detection in theory, which hinders real-world applications of OOD detection algorithms. In this paper, we are the *first* to provide the PAC theory for OOD detection. Our results imply that we cannot expect a universally consistent algorithm to handle all scenarios in OOD detection. Yet, it is still possible to make OOD detection learnable in certain scenarios. For example, when we design OOD detection algorithms, we normally only have finite ID datasets. In this real scenario, Theorem 8 provides a necessary and sufficient condition for the success of OOD detection. Our theory reveals many necessary and sufficient conditions for the learnability of OOD detection, hence *opening a door* to studying the learnability of OOD detection. In the future, we will focus on studying the robustness of OOD detection based on robust statistics [64, 69].

## Acknowledgment

JL and ZF were supported by the Australian Research Council (ARC) under FL190100149. YL is supported by the AFOSR Young Investigator Program Award. BH was supported by the RGC Early Career Scheme No. 22200720 and NSFC Young Scientists Fund No. 62006202. ZF would also like to thank Prof. Peter Bartlett and Dr. Tongliang Liu for productive discussions.

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
