# Table of Contents of Appendix

# A  Detailed Related Work

**OOD Detection Algorithms.** We will briefly review many representative OOD detection algorithms in three categories. 1) Classification-based methods use an ID classifier to detect OOD data [7][5]. Representative works consider using the maximum softmax score [7], temperature-scaled score [14] and energy-based score [23, 71] to identify OOD data. 2) Density-based methods aim to estimate an ID distribution and identify the low-density area as OOD data [10]. 3) The recent development of generative models provides promising ways to make them successful in OOD detection [11, 12, 14, 72, 73]. Distance-based methods are based on the assumption that OOD data should be relatively far away from the centroids of ID classes [9], including Mahalanobis distance [9, 45], cosine similarity [74], and kernel similarity [75].

Early works consider using the maximum softmax score to express the ID-ness [7]. Then, temperature scaling functions are used to amplify the separation between the ID and OOD data [14]. Recently, researchers propose hyperparameter-free energy scores to improve the OOD uncertainty estimation [23, 71]. Additionally, researchers also consider using the information contained in gradients to help improve the performance of OOD detection [18].

Except for the above algorithms, researchers also study the situation, where auxiliary OOD data can be obtained during the training process [13, 70]. These methods are called outlier exposure, and have much better performance than the above methods due to the appearance of OOD data. However, the exposure of OOD data is a strong assumption [4]. Thus, researchers also consider generating OOD data to help the separation of OOD and ID data [76]. In this paper, we do not make an assumption that OOD data are available during training, since this assumption may not hold in real world.

**OOD Detection Theory.** [49] rejects the typical set hypothesis, the claim that relevant OOD distributions can lie in high likelihood regions of data distribution, as implausible. [49] argues that minimal density estimation errors can lead to OOD detection failures without assuming an overlap between ID and OOD distributions. Compared to [49], our theory focuses on the PAC learnable theory of OOD detection. If detectors are generated by FCNN, our theory (Theorem 12) shows that the overlap is the sufficient condition to the failure of learnability of OOD detection, which is complementary to [49]. In addition, we identify several necessary and sufficient conditions for the learnability of OOD detection, which opens a door to studying OOD detection in theory. Beyond [49], [50] paves a new avenue to designing provable OOD detection algorithms. Compared to [50], our paper aims to characterize the learnability of OOD detection to answer the question: is OOD detection PAC learnable?

**Open-set Learning Theory.** [51] is the first to propose the agnostic PAC guarantees for open-set detection. Unfortunately, the test data must be used during the training process. [29] considers the open-set domain adaptation (OSDA) [52] and proposes the first learning bound for OSDA. [29] mainly depends on the positive-unlabeled learning techniques [77, 78, 79]. However, similar to [51], the test data must be available during training. To study open-set learning (OSL) *without accessing the test data* during training, [24] proposes and studies the almost PAC learnability for OSL, which is motivated by transfer learning [80, 81]. In our paper, we study the PAC learnability for OOD detection, which is an open problem proposed by [24].

**Learning Theory for Classification with Reject Option.** Many works [53, 54] also investigate the *classification with reject option* (CwRO) problem, which is similar to OOD detection in some cases. [55, 56, 57, 58, 59] study the learning theory and propose the agnostic PAC learning bounds for CwRO. However, compared to our work regarding OOD detection, existing CwRO theories mainly focus on how the ID risk (*i.e.*, the risk that ID data is wrongly classified) is influenced by special rejection rules. Our theory not only focuses on the ID risk, but also pays attention to the OOD risk.

**Robust Statistics.** In the field of robust statistics [60], researchers aim to propose estimators and testers that can mitigate the negative effects of outliers (similar to OOD data). The proposed estimators are supposed to be independent of the potentially high dimensionality of the data [61, 62, 63]. Existing works [64, 65, 66] in the field have identified and resolved the statistical limits of outlier robust statistics by constructing estimators and proving impossibility results. In the future, it is a promising and interesting research direction to study the robustness of OOD detection based on robust statistics.

---

[5]Note that, some methods assume that OOD data are available in advance [13, 70]. However, the exposure of OOD data is a strong assumption [4]. We do not consider this situation in our paper.

**PQ Learning Theory.** Under some conditions, PQ learning theory [67, 68] can be regarded as the PAC theory for OOD detection in the semi-supervised or transductive learning cases, *i.e.*, test data are required during the training process. Additionally, PQ learning theory in [67, 68] aims to give the PAC estimation under Realizability Assumption [21]. Our theory focuses on the PAC theory in different cases, which is more difficult and more practical than PAC theory under Realizability Assumption.

## B  Limitations and Potential Negative Societal Impacts

**Limitations.** The main limitation of our work lies in that we do not answer the most general question:

*Given any hypothesis space $\mathcal{H}$ and space $\mathscr{D}_{XY}$, what is the necessary and sufficient condition to ensure the PAC learnability of OOD detection?*

However, this question is still difficult to be addressed, due to limited mathematical skills. Yet, based on our observations and the main results in our paper, we believe the following result may hold:

**Conjecture**: *If $\mathcal{H}$ is agnostic learnable for supervised learning, then OOD detection is learnable in $\mathscr{D}_{XY}$ **if and only if** compatibility condition (i.e., Condition 3) holds.*

We leave this question as a future work.

**Potential Negative Societal Impacts.** Since our paper is a theoretical paper and the OOD detection problem is significant to ensure the safety of deploying existing machine learning algorithms, there are no potential negative societal impacts in our paper.

## C  Discussions and Details about Experiments in Figure 1

In this section, we summarize our main results, then give the details of the experiments in Figure 1.

### C.1  Summary

We summarize our main results as follows:

• A necessary condition (*i.e.*, Condition 1) for the learnability of OOD detection is proposed. Theorem 2 shows that Condition 1 is the *necessary and sufficient condition* for the learnability of OOD detection, when the domain space is the single-distribution space $\mathscr{D}_{XY}^{D_{XY}}$. This implies the Condition 1 is the necessary condition for the learnability of OOD detection.

• Theorem 3 has shown that the overlap between ID and OOD data can lead the failures of OOD detection under some mild assumptions. Furthermore, Theorem 12 shows that when $K = 1$, the overlap is the sufficient condition for the failures of OOD detection, when the hypothesis space is FCNN-based or score-based.

• Theorem 4 provides an impossibility theorem for the total space $\mathscr{D}_{XY}^{\mathrm{all}}$. OOD detection is not learnable in $\mathscr{D}_{XY}^{\mathrm{all}}$ for any non-trivial hypothesis space.

• Theorem 5 gives impossibility theorems for the separate space $\mathscr{D}_{XY}^{\mathrm{s}}$. To ensure the impossibility theorems hold, mild assumptions are required. Theorem 5 also implies that OOD detection may be learnable in the separate space $\mathscr{D}_{XY}^{\mathrm{s}}$, if the feature space is finite, *i.e.*, $|\mathcal{X}| < +\infty$. Additionally, Theorem 10 implies that the finite feature space may be the necessary condition to ensure the learnability of OOD detection in the separate space.

• When $|\mathcal{X}| < +\infty$ and $K = 1$, Theorem 6 provides the *necessary and sufficient condition* for the learnability of OOD detection in the separate space $\mathscr{D}_{XY}^{\mathrm{s}}$. Theorem 6 implies that if the OOD detection can be learnable in the distribution-agnostic case, then a large-capacity model is necessary. Based on Theorem 6, Theorem 7 studies the learnability in the $K > 1$ case.

• The compatibility condition (*i.e.*, Condition 3) for the learnability of OOD detection is proposed. Theorem 8 shows that Condition 3 is the *necessary and sufficient condition* for the learnability of OOD detection in the finite-ID-distribution space $\mathscr{D}_{XY}^{F}$. This also implies Condition 3 is the necessary

condition for any prior-unknown space. Note that we can only collect finite ID datasets to build models. Hence, Theorem 8 can handle the most practical scenarios.

- To further understand the importance of the compatibility condition (Condition 3). Theorem 9 considers the density-based space $\mathscr{D}_{XY}^{\mu,b}$. We discover that Realizability Assumption implies the compatibility condition in the density-based space. Based on this observation, we prove that OOD detection is learnable in $\mathscr{D}_{XY}^{\mu,b}$ under Realizability Assumption.

- Theorem 10 gives practical applications of our theory. In this theorem, we discover that the finite feature space is a *necessary and sufficient condition* for the learnability of OOD detection in the separate space $\mathscr{D}_{XY}^s$, when the hypothesis space is FCNN-based or score-based.

- Theorem 11 has shown that when $K = 1$ and the hypothesis space is FCNN-based or score-based, Realizability Assumption, Condition 3, Condition 1 and the learnability of OOD detection in the density-based space $\mathcal{D}_{XY}^{\mu,b}$ are all *equivalent*.

- **Meaning of Our Theory.** In classical statistical learning theory, the generalization theory guarantees that a well-trained classifier can be generalized well on the test set as long as the training and test sets are from the same distribution [21, 22]. However, since the OOD data are unseen during the training process, it is very difficult to determine whether the generalization theory holds for OOD detection.

Normally, OOD data are unseen and can be various. We hope that there exists an algorithm that can be used for the various OOD data instead of some certain OOD data, which is the reason why the generalization theory for OOD detection needs to be developed. In this paper, we investigate the generalization theory regarding OOD detection and point out when the OOD detection can be successful. Our theory is based on the PAC learning theory. The impossibility theorems and the given necessary and sufficient conditions outlined provide important perspectives from which to think about OOD detection.

## C.2 Details of Experiments in Figure 1

In this subsection, we present details of the experiments in Figure 1, including data generation, configuration and OOD detection procedure.

**Data Generation.** ID and OOD data are drawn from the following *uniform* (U) distributions (note that we use U($\mathbf{I}$) to present the uniform distribution in region $\mathbf{I}$).
- The marginal distribution of ID distribution for class $c$: for any $c \in \{1, ..., 10\}$,

$$D_{X_I|Y_I=c} = \mathrm{U}(\mathbf{I}_c), \text{ where } \mathbf{I}_c = [d_c, d_c + 4] \times [1, 5], \tag{7}$$

here $d_i = 5 + \mathrm{gap}_{II} * (i-1) + 4(i-2)$ and $\mathrm{gap}_{II}$ is a positive constant.
- The class-prior probability for class $c$: for any $c \in \{1, ..., 10\}$,

$$D_{Y_I}(y = c) = \frac{1-\alpha}{10}.$$

- The marginal distribution of OOD distribution:

$$D_{X_O} = \mathrm{U}(\mathbf{I}_{out}), \text{ where } \mathbf{I}_{out} = [d_1 - 1, d_{10} + 5] \times [5 + \mathrm{gap}_{IO}, 10 + \mathrm{gap}_{IO}]. \tag{8}$$

Figure 2 shows the OOD and ID distributions, when $\mathrm{gap}_{II} = 20$ and $\mathrm{gap}_{IO} = -2$. In Figure 1, we draw $n$ data from ID distribution ($n = 15,000, 20,000, 25,000$) and $25,000$ data from the OOD distribution.

**Configuration.** The architecture of ID classifier is a four-layer FCNN. The number of neurons in hidden layers is set to 100, and the number of neurons of output layer is set to 10. These neurons use sigmoid activations. We use the Adam optimizer [82] to optimize the network's parameters (with the $\ell_2$ loss). The learning rate is set to 0.001, and the max number of training iterations is set to $10,000$. Within each iteration, we use full batch to update the network's parameters. $\mathrm{gap}_{II}$ is set to 20 in our experiments. In Figure 1b, $\mathrm{gap}_{IO} = -2$ (the overlap exists, see Figure 2), and in Figure 1c, $\mathrm{gap}_{IO} = 100$ (no overlap).

**OOD Detection Procedure.** We first train an ID classifier with $n$ data drawn from the ID distribution. Then, according to [23], we apply the free-energy score to identify the OOD data and calculate the

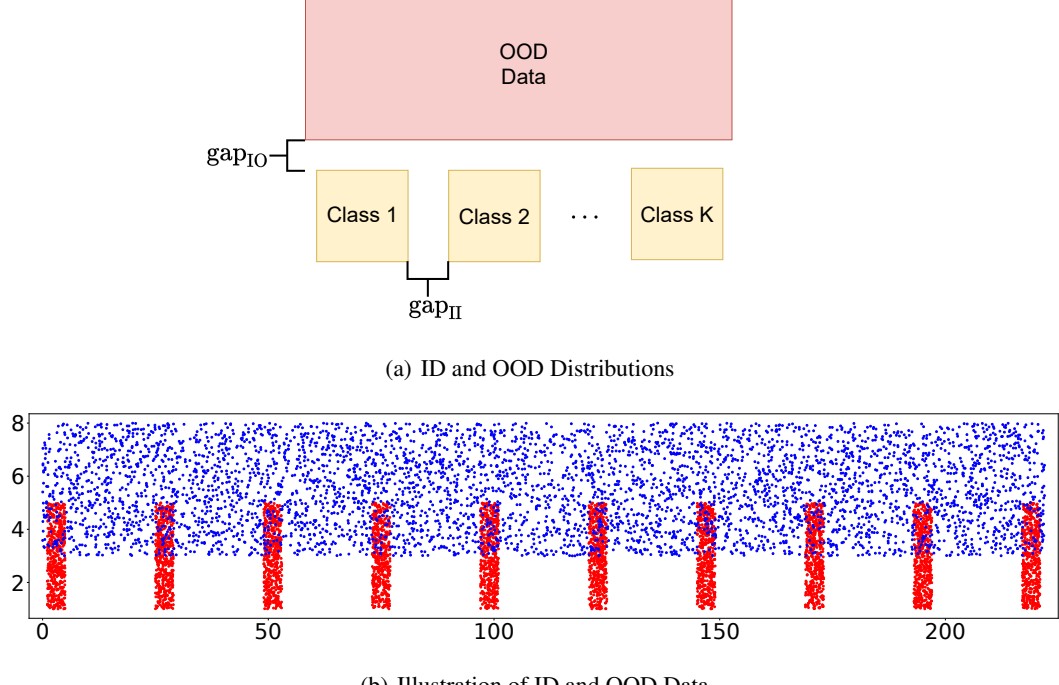

(a) ID and OOD Distributions

(b) Illustration of ID and OOD Data

Figure 2: ID and OOD distributions in Figure 1.

$\alpha$-risk (with the 0-1 loss). We repeat the above detection procedure 20 times and report the average $\alpha$-risk in Figure 1. Note that, following [23], we choose the threshold used by the free-energy method so that $95\%$ of ID data are correctly identified as the ID classes by the OOD detector.

# D Notations

## D.1 Main Notations and Their Descriptions

In this section, we summarize important notations in Table 1.

Table 1: Main notations and their descriptions.

| Notation | Description |
|---|---|
| **• Spaces and Labels** | |
| $d$ and $\mathcal{X} \subset \mathbb{R}^d$ | the feature dimension of data point and feature space |
| $\mathcal{Y}$ | ID label space $\{1, ..., K\}$ |
| $K+1$ | $K+1$ represents the OOD labels |
| $\mathcal{Y}_{\mathrm{all}}$ | $\mathcal{Y} \cup \{K+1\}$ |
| **• Distributions** | |
| $X_{\mathrm{I}}, X_{\mathrm{O}}, Y_{\mathrm{I}}, Y_{\mathrm{O}}$ | ID feature, OOD feature, ID label, OOD label random variables |
| $D_{X_{\mathrm{I}} Y_{\mathrm{I}}}, D_{X_{\mathrm{O}} Y_{\mathrm{O}}}$ | ID joint distribution and OOD joint distribution |
| $D_{XY}^\alpha$ | $D_{XY}^\alpha = (1-\alpha)D_{X_{\mathrm{I}} Y_{\mathrm{I}}} + \alpha D_{X_{\mathrm{O}} Y_{\mathrm{O}}}, \ \forall \alpha \in [0,1]$ |
| $\pi^{\mathrm{out}}$ | class-prior probability for OOD distribution |
| $D_{XY}$ | $D_{XY} = (1-\pi^{\mathrm{out}})D_{X_{\mathrm{I}} Y_{\mathrm{I}}} + \pi^{\mathrm{out}} D_{X_{\mathrm{O}} Y_{\mathrm{O}}}$, called domain |
| $D_{X_{\mathrm{I}}}, D_{X_{\mathrm{O}}}, D_X$ | marginal distributions for $D_{X_{\mathrm{I}} Y_{\mathrm{I}}}, D_{X_{\mathrm{O}} Y_{\mathrm{O}}}$ and $D_{XY}$, respectively |
| **• Domain Spaces** | |
| $\mathscr{D}_{XY}$ | domain space consisting of some domains |
| $\mathscr{D}_{XY}^{\mathrm{all}}$ | total space |
| $\mathscr{D}_{XY}^s$ | seperate space |
| $\mathscr{D}_{XY}^{D_{XY}}$ | single-distribution space |
| $\mathscr{D}_{XY}^F$ | finite-ID-distribution space |
| $\mathscr{D}_{XY}^{\mu,b}$ | density-based space |
| **• Loss Function, Function Spaces** | |
| $\ell(\cdot, \cdot)$ | loss: $\mathcal{Y}_{\mathrm{all}} \times \mathcal{Y}_{\mathrm{all}} \to \mathbb{R}_{\geq 0}$: $\ell(y_1, y_2) = 0$ if and only if $y_1 = y_2$ |
| $\mathcal{H}$ | hypothesis space |
| $\mathcal{H}^{\mathrm{in}}$ | ID hypothesis space |
| $\mathcal{H}^{\mathrm{b}}$ | hypothesis space in binary classification |
| $\mathcal{F}_l$ | scoring function space consisting some $l$ dimensional vector-valued functions |
| **• Risks and Partial Risks** | |
| $R_D(h)$ | risk corresponding to $D_{XY}$ |
| $R_D^{\mathrm{in}}(h)$ | partial risk corresponding to $D_{X_{\mathrm{I}} Y_{\mathrm{I}}}$ |
| $R_D^{\mathrm{out}}(h)$ | partial risk corresponding to $D_{X_{\mathrm{O}} Y_{\mathrm{O}}}$ |
| $R_D^\alpha(h)$ | $\alpha$-risk corresponding to $D_{XY}^\alpha$ |
| **• Fully-Connected Neural Networks** | |
| $\mathbf{q}$ | a sequence $(l_1, ..., l_g)$ to represent the architecture of FCNN |
| $\sigma$ | activation function. In this paper, we use ReLU function |
| $\mathcal{F}_{\mathbf{q}}^\sigma$ | FCNN-based scoring function space |
| $\mathcal{H}_{\mathbf{q}}^\sigma$ | FCNN-based hypothesis space |
| $\mathbf{f}_{\mathbf{w}, \mathbf{b}}$ | FCNN-based scoring function, which is from $\mathcal{F}_{\mathbf{q}}^\sigma$ |
| $h_{\mathbf{w}, \mathbf{b}}$ | FCNN-based hypothesis function, which is from $\mathcal{H}_{\mathbf{q}}^\sigma$ |
| **• Score-based Hypothesis Space** | |
| $E$ | scoring function |
| $\lambda$ | threshold |
| $\mathcal{H}_{\mathbf{q}, E}^{\sigma, \lambda}$ | score-based hypothesis space—a binary classification space |
| $h_{\mathbf{f}, E}^\lambda$ | score-based hypothesis function—a binary classifier |

Given $\mathbf{f} = [f^1, ..., f^l]^\top$, for any $\mathbf{x} \in \mathcal{X}$,

$$\arg\max_{k \in \{1,...,l\}} f^k(\mathbf{x}) := \max\{k \in \{1, ..., l\} : f^k(\mathbf{x}) \geq f^i(\mathbf{x}), \forall i = 1, ..., l\},$$

where $f^k$ is the $k$-th coordinate of $\mathbf{f}$ and $f^i$ is the $i$-th coordinate of $\mathbf{f}$. The above definition about $\arg\max$ aims to overcome some special cases. For example, there exist $k_1, k_2$ ($k_1 < k_2$) such that $f^{k_1}(\mathbf{x}) = f^{k_2}(\mathbf{x})$ and $f^{k_1}(\mathbf{x}) > f^i(\mathbf{x}), f^{k_2}(\mathbf{x}) > f^i(\mathbf{x}), \forall i \in \{1, ..., l\} - \{k_1, k_2\}$. Then, according to the above definition, $k_2 = \arg\max_{k \in \{1,...,l\}} f^k(\mathbf{x})$.

## D.2 Realizability Assumption

**Assumption 2** (Realizability Assumption). *A domain space $\mathscr{D}_{XY}$ and hypothesis space $\mathcal{H}$ satisfy the Realizability Assumption, if for each domain $D_{XY} \in \mathscr{D}_{XY}$, there exists at least one hypothesis function $h^* \in \mathcal{H}$ such that $R_D(h^*) = 0$.*

## D.3 Learnability and PAC learnability

Here we give a proof to show that Learnability given in Definition 1 and PAC learnability are equivalent.

**First**, we prove that Learnability concludes the PAC learnability.

According to Definition 1,

$$\mathbb{E}_{S \sim D_{X_I Y_I}^n} R_D(\mathbf{A}(S)) \leq \inf_{h \in \mathcal{H}} R_D(h) + \epsilon_{\text{cons}}(n),$$

which implies that

$$\mathbb{E}_{S \sim D_{X_I Y_I}^n} [R_D(\mathbf{A}(S)) - \inf_{h \in \mathcal{H}} R_D(h)] \leq \epsilon_{\text{cons}}(n).$$

Note that $R_D(\mathbf{A}(S)) - \inf_{h \in \mathcal{H}} R_D(h) \geq 0$. Therefore, by Markov's inequality, we have

$$\mathbb{P}(R_D(\mathbf{A}(S)) - \inf_{h \in \mathcal{H}} R_D(h) < \epsilon) > 1 - \mathbb{E}_{S \sim D_{X_I Y_I}^n} [R_D(\mathbf{A}(S)) - \inf_{h \in \mathcal{H}} R_D(h)] / \epsilon \geq 1 - \epsilon_{\text{cons}}(n) / \epsilon.$$

Because $\epsilon_{\text{cons}}(n)$ is monotonically decreasing, we can find a smallest $m$ such that $\epsilon_{\text{cons}}(m) \geq \epsilon \delta$ and $\epsilon_{\text{cons}}(m - 1) < \epsilon \delta$, for $\delta \in (0, 1)$. We define that $m(\epsilon, \delta) = m$. Therefore, for any $\epsilon > 0$ and $\delta \in (0, 1)$, there exists a function $m(\epsilon, \delta)$ such that when $n > m(\epsilon, \delta)$, with the probability at least $1 - \delta$, we have

$$R_D(\mathbf{A}(S)) - \inf_{h \in \mathcal{H}} R_D(h) < \epsilon,$$

which is the definition of PAC learnability.

**Second**, we prove that the PAC learnability concludes Learnability.

PAC-learnability: for any $\epsilon > 0$ and $0 < \delta < 1$, there exists a function $m(\epsilon, \delta) > 0$ such that when the sample size $n > m(\epsilon, \delta)$, we have that with the probability at least $1 - \delta > 0$,

$$R_D(\mathbf{A}(S)) - \inf_{h \in \mathcal{H}} R_D(h) \leq \epsilon.$$

Note that the loss $\ell$ defined in Section 2 has upper bound (because $\mathcal{Y} \cup \{K + 1\}$ is a finite set). We assume the upper bound of $\ell$ is $M$. Hence, according to the definition of PAC-learnability, when the sample size $n > m(\epsilon, \delta)$, we have that

$$\mathbb{E}_S[R_D(\mathbf{A}(S)) - \inf_{h \in \mathcal{H}} R_D(h)] \leq \epsilon(1 - \delta) + 2M\delta < \epsilon + 2M\delta.$$

If we set $\delta = \epsilon$, then when the sample size $n > m(\epsilon, \epsilon)$, we have that

$$\mathbb{E}_S[R_D(\mathbf{A}(S)) - \inf_{h \in \mathcal{H}} R_D(h)] < (2M + 1)\epsilon,$$

this implies that

$$\lim_{n \to +\infty} \mathbb{E}_S[R_D(\mathbf{A}(S)) - \inf_{h \in \mathcal{H}} R_D(h)] = 0,$$

which implies the Learnability in Definition 1. We have completed this proof.

## D.4 Explanations for Some Notations in Section 2

First, we explain the concept that $S \sim D_{X_I Y_I}^n$ in Eq. (2).

$S = \{(\mathbf{x}^1, y^1), ..., (\mathbf{x}^n, y^n)\}$ is training data drawn independent and identically distributed from $D_{X_I Y_I}$.

$D_{X_I Y_I}^n$ denotes the probability over $n$-tuples induced by applying $D_{X_I Y_I}$ to pick each element of the tuple independently of the other members of the tuple.

Because these samples are i.i.d. drawn $n$ times, researchers often use "$S \sim D_{X_I Y_I}^n$" to represent a sample set $S$ (of size $n$) whose each element is drawn i.i.d. from $D_{X_I Y_I}$.

Second, we explain the concept "+" in $(1 - \pi^{\text{out}}) D_{X_I} + \pi^{\text{out}} D_{X_O}$.

For convenience, let $P = (1 - \pi^{\text{out}}) D_{X_I}$ and $Q = \pi^{\text{out}} D_{X_O}$. It is clear that $P$ and $Q$ are measures. Then $P + Q$ is also a measure, which is defined as follows: for any measurable set $A \subset \mathcal{X}$, we have

$$(P + Q)(A) = P(A) + Q(A).$$

For example, when $P$ and $Q$ are discrete measures, then $P + Q$ is also discrete measure: for any $\mathbf{x} \in \mathcal{X}$,

$$(P + Q)(\mathbf{x}) = P(\mathbf{x}) + Q(\mathbf{x}).$$

When $P$ and $Q$ are continuous measures with density functions $f$ and $g$, then $P + Q$ is also continuous measure with density function $f + g$: for any measurable $A \subset \mathcal{X}$,

$$P(A) = \int_A f(\mathbf{x}) d\mathbf{x}, \quad Q(A) = \int_A g(\mathbf{x}) d\mathbf{x},$$

then

$$(P + Q)(A) = \int_A f(\mathbf{x}) + g(\mathbf{x}) d\mathbf{x}.$$

Third, we explain the concept $\mathbb{E}_{(\mathbf{x}, y) \sim D_{XY}} \ell(h(\mathbf{x}), y)$.

The concept $\mathbb{E}_{(\mathbf{x}, y) \sim D_{XY}} \ell(h(\mathbf{x}), y)$ can be computed as follows:

$$\mathbb{E}_{(\mathbf{x}, y) \sim D_{XY}} \ell(h(\mathbf{x}), y) = \int_{\mathcal{X} \times \mathcal{Y}_{\text{all}}} \ell(h(\mathbf{x}), y) dD_{XY}(\mathbf{x}, y).$$

For example, when $D_{XY}$ is a finite discrete distribution: let $\mathcal{Z} = \{(\mathbf{x}^1, y^1), ..., (\mathbf{x}^m, y^m)\}$ be the support set of $D_{XY}$, and assume that $a^i$ is the probability for $(\mathbf{x}^i, y^i)$, i.e., $a^i = D_{XY}(\mathbf{x}^i, y^i)$. Then

$$\mathbb{E}_{(\mathbf{x}, y) \sim D_{XY}} \ell(h(\mathbf{x}), y) = \int_{\mathcal{X} \times \mathcal{Y}_{\text{all}}} \ell(h(\mathbf{x}), y) dD_{XY}(\mathbf{x}, y)$$
$$= \frac{1}{m} \sum_{i=1}^{m} a^i \ell(h(\mathbf{x}^i), y^i).$$

When $D_X$ is a continuous distribution with density $f$, and $D_{Y|X}(Y = k | X = \mathbf{x})$ ($k$-th class-conditional distribution for $\mathbf{x}$) is $a^k(\mathbf{x})$, then

$$\mathbb{E}_{(\mathbf{x}, y) \sim D_{XY}} \ell(h(\mathbf{x}), y) = \int_{\mathcal{X} \times \mathcal{Y}_{\text{all}}} \ell(h(\mathbf{x}), y) dD_{XY}(\mathbf{x}, y)$$
$$= \int_{\mathcal{X}} \sum_{k=1}^{K+1} \ell(h(\mathbf{x}), k) f(\mathbf{x}) a^k(\mathbf{x}) d\mathbf{x},$$

where $D_{Y|X}(Y = k | X = \mathbf{x})$ is the $k$-th class-conditional distribution.

# E  Proof of Theorem 1

**Theorem 1.** *Given domain spaces $\mathscr{D}_{XY}$ and $\mathscr{D}'_{XY} = \{D^\alpha_{XY} : \forall D_{XY} \in \mathscr{D}_{XY}, \forall \alpha \in [0,1)\}$, then*
*1) $\mathscr{D}'_{XY}$ is a priori-unknown space and $\mathscr{D}_{XY} \subset \mathscr{D}'_{XY}$;*
*2) if $\mathscr{D}_{XY}$ is a priori-unknown space, then Definition 1 and Definition 2 are **equivalent**;*
*3) OOD detection is strongly learnable in $\mathscr{D}_{XY}$ **if and only if** OOD detection is learnable in $\mathscr{D}'_{XY}$.*

*Proof of Theorem 1.*

**Proof of the First Result.**

To prove that $\mathscr{D}'_{XY}$ is a priori-unknown space, we need to show that for any $D^{\alpha'}_{XY} \in \mathscr{D}'_{XY}$, then $D^\alpha_{XY} \in \mathscr{D}'_{XY}$ for any $\alpha \in [0,1)$.

According to the definition of $\mathscr{D}'_{XY}$, for any $D^{\alpha'}_{XY} \in \mathscr{D}'_{XY}$, we can find a domain $D_{XY} \in \mathscr{D}_{XY}$, which can be written as $D_{XY} = (1 - \pi^{\text{out}})D_{X_{\text{I}}Y_{\text{I}}} + \pi^{\text{out}}D_{X_{\text{O}}Y_{\text{O}}}$ (here $\pi^{\text{out}} \in [0,1)$) such that

$$D^{\alpha'}_{XY} = (1 - \alpha')D_{X_{\text{I}}Y_{\text{I}}} + \alpha'D_{X_{\text{O}}Y_{\text{O}}}.$$

Note that $D^\alpha_{XY} = (1 - \alpha)D_{X_{\text{I}}Y_{\text{I}}} + \alpha D_{X_{\text{O}}Y_{\text{O}}}$.

Therefore, based on the definition of $\mathscr{D}'_{XY}$, for any $\alpha \in [0,1)$, $D^\alpha_{XY} \in \mathscr{D}'_{XY}$, which implies that $\mathscr{D}'_{XY}$ is a prior-known space. Additionally, for any $D_{XY} \in \mathscr{D}_{XY}$, we can rewrite $D_{XY}$ as $D^{\pi^{\text{out}}}_{XY}$, thus $D_{XY} = D^{\pi^{\text{out}}}_{XY} \in \mathscr{D}'_{XY}$, which implies that $\mathscr{D}_{XY} \subset \mathscr{D}'_{XY}$.

**Proof of the Second Result.**

**First,** we prove that Definition 1 concludes Definition 2, if $\mathscr{D}_{XY}$ is a prior-unknown space:

---

The domain space $\mathscr{D}_{XY}$ is a priori-unknown space, and OOD detection is learnable in $\mathscr{D}_{XY}$ for $\mathcal{H}$.
$$\Downarrow$$
OOD detection is strongly learnable in $\mathscr{D}_{XY}$ for $\mathcal{H}$: there exist an algorithm $\mathbf{A} : \cup^{+\infty}_{n=1}(\mathcal{X} \times \mathcal{Y})^n \to \mathcal{H}$, and a monotonically decreasing sequence $\epsilon(n)$, such that $\epsilon(n) \to 0$, as $n \to +\infty$

$$\mathbb{E}_{S \sim D^n_{X_{\text{I}}Y_{\text{I}}}}\left[R^\alpha_D(\mathbf{A}(S)) - \inf_{h \in \mathcal{H}} R^\alpha_D(h)\right] \leq \epsilon(n), \quad \forall \alpha \in [0,1], \forall D_{XY} \in \mathscr{D}_{XY}.$$

---

In the priori-unknown space, for any $D_{XY} \in \mathscr{D}_{XY}$, we have that for any $\alpha \in [0,1)$,

$$D^\alpha_{XY} = (1 - \alpha)D_{X_{\text{I}}Y_{\text{I}}} + \alpha D_{X_{\text{O}}Y_{\text{O}}} \in \mathscr{D}_{XY}.$$

Then, according to the definition of learnability of OOD detection, we have an algorithm $\mathbf{A}$ and a monotonically decreasing sequence $\epsilon_{\text{cons}}(n) \to 0$, as $n \to +\infty$, such that for any $\alpha \in [0,1)$,

$$\mathbb{E}_{S \sim D^n_{X_{\text{I}}Y_{\text{I}}}} R_{D^\alpha}(\mathbf{A}(S)) \leq \inf_{h \in \mathcal{H}} R_{D^\alpha}(h) + \epsilon_{\text{cons}}(n), \quad \text{(by the property of priori-unknown space)}$$

where

$$R_{D^\alpha}(\mathbf{A}(S)) = \int_{\mathcal{X} \times \mathcal{Y}_{\text{all}}} \ell(\mathbf{A}(S)(\mathbf{x}), y)\mathrm{d}D^\alpha_{XY}(\mathbf{x}, y), \quad R_{D^\alpha}(h) = \int_{\mathcal{X} \times \mathcal{Y}_{\text{all}}} \ell(h(\mathbf{x}), y)\mathrm{d}D^\alpha_{XY}(\mathbf{x}, y).$$

Since $R_{D^\alpha}(\mathbf{A}(S)) = R^\alpha_D(\mathbf{A}(S))$ and $R_{D^\alpha}(h) = R^\alpha_D(h)$, we have that

$$\mathbb{E}_{S \sim D^n_{X_{\text{I}}Y_{\text{I}}}} R^\alpha_D(\mathbf{A}(S)) \leq \inf_{h \in \mathcal{H}} R^\alpha_D(h) + \epsilon_{\text{cons}}(n), \quad \forall \alpha \in [0,1). \tag{9}$$

Next, we consider the case that $\alpha = 1$. Note that

$$\liminf_{\alpha \to 1} \inf_{h \in \mathcal{H}} R^\alpha_D(h) \geq \liminf_{\alpha \to 1} \alpha \inf_{h \in \mathcal{H}} R^{\text{out}}_D(h) = \inf_{h \in \mathcal{H}} R^{\text{out}}_D(h). \tag{10}$$

Then, we assume that $h_\epsilon \in \mathcal{H}$ satisfies that

$$R^{\text{out}}_D(h_\epsilon) - \inf_{h \in \mathcal{H}} R^{\text{out}}_D(h) \leq \epsilon.$$

It is obvious that

$$R^\alpha_D(h_\epsilon) \geq \inf_{h \in \mathcal{H}} R^\alpha_D(h).$$

Let $\alpha \to 1$. Then, for any $\epsilon > 0$,

$$R_D^{\text{out}}(h_\epsilon) = \lim_{\alpha \to 1} R_D^\alpha(h_\epsilon) = \limsup_{\alpha \to 1} R_D^\alpha(h_\epsilon) \geq \limsup_{\alpha \to 1} \inf_{h \in \mathcal{H}} R_D^\alpha(h),$$

which implies that

$$\inf_{h \in \mathcal{H}} R_D^{\text{out}}(h) = \lim_{\epsilon \to 0} R_D^{\text{out}}(h_\epsilon) \geq \lim_{\epsilon \to 0} \limsup_{\alpha \to 1} \inf_{h \in \mathcal{H}} R_D^\alpha(h) = \limsup_{\alpha \to 1} \inf_{h \in \mathcal{H}} R_D^\alpha(h). \tag{11}$$

Combining Eq. (10) with Eq. (11), we have

$$\inf_{h \in \mathcal{H}} R_D^{\text{out}}(h) = \limsup_{\alpha \to 1} \inf_{h \in \mathcal{H}} R_D^\alpha(h) = \liminf_{\alpha \to 1} \inf_{h \in \mathcal{H}} R_D^\alpha(h), \tag{12}$$

which implies that

$$\inf_{h \in \mathcal{H}} R_D^{\text{out}}(h) = \lim_{\alpha \to 1} \inf_{h \in \mathcal{H}} R_D^\alpha(h). \tag{13}$$

Note that

$$\mathbb{E}_{S \sim D_{X_I Y_I}^n} R_D^\alpha(\mathbf{A}(S)) = (1 - \alpha) \mathbb{E}_{S \sim D_{X_I Y_I}^n} R_D^{\text{in}}(\mathbf{A}(S)) + \alpha \mathbb{E}_{S \sim D_{X_I Y_I}^n} R_D^{\text{out}}(\mathbf{A}(S)).$$

Hence, Lebesgue's Dominated Convergence Theorem [36] implies that

$$\lim_{\alpha \to 1} \mathbb{E}_{S \sim D_{X_I Y_I}^n} R_D^\alpha(\mathbf{A}(S)) = \mathbb{E}_{S \sim D_{X_I Y_I}^n} R_D^{\text{out}}(\mathbf{A}(S)). \tag{14}$$

Using Eq. (9), we have that

$$\lim_{\alpha \to 1} \mathbb{E}_{S \sim D_{X_I Y_I}^n} R_D^\alpha(\mathbf{A}(S)) \leq \lim_{\alpha \to 1} \inf_{h \in \mathcal{H}} R_D^\alpha(h) + \epsilon_{\text{cons}}(n). \tag{15}$$

Combining Eq. (13), Eq. (14) with Eq. (15), we obtain that

$$\mathbb{E}_{S \sim D_{X_I Y_I}^n} R_D^{\text{out}}(\mathbf{A}(S)) \leq \inf_{h \in \mathcal{H}} R_D^{\text{out}}(h) + \epsilon_{\text{cons}}(n).$$

Since $R_D^{\text{out}}(\mathbf{A}(S)) = R_D^1(\mathbf{A}(S))$ and $R_D^{\text{out}}(h) = R_D^1(h)$, we obtain that

$$\mathbb{E}_{S \sim D_{X_I Y_I}^n} R_D^1(\mathbf{A}(S)) \leq \inf_{h \in \mathcal{H}} R_D^1(h) + \epsilon_{\text{cons}}(n). \tag{16}$$

Combining Eq. (9) and Eq. (16), we have proven that: if the domain space $\mathscr{D}_{XY}$ is a priori-unknown space, then OOD detection is learnable in $\mathscr{D}_{XY}$ for $\mathcal{H}$.

$$\Downarrow$$

OOD detection is strongly learnable in $\mathscr{D}_{XY}$ for $\mathcal{H}$: there exist an algorithm $\mathbf{A} : \cup_{n=1}^{+\infty}(\mathcal{X} \times \mathcal{Y})^n \to \mathcal{H}$, and a monotonically decreasing sequence $\epsilon(n)$, such that $\epsilon(n) \to 0$, as $n \to +\infty$,

$$\mathbb{E}_{S \sim D_{X_I Y_I}^n} R_D^\alpha(\mathbf{A}(S)) \leq \inf_{h \in \mathcal{H}} R_D^\alpha(h) + \epsilon(n), \quad \forall \alpha \in [0, 1], \ \forall D_{XY} \in \mathscr{D}_{XY}.$$

**Second**, we prove that Definition 2 concludes Definition 1:

---
OOD detection is strongly learnable in $\mathscr{D}_{XY}$ for $\mathcal{H}$: there exist an algorithm $\mathbf{A} : \cup_{n=1}^{+\infty}(\mathcal{X} \times \mathcal{Y})^n \to \mathcal{H}$, and a monotonically decreasing sequence $\epsilon(n)$, such that $\epsilon(n) \to 0$, as $n \to +\infty$,

$$\mathbb{E}_{S \sim D_{X_I Y_I}^n} \left[ R_D^\alpha(\mathbf{A}(S)) - \inf_{h \in \mathcal{H}} R_D^\alpha(h) \right] \leq \epsilon(n), \quad \forall \alpha \in [0, 1], \ \forall D_{XY} \in \mathscr{D}_{XY}.$$

$$\Downarrow$$

OOD detection is learnable in $\mathscr{D}_{XY}$ for $\mathcal{H}$.

---

If we set $\alpha = \pi^{\text{out}}$, then $\mathbb{E}_{S \sim D_{X_I Y_I}^n} R_D^\alpha(\mathbf{A}(S)) \leq \inf_{h \in \mathcal{H}} R_D^\alpha(h) + \epsilon(n)$ implies that

$$\mathbb{E}_{S \sim D_{X_I Y_I}^n} R_D(\mathbf{A}(S)) \leq \inf_{h \in \mathcal{H}} R_D(h) + \epsilon(n),$$

which means that OOD detection is learnable in $\mathscr{D}_{XY}$ for $\mathcal{H}$. We have completed this proof.

**Proof of the Third Result.**

The third result is a simple conclusion of the second result. Hence, we omit it. $\qquad\square$

# F  Proof of Theorem 2

Before introducing the proof of Theorem 2, we extend Condition 1 to a general version (Condition 4). Then, Lemma 1 proves that Conditions 1 and 4 are the necessary conditions for the learnability of OOD detection. First, we provide the details of Condition 4.

Let $\Delta_l^{\circ} = \{(\lambda_1, ..., \lambda_l) : \sum_{j=1}^{l} \lambda_j < 1 \text{ and } \lambda_j \geq 0, \forall j = 1, ..., l\}$, where $l$ is a positive integer. Next, we introduce an important definition as follows:

**Definition 6** (OOD Convex Decomposition and Convex Domain). *Given any domain $D_{XY} \in \mathscr{D}_{XY}$, we say joint distributions $Q_1, ..., Q_l$, which are defined over $\mathcal{X} \times \{K + 1\}$, are the OOD convex decomposition for $D_{XY}$, if*

$$D_{XY} = (1 - \sum_{j=1}^{l} \lambda_j) D_{X_{\mathrm{I}}Y_{\mathrm{I}}} + \sum_{j=1}^{l} \lambda_j Q_j,$$

*for some $(\lambda_1, ..., \lambda_l) \in \Delta_l^{\circ}$. We also say domain $D_{XY} \in \mathscr{D}_{XY}$ is an OOD convex domain corresponding to OOD convex decomposition $Q_1, ..., Q_l$, if for any $(\alpha_1, ..., \alpha_l) \in \Delta_l^{\circ}$,*

$$(1 - \sum_{j=1}^{l} \alpha_j) D_{X_{\mathrm{I}}Y_{\mathrm{I}}} + \sum_{j=1}^{l} \alpha_j Q_j \in \mathscr{D}_{XY}.$$

We extend the linear condition (Condition 1) to a multi-linear scenario.

**Condition 4** (Multi-linear Condition). *For each OOD convex domain $D_{XY} \in \mathscr{D}_{XY}$ corresponding to OOD convex decomposition $Q_1, ..., Q_l$, the following function*

$$f_{D,Q}(\alpha_1, ..., \alpha_l) := \inf_{h \in \mathcal{H}} \left( (1 - \sum_{j=1}^{l} \alpha_j) R_D^{\mathrm{in}}(h) + \sum_{j=1}^{l} \alpha_j R_{Q_j}(h) \right), \quad \forall (\alpha_1, ..., \alpha_l) \in \Delta_l^{\circ}$$

*satisfies that*

$$f_{D,Q}(\alpha_1, ..., \alpha_l) = (1 - \sum_{j=1}^{l} \alpha_j) f_{D,Q}(\mathbf{0}) + \sum_{j=1}^{l} \alpha_j f_{D,Q}(\boldsymbol{\alpha}_j),$$

*where $\mathbf{0}$ is the $1 \times l$ vector, whose elements are 0, and $\boldsymbol{\alpha}_j$ is the $1 \times l$ vector, whose $j$-th element is 1 and other elements are 0.*

When $l = 1$ and the domain space $\mathscr{D}_{XY}$ is a priori-unknown space, Condition 4 degenerates into Condition 1. Lemma 1 shows that Condition 4 is necessary for the learnability of OOD detection.

**Lemma 1.** *Given a priori-unknown space $\mathscr{D}_{XY}$ and a hypothesis space $\mathcal{H}$, if OOD detection is learnable in $\mathscr{D}_{XY}$ for $\mathcal{H}$, then Conditions 1 and 4 hold.*

*Proof of Lemma 1.*

Since Condition 1 is a special case of Condition 4, we only need to prove that Condition 4 holds.

For any OOD convex domain $D_{XY} \in \mathscr{D}_{XY}$ corresponding to OOD convex decomposition $Q_1, ..., Q_l$, and any $(\alpha_1, ..., \alpha_l) \in \Delta_l^{\circ}$, we set

$$Q^{\boldsymbol{\alpha}} = \frac{1}{\sum_{i=1}^{l} \alpha_i} \sum_{j=1}^{l} \alpha_j Q_j.$$

Then, we define

$$D_{XY}^{\boldsymbol{\alpha}} = (1 - \sum_{i=1}^{l} \alpha_i) D_{X_{\mathrm{I}}Y_{\mathrm{I}}} + (\sum_{i=1}^{l} \alpha_i) Q^{\boldsymbol{\alpha}}, \text{ which belongs to } \mathscr{D}_{XY}.$$

Let

$$R_D^{\boldsymbol{\alpha}}(h) = \int_{\mathcal{X} \times \mathcal{Y}_{\mathrm{all}}} \ell(h(\mathbf{x}), y) \mathrm{d} D_{XY}^{\boldsymbol{\alpha}}(\mathbf{x}, y).$$

Since OOD detection is learnable in $\mathscr{D}_{XY}$ for $\mathcal{H}$, there exist an algorithm $\mathbf{A} : \cup_{n=1}^{+\infty}(\mathcal{X} \times \mathcal{Y})^n \to \mathcal{H}$, and a monotonically decreasing sequence $\epsilon(n)$, such that $\epsilon(n) \to 0$, as $n \to +\infty$, and

$$0 \leq \mathbb{E}_{S \sim D_{X_I Y_I}^n} R_D^{\boldsymbol{\alpha}}(\mathbf{A}(S)) - \inf_{h \in \mathcal{H}} R_D^{\boldsymbol{\alpha}}(h) \leq \epsilon(n).$$

Note that

$$\mathbb{E}_{S \sim D_{X_I Y_I}^n} R_D^{\boldsymbol{\alpha}}(\mathbf{A}(S)) = (1 - \sum_{j=1}^{l} \alpha_j)\mathbb{E}_{S \sim D_{X_I Y_I}^n} R_D^{\mathrm{in}}(\mathbf{A}(S)) + \sum_{j=1}^{l} \alpha_j \mathbb{E}_{S \sim D_{X_I Y_I}^n} R_{Q_j}(\mathbf{A}(S)),$$

and

$$\inf_{h \in \mathcal{H}} R_D^{\boldsymbol{\alpha}}(h) = f_{D,Q}(\alpha_1, ..., \alpha_l),$$

where

$$R_{Q_j}(\mathbf{A}(S)) = \int_{\mathcal{X} \times \{K+1\}} \ell(\mathbf{A}(S)(\mathbf{x}), y) \mathrm{d}Q_j(\mathbf{x}, y).$$

Therefore, we have that for any $(\alpha_1, ..., \alpha_l) \in \Delta_l^{\mathrm{o}}$,

$$\left|(1 - \sum_{j=1}^{l} \alpha_j)\mathbb{E}_{S \sim D_{X_I Y_I}^n} R_D^{\mathrm{in}}(\mathbf{A}(S)) + \sum_{j=1}^{l} \alpha_j \mathbb{E}_{S \sim D_{X_I Y_I}^n} R_{Q_j}(\mathbf{A}(S)) - f_{D,Q}(\alpha_1, ..., \alpha_l)\right| \leq \epsilon(n).$$

$$(17)$$

Let

$$g_n(\alpha_1, ..., \alpha_l) = (1 - \sum_{j=1}^{l} \alpha_j)\mathbb{E}_{S \sim D_{X_I Y_I}^n} R_D^{\mathrm{in}}(\mathbf{A}(S)) + \sum_{j=1}^{l} \alpha_j \mathbb{E}_{S \sim D_{X_I Y_I}^n} R_{Q_j}(\mathbf{A}(S)).$$

Note that Eq. (17) implies that

$$\lim_{n \to +\infty} g_n(\alpha_1, ..., \alpha_l) = f_{D,Q}(\alpha_1, ..., \alpha_l), \quad \forall(\alpha_1, ..., \alpha_l) \in \Delta_l^{\mathrm{o}},$$

$$\lim_{n \to +\infty} g_n(\mathbf{0}) = f_{D,Q}(\mathbf{0}).$$

$$(18)$$

**Step 1.** Since $\boldsymbol{\alpha}_j \notin \Delta_l^{\mathrm{o}}$, we need to prove that

$$\lim_{n \to +\infty} \mathbb{E}_{S \sim D_{X_I Y_I}^n} R_{Q_j}(\mathbf{A}(S)) = f(\boldsymbol{\alpha}_j), i.e., \lim_{n \to +\infty} g_n(\boldsymbol{\alpha}_j) = f(\boldsymbol{\alpha}_j), \tag{19}$$

where $\boldsymbol{\alpha}_j$ is the $1 \times l$ vector, whose $j$-th element is 1 and other elements are 0.

Let $\tilde{D}_{XY} = 0.5 * D_{X_I Y_I} + 0.5 * Q_j$. The second result of Theorem 1 implies that

$$\mathbb{E}_{S \sim D_{X_I Y_I}^n} R_{\tilde{D}}^{\mathrm{out}}(\mathbf{A}(S)) \leq \inf_{h \in \mathcal{H}} R_{\tilde{D}}^{\mathrm{out}}(h) + \epsilon(n).$$

Since $R_{\tilde{D}}^{\mathrm{out}}(\mathbf{A}(S)) = R_{Q_j}(\mathbf{A}(S))$ and $R_{\tilde{D}}^{\mathrm{out}}(h) = R_{Q_j}(h)$,

$$\mathbb{E}_{S \sim D_{X_I Y_I}^n} R_{Q_j}(\mathbf{A}(S)) \leq \inf_{h \in \mathcal{H}} R_{Q_j}(h) + \epsilon(n).$$

Note that $\inf_{h \in \mathcal{H}} R_{Q_j}(h) \leq \mathbb{E}_{S \sim D_{X_I Y_I}^n} R_{Q_j}(\mathbf{A}(S))$. We have

$$0 \leq \mathbb{E}_{S \sim D_{X_I Y_I}^n} R_{Q_j}(\mathbf{A}(S)) - \inf_{h \in \mathcal{H}} R_{Q_j}(h) \leq \epsilon(n). \tag{20}$$

Eq. (20) implies that

$$\lim_{n \to +\infty} \mathbb{E}_{S \sim D_{X_I Y_I}^n} R_{Q_j}(\mathbf{A}(S)) = \inf_{h \in \mathcal{H}} R_{Q_j}(h). \tag{21}$$

We note that $\inf_{h \in \mathcal{H}} R_{Q_j}(h) = f_{D,Q}(\boldsymbol{\alpha}_j)$. Therefore,

$$\lim_{n \to +\infty} \mathbb{E}_{S \sim D_{X_I Y_I}^n} R_{Q_j}(\mathbf{A}(S)) = f_{D,Q}(\boldsymbol{\alpha}_j), i.e., \lim_{n \to +\infty} g_n(\boldsymbol{\alpha}_j) = f(\boldsymbol{\alpha}_j). \tag{22}$$

**Step 2.** It is easy to check that for any $(\alpha_1, ..., \alpha_l) \in \Delta_l^o$,

$$
\begin{aligned}
\lim_{n \to +\infty} g_n(\alpha_1, ..., \alpha_l) &= \lim_{n \to +\infty} \left( (1 - \sum_{j=1}^l \alpha_j) g_n(\mathbf{0}) + \sum_{j=1}^l \alpha_j g_n(\boldsymbol{\alpha}_j) \right) \\
&= (1 - \sum_{j=1}^l \alpha_j) \lim_{n \to +\infty} g_n(\mathbf{0}) + \sum_{j=1}^l \alpha_j \lim_{n \to +\infty} g_n(\boldsymbol{\alpha}_j).
\end{aligned}
\tag{23}
$$

According to Eq. (18) and Eq. (22), we have

$$
\begin{aligned}
\lim_{n \to +\infty} g_n(\alpha_1, ..., \alpha_l) &= f_{D,Q}(\alpha_1, ..., \alpha_l), \quad \forall (\alpha_1, ..., \alpha_l) \in \Delta_l^o, \\
\lim_{n \to +\infty} g_n(\mathbf{0}) &= f_{D,Q}(\mathbf{0}), \\
\lim_{n \to +\infty} g_n(\boldsymbol{\alpha}_j) &= f(\boldsymbol{\alpha}_j),
\end{aligned}
\tag{24}
$$

Combining Eq. (24) with Eq. (23), we complete the proof. $\qquad\square$

**Lemma 2.**
$$
\inf_{h \in \mathcal{H}} R_D^\alpha(h) = (1 - \alpha) \inf_{h \in \mathcal{H}} R_D^{\text{in}}(h) + \alpha \inf_{h \in \mathcal{H}} R_D^{\text{out}}(h), \forall \alpha \in [0, 1),
$$
*if and only if* for any $\epsilon > 0$,
$$
\{h' \in \mathcal{H} : R_D^{\text{in}}(h') \le \inf_{h \in \mathcal{H}} R_D^{\text{in}}(h) + 2\epsilon\} \cap \{h' \in \mathcal{H} : R_D^{\text{out}}(h') \le \inf_{h \in \mathcal{H}} R_D^{\text{out}}(h) + 2\epsilon\} \ne \emptyset.
$$

*Proof of Lemma 2.* For the sake of convenience, we set $f_D(\alpha) = \inf_{h \in \mathcal{H}} R_D^\alpha(h)$, for any $\alpha \in [0, 1]$.
**First**, we prove that $f_D(\alpha) = (1 - \alpha) f_D(0) + \alpha f_D(1), \ \forall \alpha \in [0, 1)$ implies
$$
\{h' \in \mathcal{H} : R_D^{\text{in}}(h') \le \inf_{h \in \mathcal{H}} R_D^{\text{in}}(h) + 2\epsilon\} \cap \{h' \in \mathcal{H} : R_D^{\text{out}}(h') \le \inf_{h \in \mathcal{H}} R_D^{\text{out}}(h) + 2\epsilon\} \ne \emptyset.
$$

For any $\epsilon > 0$ and $0 \le \alpha < 1$, we can find $h_\epsilon^\alpha \in \mathcal{H}$ satisfying that
$$
R_D^\alpha(h_\epsilon^\alpha) \le \inf_{h \in \mathcal{H}} R_D^\alpha(h) + \epsilon.
$$

Note that
$$
\inf_{h \in \mathcal{H}} R_D^\alpha(h) = \inf_{h \in \mathcal{H}} \left( (1 - \alpha) R_D^{\text{in}}(h) + \alpha R_D^{\text{out}}(h) \right) \ge (1 - \alpha) \inf_{h \in \mathcal{H}} R_D^{\text{in}}(h) + \alpha \inf_{h \in \mathcal{H}} R_D^{\text{out}}(h).
$$

Therefore,
$$
(1 - \alpha) \inf_{h \in \mathcal{H}} R_D^{\text{in}}(h) + \alpha \inf_{h \in \mathcal{H}} R_D^{\text{out}}(h) \le \inf_{h \in \mathcal{H}} R_D^\alpha(h) \le R_D^\alpha(h_\epsilon^\alpha) \le \inf_{h \in \mathcal{H}} R_D^\alpha(h) + \epsilon.
\tag{25}
$$

Note that $f_D(\alpha) = (1 - \alpha) f_D(0) + \alpha f_D(1), \forall \alpha \in [0, 1)$, *i.e.*,
$$
\inf_{h \in \mathcal{H}} R_D^\alpha(h) = (1 - \alpha) \inf_{h \in \mathcal{H}} R_D^{\text{in}}(h) + \alpha \inf_{h \in \mathcal{H}} R_D^{\text{out}}(h), \forall \alpha \in [0, 1).
\tag{26}
$$

Using Eqs. (25) and (26), we have that for any $0 \le \alpha < 1$,
$$
\epsilon \ge \left| R_D^\alpha(h_\epsilon^\alpha) - \inf_{h \in \mathcal{H}} R_D^\alpha(h) \right| = \left| (1 - \alpha) \left( R_D^{\text{in}}(h_\epsilon^\alpha) - \inf_{h \in \mathcal{H}} R_D^{\text{in}}(h) \right) + \alpha \left( R_D^{\text{out}}(h_\epsilon^\alpha) - \inf_{h \in \mathcal{H}} R_D^{\text{out}}(h) \right) \right|.
\tag{27}
$$

Since $R_D^{\text{out}}(h_\epsilon^\alpha) - \inf_{h \in \mathcal{H}} R_D^{\text{out}}(h) \ge 0$ and $R_D^{\text{in}}(h_\epsilon^\alpha) - \inf_{h \in \mathcal{H}} R_D^{\text{in}}(h) \ge 0$, Eq. (27) implies that: for any $0 < \alpha < 1$,
$$
\begin{aligned}
R_D^{\text{in}}(h_\epsilon^\alpha) &\le \inf_{h \in \mathcal{H}} R_D^{\text{in}}(h) + \epsilon/(1 - \alpha), \\
R_D^{\text{out}}(h_\epsilon^\alpha) &\le \inf_{h \in \mathcal{H}} R_D^{\text{out}}(h) + \epsilon/\alpha.
\end{aligned}
$$

Therefore,
$$
h_\epsilon^\alpha \in \{h' \in \mathcal{H} : R_D^{\text{in}}(h') \le \inf_{h \in \mathcal{H}} R_D^{\text{in}}(h) + \epsilon/(1 - \alpha)\} \cap \{h' \in \mathcal{H} : R_D^{\text{out}}(h') \le \inf_{h \in \mathcal{H}} R_D^{\text{out}}(h) + \epsilon/\alpha\}.
$$

If we set $\alpha = 0.5$, we obtain that for any $\epsilon > 0$,

$$\{h' \in \mathcal{H} : R_D^{\text{in}}(h') \leq \inf_{h \in \mathcal{H}} R_D^{\text{in}}(h) + 2\epsilon\} \cap \{h' \in \mathcal{H} : R_D^{\text{out}}(h') \leq \inf_{h \in \mathcal{H}} R_D^{\text{out}}(h) + 2\epsilon\} \neq \emptyset.$$

**Second**, we prove that for any $\epsilon > 0$, if

$$\{h' \in \mathcal{H} : R_D^{\text{in}}(h') \leq \inf_{h \in \mathcal{H}} R_D^{\text{in}}(h) + 2\epsilon\} \cap \{h' \in \mathcal{H} : R_D^{\text{out}}(h') \leq \inf_{h \in \mathcal{H}} R_D^{\text{out}}(h) + 2\epsilon\} \neq \emptyset,$$

then $f_D(\alpha) = (1 - \alpha)f_D(0) + \alpha f_D(1)$, for any $\alpha \in [0, 1)$.

Let $h_\epsilon \in \{h' \in \mathcal{H} : R_D^{\text{in}}(h') \leq \inf_{h \in \mathcal{H}} R_D^{\text{in}}(h) + 2\epsilon\} \cap \{h' \in \mathcal{H} : R_D^{\text{out}}(h') \leq \inf_{h \in \mathcal{H}} R_D^{\text{out}}(h) + 2\epsilon\}$. Then,

$$\inf_{h \in \mathcal{H}} R_D^\alpha(h) \leq R_D^\alpha(h_\epsilon) \leq (1 - \alpha) \inf_{h \in \mathcal{H}} R_D^{\text{in}}(h) + \alpha \inf_{h \in \mathcal{H}} R_D^{\text{out}}(h) + 2\epsilon \leq \inf_{h \in \mathcal{H}} R_D^\alpha(h) + 2\epsilon,$$

which implies that $|f_D(\alpha) - (1 - \alpha)f_D(0) - \alpha f_D(1)| \leq 2\epsilon$.

As $\epsilon \to 0$, $|f_D(\alpha) - (1 - \alpha)f_D(0) - \alpha f_D(1)| \leq 0$. We have completed the proof. $\qquad \square$

**Theorem 2.** *Given a hypothesis space $\mathcal{H}$ and a domain $D_{XY}$, OOD detection is learnable in the single-distribution space $\mathscr{D}_{XY}^{D_{XY}}$ for $\mathcal{H}$ **if and only if** linear condition (i.e., Condition 1) holds.*

*Proof of Theorem 2.* Based on Lemma 1, we obtain that Condition 1 is the necessary condition for the learnability of OOD detection in the single-distribution space $\mathscr{D}_{XY}^{D_{XY}}$. Next, it suffices to prove that Condition 1 is the sufficient condition for the learnability of OOD detection in the single-distribution space $\mathscr{D}_{XY}^{D_{XY}}$. We use Lemma 2 to prove the sufficient condition.

Let $\mathscr{F}$ be the infinite sequence set that consists of all infinite sequences, whose coordinates are hypothesis functions, *i.e.*,

$$\mathscr{F} = \{\boldsymbol{h} = (h_1, ..., h_n, ...) : \forall h_n \in \mathcal{H}, n = 1, ...., +\infty\}.$$

For each $\boldsymbol{h} \in \mathscr{F}$, there is a corresponding algorithm $\mathbf{A}_{\boldsymbol{h}}$[6]: $\mathbf{A}_{\boldsymbol{h}}(S) = h_n$, if $|S| = n$. $\mathscr{F}$ generates an algorithm class $\mathscr{A} = \{\mathbf{A}_{\boldsymbol{h}} : \forall \boldsymbol{h} \in \mathscr{F}\}$. We select a consistent algorithm from the algorithm class $\mathscr{A}$.

We construct a special infinite sequence $\tilde{\boldsymbol{h}} = (\tilde{h}_1, ..., \tilde{h}_n, ...) \in \mathscr{F}$. For each positive integer $n$, we select $\tilde{h}_n$ from $\{h' \in \mathcal{H} : R_D^{\text{in}}(h') \leq \inf_{h \in \mathcal{H}} R_D^{\text{in}}(h) + 2/n\} \cap \{h' \in \mathcal{H} : R_D^{\text{out}}(h') \leq \inf_{h \in \mathcal{H}} R_D^{\text{out}}(h) + 2/n\}$ (the existence of $\tilde{h}_n$ is based on Lemma 2). It is easy to check that

$$\mathbb{E}_{S \sim D_{X_I Y_I}^n} R_D^{\text{in}}(\mathbf{A}_{\tilde{\boldsymbol{h}}}(S)) \leq \inf_{h \in \mathcal{H}} R_D^{\text{in}}(h) + 2/n.$$

$$\mathbb{E}_{S \sim D_{X_I Y_I}^n} R_D^{\text{out}}(\mathbf{A}_{\tilde{\boldsymbol{h}}}(S)) \leq \inf_{h \in \mathcal{H}} R_D^{\text{out}}(h) + 2/n.$$

Since $(1-\alpha) \inf_{h \in \mathcal{H}} R_D^{\text{in}}(h) + \alpha \inf_{h \in \mathcal{H}} R_D^{\text{out}}(h) \leq \inf_{h \in \mathcal{H}} R_D^\alpha(h)$, we obtain that for any $\alpha \in [0, 1]$,

$$\mathbb{E}_{S \sim D_{X_I Y_I}^n} R_D^\alpha(\mathbf{A}_{\tilde{\boldsymbol{h}}}(S)) \leq \inf_{h \in \mathcal{H}} R_D^\alpha(h) + 2/n.$$

We have completed this proof. $\qquad \square$

# G  Proofs of Theorem 3 and Theorem 4

## G.1  Proof of Theorem 3

**Theorem 3.** *Given a hypothesis space $\mathcal{H}$ and a prior-unknown space $\mathscr{D}_{XY}$, if there is $D_{XY} \in \mathscr{D}_{XY}$, which has overlap between ID and OOD, and $\inf_{h \in \mathcal{H}} R_D^{\text{in}}(h) = 0$ and $\inf_{h \in \mathcal{H}} R_D^{\text{out}}(h) = 0$, then Condition 1 does not hold. Therefore, OOD detection is not learnable in $\mathscr{D}_{XY}$ for $\mathcal{H}$.*

---

[6]In this paper, we regard an algorithm as a mapping from $\cup_{n=1}^{+\infty}(\mathcal{X} \times \mathcal{Y})^n$ to $\mathcal{H}$. So we can design an algorithm like this.

*Proof of Theorem 3.* We **first** explain how we get $f_\mathrm{I}$ and $f_\mathrm{O}$ in Definition 4. Since $D_X$ is absolutely continuous respect to $\mu$ ($D_X \ll \mu$), then $D_{X_\mathrm{I}} \ll \mu$ and $D_{X_\mathrm{O}} \ll \mu$. By Radon-Nikodym Theorem [36], we know there exist two non-negative functions defined over $\mathcal{X}$: $f_\mathrm{I}$ and $f_\mathrm{O}$ such that for any $\mu$-measurable set $A \subset \mathcal{X}$,

$$D_{X_\mathrm{I}}(A) = \int_A f_\mathrm{I}(\mathbf{x})\mathrm{d}\mu(\mathbf{x}), \ \ D_{X_\mathrm{O}}(A) = \int_A f_\mathrm{O}(\mathbf{x})\mathrm{d}\mu(\mathbf{x}).$$

**Second**, we prove that for any $\alpha \in (0,1)$, $\inf_{h\in\mathcal{H}} R_D^\alpha(h) > 0$.

We define $A_m = \{\mathbf{x} \in \mathcal{X} : f_\mathrm{I}(\mathbf{x}) \geq \frac{1}{m} \text{ and } f_\mathrm{O}(\mathbf{x}) \geq \frac{1}{m}\}$. It is clear that

$$\cup_{m=1}^{+\infty} A_m = \{\mathbf{x} \in \mathcal{X} : f_\mathrm{I}(\mathbf{x}) > 0 \text{ and } f_\mathrm{O}(\mathbf{x}) > 0\} = A_\mathrm{overlap},$$

and

$$A_m \subset A_{m+1}.$$

Therefore,

$$\lim_{m\to+\infty} \mu(A_m) = \mu(A_\mathrm{overlap}) > 0,$$

which implies that there exists $m_0$ such that

$$\mu(A_{m_0}) > 0.$$

For any $\alpha \in (0,1)$, we define $c_\alpha = \min_{y_1\in\mathcal{Y}_\mathrm{all}} \big((1-\alpha)\min_{y_2\in\mathcal{Y}} \ell(y_1, y_2) + \alpha\ell(y_1, K+1)\big)$. It is clear that $c_\alpha > 0$ for $\alpha \in (0,1)$. Then, for any $h \in \mathcal{H}$,

$$
\begin{aligned}
&R_D^\alpha(h) \\
&= \int_{\mathcal{X}\times\mathcal{Y}_\mathrm{all}} \ell(h(\mathbf{x}), y)\mathrm{d}D_{XY}^\alpha(\mathbf{x}, y) \\
&= \int_{\mathcal{X}\times\mathcal{Y}} (1-\alpha)\ell(h(\mathbf{x}), y)\mathrm{d}D_{X_\mathrm{I}Y_\mathrm{I}}(\mathbf{x}, y) + \int_{\mathcal{X}\times\{K+1\}} \alpha\ell(h(\mathbf{x}), y)\mathrm{d}D_{X_\mathrm{O}Y_\mathrm{O}}(\mathbf{x}, y) \\
&\geq \int_{A_{m_0}\times\mathcal{Y}} (1-\alpha)\ell(h(\mathbf{x}), y)\mathrm{d}D_{X_\mathrm{I}Y_\mathrm{I}}(\mathbf{x}, y) + \int_{A_{m_0}\times\{K+1\}} \alpha\ell(h(\mathbf{x}), y)\mathrm{d}D_{X_\mathrm{O}Y_\mathrm{O}}(\mathbf{x}, y) \\
&= \int_{A_{m_0}} \Big((1-\alpha)\int_{\mathcal{Y}} \ell(h(\mathbf{x}), y)\mathrm{d}D_{Y_\mathrm{I}|X_\mathrm{I}}(y|\mathbf{x})\Big)\mathrm{d}D_{X_\mathrm{I}}(\mathbf{x}) \\
&\quad + \int_{A_{m_0}} \alpha\ell(h(\mathbf{x}), K+1)\mathrm{d}D_{X_\mathrm{O}}(\mathbf{x}) \\
&\geq \int_{A_{m_0}} (1-\alpha)\min_{y_2\in\mathcal{Y}} \ell(h(\mathbf{x}), y_2)\mathrm{d}D_{X_\mathrm{I}}(\mathbf{x}) + \int_{A_{m_0}} \alpha\ell(h(\mathbf{x}), K+1)\mathrm{d}D_{X_\mathrm{O}}(\mathbf{x}) \\
&\geq \int_{A_{m_0}} (1-\alpha)\min_{y_2\in\mathcal{Y}} \ell(h(\mathbf{x}), y_2)f_\mathrm{I}(\mathbf{x})\mathrm{d}\mu(\mathbf{x}) + \int_{A_{m_0}} \alpha\ell(h(\mathbf{x}), K+1)f_\mathrm{O}(\mathbf{x})\mathrm{d}\mu(\mathbf{x}) \\
&\geq \frac{1}{m_0}\int_{A_{m_0}} (1-\alpha)\min_{y_2\in\mathcal{Y}} \ell(h(\mathbf{x}), y_2)\mathrm{d}\mu(\mathbf{x}) + \frac{1}{m_0}\int_{A_{m_0}} \alpha\ell(h(\mathbf{x}), K+1)\mathrm{d}\mu(\mathbf{x}) \\
&= \frac{1}{m_0}\int_{A_{m_0}} \big((1-\alpha)\min_{y_2\in\mathcal{Y}} \ell(h(\mathbf{x}), y_2) + \alpha\ell(h(\mathbf{x}), K+1)\big)\mathrm{d}\mu(\mathbf{x}) \geq \frac{c_\alpha}{m_0}\mu(A_{m_0}) > 0.
\end{aligned}
$$

Therefore,

$$\inf_{h\in\mathcal{H}} R_D^\alpha(h) \geq \frac{c_\alpha}{m_0}\mu(A_{m_0}) > 0.$$

**Third**, Condition 1 indicates that $\inf_{h\in\mathcal{H}} R_D^\alpha(h) = (1-\alpha)\inf_{h\in\mathcal{H}} R_D^\mathrm{in}(h) + \alpha\inf_{h\in\mathcal{H}} R_D^\mathrm{in}(h) = 0$ (here we have used conditions $\inf_{h\in\mathcal{H}} R_D^\mathrm{in}(h) = 0$ and $\inf_{h\in\mathcal{H}} R_D^\mathrm{out}(h) = 0$), which contradicts with $\inf_{h\in\mathcal{H}} R_D^\alpha(h) > 0$ ($\alpha \in (0,1)$). Therefore, Condition 1 does not hold. Using Lemma 1, we obtain that OOD detection in $\mathscr{D}_{XY}$ is not learnable for $\mathcal{H}$. $\qquad\square$

### G.2 Proof of Theorem 4

**Theorem 4** (Impossibility Theorem for Total Space). *OOD detection is not learnable in the total space $\mathscr{D}_{XY}^{\mathrm{all}}$ for $\mathcal{H}$, if $|\phi \circ \mathcal{H}| > 1$, where $\phi$ maps ID labels to $1$ and maps OOD labels to $2$.*

*Proof of Theorem 4.* We need to prove that OOD detection is not learnable in the total space $\mathscr{D}_{XY}^{\mathrm{all}}$ for $\mathcal{H}$, if $\mathcal{H}$ is non-trivial, *i.e.*, $\{\mathbf{x} \in \mathcal{X} : \exists h_1, h_2 \in \mathcal{H}, \text{s.t. } h_1(\mathbf{x}) \in \mathcal{Y}, h_2(\mathbf{x}) = K+1\} \neq \emptyset$.

The main idea is to construct a domain $D_{XY}$ satisfying that:
1) the ID and OOD distributions have overlap (Definition 4); and 2) $R_D^{\mathrm{in}}(h_1) = 0$, $R_D^{\mathrm{out}}(h_2) = 0$.

According to the condition that $\mathcal{H}$ is non-trivial, we know that there exist $h_1, h_2 \in \mathcal{H}$ such that $h_1(\mathbf{x}_1) \in \mathcal{Y}, h_2(\mathbf{x}_1) = K+1$, for some $\mathbf{x}_1 \in \mathcal{X}$. We set $D_{XY} = 0.5 * \delta_{(\mathbf{x}_1, h_1(\mathbf{x}_1))} + 0.5 * \delta_{(\mathbf{x}_1, h_2(\mathbf{x}_1))}$, where $\delta$ is the Dirac measure. It is easy to check that $R_D^{\mathrm{in}}(h_1) = 0$, $R_D^{\mathrm{out}}(h_2) = 0$, which implies that $\inf_{h \in \mathcal{H}} R_D^{\mathrm{in}}(h) = 0$ and $\inf_{h \in \mathcal{H}} R_D^{\mathrm{out}}(h) = 0$. In addition, the ID distribution $\delta_{(\mathbf{x}_1, h_1(\mathbf{x}_1))}$ and OOD distribution $\delta_{(\mathbf{x}_1, h_2(\mathbf{x}_1))}$ have overlap $\mathbf{x}_1$. By using Theorem 3, we have completed this proof. $\qquad\square$

## H   Proof of Theorem 5

Before proving Theorem 5, we need three important lemmas.

**Lemma 3.** *Suppose that $D_{XY}$ is a domain with OOD convex decomposition $Q_1, ..., Q_l$ (convex decomposition is given by Definition 6 in Appendix F), and $D_{XY}$ is a finite discrete distribution, then (the definition of $f_{D,Q}$ is given in Condition 4)*

$$f_{D,Q}(\alpha_1, ..., \alpha_l) = (1 - \sum_{j=1}^{l} \alpha_j) f_{D,Q}(\mathbf{0}) + \sum_{j=1}^{l} \alpha_j f_{D,Q}(\boldsymbol{\alpha}_j), \quad \forall(\alpha_1, ..., \alpha_l) \in \Delta_l^{\mathrm{o}},$$

*if and only if*

$$\arg\min_{h \in \mathcal{H}} R_D(h) = \bigcap_{j=1}^{l} \arg\min_{h \in \mathcal{H}} R_{Q_j}(h) \bigcap \arg\min_{h \in \mathcal{H}} R_D^{\mathrm{in}}(h),$$

*where $\mathbf{0}$ is the $1 \times l$ vector, whose elements are $0$, and $\boldsymbol{\alpha}_j$ is the $1 \times l$ vector, whose $j$-th element is $1$ and other elements are $0$, and*

$$R_{Q_j}(h) = \int_{\mathcal{X} \times \{K+1\}} \ell(h(\mathbf{x}), y) \mathrm{d}Q_j(\mathbf{x}, y).$$

*Proof of Lemma 3.* To better understand this proof, we recall the definition of $f_{D,Q}(\alpha_1, ..., \alpha_l)$:

$$f_{D,Q}(\alpha_1, ..., \alpha_l) = \inf_{h \in \mathcal{H}} \left( (1 - \sum_{j=1}^{l} \alpha_j) R_D^{\mathrm{in}}(h) + \sum_{j=1}^{l} \alpha_j R_{Q_j}(h) \right), \quad \forall(\alpha_1, ..., \alpha_l) \in \Delta_l^{\mathrm{o}}$$

**First**, we prove that if

$$f_{D,Q}(\alpha_1, ..., \alpha_l) = (1 - \sum_{j=1}^{l} \alpha_j) f_{D,Q}(\mathbf{0}) + \sum_{j=1}^{l} \alpha_j f_{D,Q}(\boldsymbol{\alpha}_j), \quad \forall(\alpha_1, ..., \alpha_l) \in \Delta_l^{\mathrm{o}},$$

then,

$$\arg\min_{h \in \mathcal{H}} R_D(h) = \bigcap_{j=1}^{l} \arg\min_{h \in \mathcal{H}} R_{Q_j}(h) \bigcap \arg\min_{h \in \mathcal{H}} R_D^{\mathrm{in}}(h).$$

Let $D_{XY} = (1 - \sum_{j=1}^{l} \lambda_j) D_{X_{\mathrm{I}} Y_{\mathrm{I}}} + \sum_{j=1}^{l} \lambda_j Q_j$, for some $(\lambda_1, ..., \lambda_l) \in \Delta_l^{\mathrm{o}}$. Since $D_{XY}$ has finite support set, we have

$$\arg\min_{h \in \mathcal{H}} R_D(h) = \arg\min_{h \in \mathcal{H}} \left( (1 - \sum_{j=1}^{l} \lambda_j) R_D^{\mathrm{in}}(h) + \sum_{j=1}^{l} \lambda_j R_{Q_j}(h) \right) \neq \emptyset.$$

We can find that $h_0 \in \arg\min_{h \in \mathcal{H}} \left( (1 - \sum_{j=1}^{l} \lambda_j) R_D^{\text{in}}(h) + \sum_{j=1}^{l} \lambda_j R_{Q_j}(h) \right)$. Hence,

$$(1 - \sum_{j=1}^{l} \lambda_j) R_D^{\text{in}}(h_0) + \sum_{j=1}^{l} \lambda_j R_{Q_j}(h_0) = \inf_{h \in \mathcal{H}} \left( (1 - \sum_{j=1}^{l} \lambda_j) R_D^{\text{in}}(h) + \sum_{j=1}^{l} \lambda_j R_{Q_j}(h) \right). \quad (28)$$

Note that the condition $f_{D,Q}(\alpha_1, ..., \alpha_l) = (1 - \sum_{j=1}^{l} \alpha_j) f_{D,Q}(\mathbf{0}) + \sum_{j=1}^{l} \alpha_j f_{D,Q}(\boldsymbol{\alpha}_j)$ implies

$$(1 - \sum_{j=1}^{l} \lambda_j) \inf_{h \in \mathcal{H}} R_D^{\text{in}}(h) + \sum_{j=1}^{l} \lambda_j \inf_{h \in \mathcal{H}} R_{Q_j}(h) = \inf_{h \in \mathcal{H}} \left( (1 - \sum_{j=1}^{l} \lambda_j) R_D^{\text{in}}(h) + \sum_{j=1}^{l} \lambda_j R_{Q_j}(h) \right). \quad (29)$$

Therefore, Eq. (28) and Eq. (29) imply that

$$(1 - \sum_{j=1}^{l} \lambda_j) \inf_{h \in \mathcal{H}} R_D^{\text{in}}(h) + \sum_{j=1}^{l} \lambda_j \inf_{h \in \mathcal{H}} R_{Q_j}(h) = (1 - \sum_{j=1}^{l} \lambda_j) R_D^{\text{in}}(h_0) + \sum_{j=1}^{l} \lambda_j R_{Q_j}(h_0). \quad (30)$$

Since $R_D^{\text{in}}(h_0) \geq \inf_{h \in \mathcal{H}} R_D^{\text{in}}(h)$ and $R_{Q_j}(h_0) \geq \inf_{h \in \mathcal{H}} R_{Q_j}^{\text{in}}(h)$, for $j = 1, ..., l$, then using Eq. (30), we have that

$$R_D^{\text{in}}(h_0) = \inf_{h \in \mathcal{H}} R_D^{\text{in}}(h),$$
$$R_{Q_j}(h_0) = \inf_{h \in \mathcal{H}} R_{Q_j}(h), \quad \forall j = 1, ..., l,$$

which implies that

$$h_0 \in \bigcap_{j=1}^{l} \arg\min_{h \in \mathcal{H}} R_{Q_j}(h) \bigcap \arg\min_{h \in \mathcal{H}} R_D^{\text{in}}(h).$$

Therefore,

$$\arg\min_{h \in \mathcal{H}} R_D(h) \subset \bigcap_{j=1}^{l} \arg\min_{h \in \mathcal{H}} R_{Q_j}(h) \bigcap \arg\min_{h \in \mathcal{H}} R_D^{\text{in}}(h). \quad (31)$$

Additionally, using

$$f_{D,Q}(\alpha_1, ..., \alpha_l) = (1 - \sum_{j=1}^{l} \alpha_j) f_{D,Q}(\mathbf{0}) + \sum_{j=1}^{l} \alpha_j f_{D,Q}(\boldsymbol{\alpha}_j), \ \forall (\alpha_1, ..., \alpha_l) \in \Delta_l^{\text{o}},$$

we obtain that for any $h' \in \bigcap_{j=1}^{l} \arg\min_{h \in \mathcal{H}} R_{Q_j}(h) \bigcap \arg\min_{h \in \mathcal{H}} R_D^{\text{in}}(h)$,

$$\inf_{h \in \mathcal{H}} R_D(h) = \inf_{h \in \mathcal{H}} \left( (1 - \sum_{j=1}^{l} \lambda_j) R_D^{\text{in}}(h) + \sum_{j=1}^{l} \lambda_j R_{Q_j}(h) \right)$$

$$= (1 - \sum_{j=1}^{l} \lambda_j) \inf_{h \in \mathcal{H}} R_D^{\text{in}}(h) + \sum_{j=1}^{l} \lambda_j \inf_{h \in \mathcal{H}} R_{Q_j}(h)$$

$$= (1 - \sum_{j=1}^{l} \lambda_j) R_D^{\text{in}}(h') + \sum_{j=1}^{l} \lambda_j R_{Q_j}(h') = R_D(h'),$$

which implies that

$$h' \in \arg\min_{h \in \mathcal{H}} R_D(h).$$

Therefore,

$$\bigcap_{j=1}^{l} \arg\min_{h \in \mathcal{H}} R_{Q_j}(h) \bigcap \arg\min_{h \in \mathcal{H}} R_D^{\text{in}}(h) \subset \arg\min_{h \in \mathcal{H}} R_D(h). \quad (32)$$

Combining Eq. (31) with Eq. (32), we obtain that

$$\bigcap_{j=1}^{l} \underset{h \in \mathcal{H}}{\arg\min}\, R_{Q_j}(h) \bigcap \underset{h \in \mathcal{H}}{\arg\min}\, R_D^{\mathrm{in}}(h) = \underset{h \in \mathcal{H}}{\arg\min}\, R_D(h).$$

**Second**, we prove that if

$$\underset{h \in \mathcal{H}}{\arg\min}\, R_D(h) = \bigcap_{j=1}^{l} \underset{h \in \mathcal{H}}{\arg\min}\, R_{Q_j}(h) \bigcap \underset{h \in \mathcal{H}}{\arg\min}\, R_D^{\mathrm{in}}(h),$$

then,

$$f_{D,Q}(\alpha_1, ..., \alpha_l) = (1 - \sum_{j=1}^{l} \alpha_j) f_{D,Q}(\mathbf{0}) + \sum_{j=1}^{l} \alpha_j f_{D,Q}(\boldsymbol{\alpha}_j), \quad \forall (\alpha_1, ..., \alpha_l) \in \Delta_l^{\mathrm{o}}.$$

We set

$$h_0 \in \bigcap_{j=1}^{l} \underset{h \in \mathcal{H}}{\arg\min}\, R_{Q_j}(h) \bigcap \underset{h \in \mathcal{H}}{\arg\min}\, R_D^{\mathrm{in}}(h),$$

then, for any $(\alpha_1, ..., \alpha_l) \in \Delta_l^{\mathrm{o}}$,

$$(1 - \sum_{j=1}^{l} \alpha_j) \inf_{h \in \mathcal{H}} R_D^{\mathrm{in}}(h) + \sum_{j=1}^{l} \alpha_j \inf_{h \in \mathcal{H}} R_{Q_j}(h) \leq \inf_{h \in \mathcal{H}} \left( (1 - \sum_{j=1}^{l} \alpha_j) R_D^{\mathrm{in}}(h) + \sum_{j=1}^{l} \alpha_j R_{Q_j}(h) \right)$$

$$\leq (1 - \sum_{j=1}^{l} \alpha_j) R_D^{\mathrm{in}}(h_0) + \sum_{j=1}^{l} \alpha_j R_{Q_j}(h_0)$$

$$= (1 - \sum_{j=1}^{l} \alpha_j) \inf_{h \in \mathcal{H}} R_D^{\mathrm{in}}(h) + \sum_{j=1}^{l} \alpha_j \inf_{h \in \mathcal{H}} R_{Q_j}(h).$$

Therefore, for any $(\alpha_1, ..., \alpha_l) \in \Delta_l^{\mathrm{o}}$,

$$(1 - \sum_{j=1}^{l} \alpha_j) \inf_{h \in \mathcal{H}} R_D^{\mathrm{in}}(h) + \sum_{j=1}^{l} \alpha_j \inf_{h \in \mathcal{H}} R_{Q_j}(h) = \inf_{h \in \mathcal{H}} \left( (1 - \sum_{j=1}^{l} \alpha_j) R_D^{\mathrm{in}}(h) + \sum_{j=1}^{l} \alpha_j R_{Q_j}(h) \right),$$

which implies that: for any $(\alpha_1, ..., \alpha_l) \in \Delta_l^{\mathrm{o}}$,

$$f_{D,Q}(\alpha_1, ..., \alpha_l) = (1 - \sum_{j=1}^{l} \alpha_j) f_{D,Q}(\mathbf{0}) + \sum_{j=1}^{l} \alpha_j f_{D,Q}(\boldsymbol{\alpha}_j).$$

We have completed this proof. □

**Lemma 4.** *Suppose that Assumption 1 holds. If there is a finite discrete domain $D_{XY} \in \mathscr{D}_{XY}^s$ such that $\inf_{h \in \mathcal{H}} R_D^{\text{out}}(h) > 0$, then OOD detection is not learnable in $\mathscr{D}_{XY}^s$ for $\mathcal{H}$.*

*Proof of Lemma 4.* Suppose that $\text{supp}D_{X_O} = \{\mathbf{x}_1^{\text{out}}, ..., \mathbf{x}_l^{\text{out}}\}$, then it is clear that $D_{XY}$ has OOD convex decomposition $\delta_{\mathbf{x}_1^{\text{out}}}, ..., \delta_{\mathbf{x}_l^{\text{out}}}$, where $\delta_{\mathbf{x}}$ is the dirac measure whose support set is $\{\mathbf{x}\}$.

Since $\mathcal{H}$ is the separate space for OOD (*i.e.*, Assumption 1 holds), then $\forall j = 1, ..., l$,

$$\inf_{h \in \mathcal{H}} R_{\delta_{\mathbf{x}_j^{\text{out}}}}(h) = 0,$$

where

$$R_{\delta_{\mathbf{x}_j^{\text{out}}}}(h) = \int_{\mathcal{X}} \ell(h(\mathbf{x}), K+1) \mathrm{d}\delta_{\mathbf{x}_j^{\text{out}}}(\mathbf{x}).$$

This implies that: if $\bigcap_{j=1}^l \arg\min_{h \in \mathcal{H}} R_{\delta_{\mathbf{x}_j^{\text{out}}}}(h) \neq \emptyset$, then for $\forall h' \in \bigcap_{j=1}^l \arg\min_{h \in \mathcal{H}} R_{\delta_{\mathbf{x}_j^{\text{out}}}}(h)$,

$$h'(\mathbf{x}_i^{\text{out}}) = K + 1, \ \forall i = 1, ..., l.$$

Therefore, if $\bigcap_{j=1}^l \arg\min_{h \in \mathcal{H}} R_{\delta_{\mathbf{x}_j^{\text{out}}}}(h) \bigcap \arg\min_{h \in \mathcal{H}} R_D^{\text{in}}(h) \neq \emptyset$,

then for any $h^* \in \bigcap_{j=1}^l \arg\min_{h \in \mathcal{H}} R_{\delta_{\mathbf{x}_j^{\text{out}}}}(h) \bigcap \arg\min_{h \in \mathcal{H}} R_D^{\text{in}}(h)$, we have that

$$h^*(\mathbf{x}_i^{\text{out}}) = K + 1, \ \forall i = 1, ..., l.$$

**Proof by Contradiction**: assume OOD detection is learnable in $\mathscr{D}_{XY}^s$ for $\mathcal{H}$, then Lemmas 1 and 3 imply that

$$\bigcap_{j=1}^l \arg\min_{h \in \mathcal{H}} R_{\delta_{\mathbf{x}_j^{\text{out}}}}(h) \bigcap \arg\min_{h \in \mathcal{H}} R_D^{\text{in}}(h) = \arg\min_{h \in \mathcal{H}} R_D(h) \neq \emptyset.$$

Therefore, for any $h^* \in \arg\min_{h \in \mathcal{H}} R_D(h)$, we have that

$$h^*(\mathbf{x}_i^{\text{out}}) = K + 1, \ \forall i = 1, ..., l,$$

which implies that for any $h^* \in \arg\min_{h \in \mathcal{H}} R_D(h)$, we have $R_D^{\text{out}}(h^*) = 0$, which implies that $\inf_{h \in \mathcal{H}} R_D^{\text{out}}(h) = 0$.

It is clear that $\inf_{h \in \mathcal{H}} R_D^{\text{out}}(h) = 0$ is **inconsistent** with the condition $\inf_{h \in \mathcal{H}} R_D^{\text{out}}(h) > 0$. Therefore, OOD detection is not learnable in $\mathscr{D}_{XY}^s$ for $\mathcal{H}$. $\qquad\square$

**Lemma 5.** *If Assumption 1 holds, $\text{VCdim}(\phi \circ \mathcal{H}) = v < +\infty$ and $\sup_{h \in \mathcal{H}} |\{\mathbf{x} \in \mathcal{X} : h(\mathbf{x}) \in \mathcal{Y}\}| > m$ such that $v < m$, then OOD detection is not learnable in $\mathscr{D}_{XY}^s$ for $\mathcal{H}$, where $\phi$ maps ID's labels to 1 and maps OOD's labels to 2.*

*Proof of Lemma 5.* Due to $\sup_{h \in \mathcal{H}} |\{\mathbf{x} \in \mathcal{X} : h(\mathbf{x}) \in \mathcal{Y}\}| > m$, we can obtain a set

$$C = \{\mathbf{x}_1, ..., \mathbf{x}_m, \mathbf{x}_{m+1}\},$$

which satisfies that there exists $\tilde{h} \in \mathcal{H}$ such that $\tilde{h}(\mathbf{x}_i) \in \mathcal{Y}$ for any $i = 1, ..., m, m+1$.

Let $\mathcal{H}_C^\phi = \{(\phi \circ h(\mathbf{x}_1), ..., \phi \circ h(\mathbf{x}_m), \phi \circ h(\mathbf{x}_{m+1}) : h \in \mathcal{H}\}$. It is clear that

$$(1, 1, ..., 1) = (\phi \circ \tilde{h}(\mathbf{x}_1), ..., \phi \circ \tilde{h}(\mathbf{x}_m), \phi \circ \tilde{h}(\mathbf{x}_{m+1})) \in \mathcal{H}_C^\phi,$$

where $(1, 1, ..., 1)$ means all elements are 1.

Let $\mathcal{H}_{m+1}^\phi = \{(\phi \circ h(\mathbf{x}_1), ..., \phi \circ h(\mathbf{x}_m), \phi \circ h(\mathbf{x}_{m+1}) : h \text{ is any hypothesis function from } \mathcal{X} \text{ to } \mathcal{Y}_{\text{all}}\}$.

Clearly, $\mathcal{H}_C^\phi \subset \mathcal{H}_{m+1}^\phi$ and $|\mathcal{H}_{m+1}^\phi| = 2^{m+1}$. Sauer-Shelah-Perles Lemma (Lemma 6.10 in [21]) implies that

$$|\mathcal{H}_C^\phi| \leq \sum_{i=0}^v \binom{m+1}{i}.$$

Since $\sum_{i=0}^{v} \binom{m+1}{i} < 2^{m+1} - 1$ (because $v < m$), we obtain that $|\mathcal{H}_C^{\phi}| \leq 2^{m+1} - 2$. Therefore, $\mathcal{H}_C^{\phi} \cup \{(2, 2..., 2)\}$ is a proper subset of $\mathcal{H}_{m+1}^{\phi}$, where $(2, 2, ..., 2)$ means that all elements are 2. Note that $(1, 1..., 1)$ (all elements are 1) also belongs to $\mathcal{H}_C^{\phi}$. Hence, $\mathcal{H}_C^{\phi} \cup \{(2, 2..., 2)\} \cup \{(1, 1..., 1)\}$ is a proper subset of $\mathcal{H}_{m+1}^{\phi}$, which implies that we can obtain a hypothesis function $h'$ satisfying that:

1)$(\phi \circ h'(\mathbf{x}_1), ..., \phi \circ h'(\mathbf{x}_m), \phi \circ h'(\mathbf{x}_{m+1})) \notin \mathcal{H}_C^{\phi}$;

2) There exist $\mathbf{x}_j, \mathbf{x}_p \in C$ such that $\phi \circ h'(\mathbf{x}_j) = 2$ and $\phi \circ h'(\mathbf{x}_p) = 1$.

Let $C_{\mathrm{I}} = C \cap \{\mathbf{x} \in \mathcal{X} : \phi \circ h'(\mathbf{x}) = 1\}$ and $C_{\mathrm{O}} = C \cap \{\mathbf{x} \in \mathcal{X} : \phi \circ h'(\mathbf{x}) = 2\}$;

Then, we construct a special domain $D_{XY}$:

$$D_{XY} = 0.5 * D_{X_{\mathrm{I}}} * D_{Y_{\mathrm{I}}|X_{\mathrm{I}}} + 0.5 * D_{X_{\mathrm{O}}} * D_{Y_{\mathrm{O}}|X_{\mathrm{O}}}, \text{ where}$$

$$D_{X_{\mathrm{I}}} = \frac{1}{|C_{\mathrm{I}}|} \sum_{\mathbf{x} \in C_{\mathrm{I}}} \delta_{\mathbf{x}} \text{ and } D_{Y_{\mathrm{I}}|X_{\mathrm{I}}}(y|\mathbf{x}) = 1, \text{ if } \tilde{h}(\mathbf{x}) = y \text{ and } \mathbf{x} \in C_{\mathrm{I}};$$

and

$$D_{X_{\mathrm{O}}} = \frac{1}{|C_{\mathrm{O}}|} \sum_{\mathbf{x} \in C_{\mathrm{O}}} \delta_{\mathbf{x}} \text{ and } D_{Y_{\mathrm{O}}|X_{\mathrm{O}}}(K+1|\mathbf{x}) = 1, \text{ if } \mathbf{x} \in C_{\mathrm{O}}.$$

Since $D_{XY}$ is a finite discrete distribution and $(\phi \circ h'(\mathbf{x}_1), ..., \phi \circ h'(\mathbf{x}_m), \phi \circ h'(\mathbf{x}_{m+1})) \notin \mathcal{H}_C^{\phi}$, it is clear that $\arg\min_{h \in \mathcal{H}} R_D(h) \neq \emptyset$ and $\inf_{h \in \mathcal{H}} R_D(h) > 0$.

Additionally, $R_D^{\mathrm{in}}(\tilde{h}) = 0$. Therefore, $\inf_{h \in \mathcal{H}} R_D^{\mathrm{in}}(h) = 0$.

**Proof by Contradiction**: suppose that OOD detection is learnable in $\mathscr{D}_{XY}^s$ for $\mathcal{H}$, then Lemma 1 implies that

$$\inf_{h \in \mathcal{H}} R_D(h) = 0.5 * \inf_{h \in \mathcal{H}} R_D^{\mathrm{in}}(h) + 0.5 * \inf_{h \in \mathcal{H}} R_D^{\mathrm{out}}(h).$$

Therefore, if OOD detection is learnable in $\mathscr{D}_{XY}^s$ for $\mathcal{H}$, then $\inf_{h \in \mathcal{H}} R_D^{\mathrm{out}}(h) > 0$.

Until now, we have constructed a domain $D_{XY}$ (defined over $\mathcal{X} \times \mathcal{Y}_{\mathrm{all}}$) with finite support and satisfying that $\inf_{h \in \mathcal{H}} R_D^{\mathrm{out}}(h) > 0$. Note that $\mathcal{H}$ is the separate space for OOD data (Assumption 1 holds). Using Lemma 4, we know that OOD detection is not learnable in $\mathscr{D}_{XY}^s$ for $\mathcal{H}$, which is **inconsistent** with our assumption that OOD detection is learnable in $\mathscr{D}_{XY}^s$ for $\mathcal{H}$. Therefore, OOD detection is not learnable in $\mathscr{D}_{XY}^s$ for $\mathcal{H}$. We have completed the proof. $\square$

**Theorem 5** (Impossibility Theorem for Separate Space). *If Assumption 1 holds,* $\mathrm{VCdim}(\phi \circ \mathcal{H}) < +\infty$ *and* $\sup_{h \in \mathcal{H}} |\{\mathbf{x} \in \mathcal{X} : h(\mathbf{x}) \in \mathcal{Y}\}| = +\infty$, *then OOD detection is not learnable in separate space* $\mathscr{D}_{XY}^s$ *for* $\mathcal{H}$, *where* $\phi$ *maps ID labels to 1 and maps OOD labels to 2.*

*Proof of Theorem 5.* Let $\mathrm{VCdim}(\phi \circ \mathcal{H}) = v$. Since $\sup_{h \in \mathcal{H}} |\{\mathbf{x} \in \mathcal{X} : h(\mathbf{x}) \in \mathcal{Y}\}| = +\infty$, it is clear that $\sup_{h \in \mathcal{H}} |\{\mathbf{x} \in \mathcal{X} : h(\mathbf{x}) \in \mathcal{Y}\}| > v$. Using Lemma 5, we complete this proof. $\square$

# I  Proofs of Theorem 6 and Theorem 7

## I.1  Proof of Theorem 6

Firstly, we need two lemmas, which are motivated by Lemma 19.2 and Lemma 19.3 in [21].

**Lemma 6.** *Let* $C_1, ..., C_r$ *be a cover of space* $\mathcal{X}$, *i.e.,* $\sum_{i=1}^{r} C_i = \mathcal{X}$. *Let* $S_X = \{\mathbf{x}^1, ..., \mathbf{x}^n\}$ *be a sequence of* $n$ *data drawn from* $D_{X_{\mathrm{I}}}$, *i.i.d. Then*

$$\mathbb{E}_{S_X \sim D_{X_{\mathrm{I}}}^n} \Big( \sum_{i:C_i \cap S_X = \emptyset} D_{X_{\mathrm{I}}}(C_i) \Big) \leq \frac{r}{en}.$$

*Proof of Lemma 6.*

$$\mathbb{E}_{S_X \sim D_{X_{\mathrm{I}}}^n} \Big( \sum_{i:C_i \cap S_X = \emptyset} D_{X_{\mathrm{I}}}(C_i) \Big) = \sum_{i=1}^{r} \Big( D_{X_{\mathrm{I}}}(C_i) \cdot \mathbb{E}_{S_X \sim D_{X_{\mathrm{I}}}^n} \big( \mathbf{1}_{C_i \cap S_X = \emptyset} \big) \Big),$$

where $\mathbf{1}$ is the characteristic function.

For each $i$,

$$\mathbb{E}_{S_X \sim D_{X_{\mathrm{I}}}^n} \big( \mathbf{1}_{C_i \cap S_X = \emptyset} \big) = \int_{\mathcal{X}^n} \mathbf{1}_{C_i \cap S_X = \emptyset} \mathrm{d} D_{X_{\mathrm{I}}}^n(S_X)$$

$$= \Big( \int_{\mathcal{X}} \mathbf{1}_{C_i \cap \{\mathbf{x}\} = \emptyset} \mathrm{d} D_{X_{\mathrm{I}}}(\mathbf{x}) \Big)^n$$

$$= \big( 1 - D_{X_{\mathrm{I}}}(C_i) \big)^n \le e^{-n D_{X_{\mathrm{I}}}(C_i)}.$$

Therefore,

$$\mathbb{E}_{S_X \sim D_{X_{\mathrm{I}}}^n} \Big( \sum_{i:C_i \cap S = \emptyset} D_{X_{\mathrm{I}}}(C_i) \Big) \le \sum_{i=1}^{r} D_{X_{\mathrm{I}}}(C_i) e^{-n D_{X_{\mathrm{I}}}(C_i)}$$

$$\le r \max_{i \in \{1,...,r\}} D_{X_{\mathrm{I}}}(C_i) e^{-n D_{X_{\mathrm{I}}}(C_i)} \le \frac{r}{ne},$$

here we have used inequality: $\max_{i \in \{1,...,r\}} a_i e^{-n a_i} \le 1/(ne)$. The proof has been completed. $\square$

**Lemma 7.** *Let $K = 1$. When $\mathcal{X} \subset \mathbb{R}^d$ is a bounded set, there exists a monotonically decreasing sequence $\epsilon_{\mathrm{cons}}(m)$ satisfying that $\epsilon_{\mathrm{cons}}(m) \to 0$, as $m \to 0$, such that*

$$\mathbb{E}_{\mathbf{x} \sim D_{X_{\mathrm{I}}}, S \sim D_{X_{\mathrm{I}} Y_{\mathrm{I}}}^n} \mathrm{dist}(\mathbf{x}, \pi_1(\mathbf{x}, S)) < \epsilon_{\mathrm{cons}}(n),$$

*where dist is the Euclidean distance, $\pi_1(\mathbf{x}, S) = \arg\min_{\tilde{\mathbf{x}} \in S_X} \mathrm{dist}(\mathbf{x}, \tilde{\mathbf{x}})$, here $S_X$ is the feature part of $S$, i.e., $S_X = \{\mathbf{x}^1, ..., \mathbf{x}^n\}$, if $S = \{(\mathbf{x}^1, y^1), ..., (\mathbf{x}^n, y^n)\}$.*

*Proof of Lemma 7.* Since $\mathcal{X}$ is bounded, without loss of generality, we set $\mathcal{X} \subset [0, 1)^d$. Fix $\epsilon = 1/T$, for some integer $T$. Let $r = T^d$ and $C_1, C_2, ..., C_r$ be a cover of $\mathcal{X}$: for every $(a_1, ..., a_T) \in [T]^d :=$ $[1, ..., T]^d$, there exists a $C_i = \{\mathbf{x} = (x_1, ..., x_d) : \forall j \in \{1, ..., d\}, x_j \in [(a_j - 1)/T, a_j/T)\}$.

If $\mathbf{x}, \mathbf{x}'$ belong to some $C_i$, then $\mathrm{dist}(\mathbf{x}, \mathbf{x}') \le \sqrt{d}\epsilon$; otherwise, $\mathrm{dist}(\mathbf{x}, \mathbf{x}') \le \sqrt{d}$. Therefore,

$$\mathbb{E}_{\mathbf{x} \sim D_{X_{\mathrm{I}}}, S \sim D_{X_{\mathrm{I}} Y_{\mathrm{I}}}^n} \mathrm{dist}(\mathbf{x}, \pi_1(\mathbf{x}, S))$$

$$\le \mathbb{E}_{S \sim D_{X_{\mathrm{I}} Y_{\mathrm{I}}}^n} \Big( \sqrt{d}\epsilon \sum_{i:C_i \cap S_X \ne \emptyset} D_{X_{\mathrm{I}}}(C_i) + \sqrt{d} \sum_{i:C_i \cap S_X = \emptyset} D_{X_{\mathrm{I}}}(C_i) \Big)$$

$$\le \mathbb{E}_{S_X \sim D_{X_{\mathrm{I}}}^n} \Big( \sqrt{d}\epsilon \sum_{i:C_i \cap S_X \ne \emptyset} D_{X_{\mathrm{I}}}(C_i) + \sqrt{d} \sum_{i:C_i \cap S_X = \emptyset} D_{X_{\mathrm{I}}}(C_i) \Big).$$

Note that $C_1, ..., C_r$ are disjoint.

Therefore, $\sum_{i:C_i \cap S_X \ne \emptyset} D_{X_{\mathrm{I}}}(C_i) \le D_{X_{\mathrm{I}}}(\sum_{i:C_i \cap S_X \ne \emptyset} C_i) \le 1$. Using Lemma 6, we obtain

$$\mathbb{E}_{\mathbf{x} \sim D_{X_{\mathrm{I}}}, S \sim D_{X_{\mathrm{I}} Y_{\mathrm{I}}}^n} \mathrm{dist}(\mathbf{x}, \pi_1(\mathbf{x}, S)) \le \sqrt{d}\epsilon + \frac{r\sqrt{d}}{ne} = \sqrt{d}\epsilon + \frac{\sqrt{d}}{ne\epsilon^d}.$$

If we set $\epsilon = 2n^{-1/(d+1)}$, then

$$\mathbb{E}_{\mathbf{x} \sim D_{X_{\mathrm{I}}}, S \sim D_{X_{\mathrm{I}} Y_{\mathrm{I}}}^n} \mathrm{dist}(\mathbf{x}, \pi_1(\mathbf{x}, S)) \le \frac{2\sqrt{d}}{n^{1/(d+1)}} + \frac{\sqrt{d}}{2^d e n^{1/(d+1)}}.$$

If we set $\epsilon_{\mathrm{cons}}(n) = \frac{2\sqrt{d}}{n^{1/(d+1)}} + \frac{\sqrt{d}}{2^d e n^{1/(d+1)}}$, we complete this proof. $\square$

**Theorem 6.** *Let $K = 1$ and $|\mathcal{X}| < +\infty$. Suppose that Assumption 1 holds and the constant function $h^{\mathrm{in}} := 1 \in \mathcal{H}$. Then OOD detection is learnable in $\mathscr{D}_{XY}^s$ for $\mathcal{H}$ **if and only if** $\mathcal{H}_{\mathrm{all}} - \{h^{\mathrm{out}}\} \subset \mathcal{H}$, where $\mathcal{H}_{\mathrm{all}}$ is the hypothesis space consisting of all hypothesis functions, and $h^{\mathrm{out}}$ is a constant function that $h^{\mathrm{out}} := 2$, here $1$ represents ID data and $2$ represents OOD data.*

*Proof of Theorem 6.* **First**, we prove that if the hypothesis space $\mathcal{H}$ is a separate space for OOD (*i.e.*, Assumption 1 holds), the constant function $h^{\mathrm{in}} := 1 \in \mathcal{H}$, then that OOD detection is learnable in $\mathscr{D}_{XY}^s$ for $\mathcal{H}$ implies $\mathcal{H}_{\mathrm{all}} - \{h^{\mathrm{out}}\} \subset \mathcal{H}$.

**Proof by Contradiction**: suppose that there exists $h' \in \mathcal{H}_{\mathrm{all}}$ such that $h' \neq h^{\mathrm{out}}$ and $h' \notin \mathcal{H}$.

Let $\mathcal{X} = \{\mathbf{x}_1, ..., \mathbf{x}_m\}$, $C_{\mathrm{I}} = \{\mathbf{x} \in \mathcal{X} : h'(\mathbf{x}) \in \mathcal{Y}\}$ and $C_{\mathrm{O}} = \{\mathbf{x} \in \mathcal{X} : h'(\mathbf{x}) = K + 1\}$.

Because $h' \neq h^{\mathrm{out}}$, we know that $C_{\mathrm{I}} \neq \emptyset$.

We construct a special domain $D_{XY} \in \mathscr{D}_{XY}^s$: if $C_{\mathrm{O}} = \emptyset$, then $D_{XY} = D_{X_{\mathrm{I}}} * D_{Y_{\mathrm{I}}|X_{\mathrm{I}}}$; otherwise,

$$D_{XY} = 0.5 * D_{X_{\mathrm{I}}} * D_{Y_{\mathrm{I}}|X_{\mathrm{I}}} + 0.5 * D_{X_{\mathrm{O}}} * D_{Y_{\mathrm{O}}|X_{\mathrm{O}}}, \quad \text{where}$$

$$D_{X_{\mathrm{I}}} = \frac{1}{|C_{\mathrm{I}}|} \sum_{\mathbf{x} \in C_{\mathrm{I}}} \delta_{\mathbf{x}} \quad \text{and} \quad D_{Y_{\mathrm{I}}|X_{\mathrm{I}}}(y|\mathbf{x}) = 1, \ \text{ if } h'(\mathbf{x}) = y \text{ and } \mathbf{x} \in C_{\mathrm{I}},$$

and

$$D_{X_{\mathrm{O}}} = \frac{1}{|C_{\mathrm{O}}|} \sum_{\mathbf{x} \in C_{\mathrm{O}}} \delta_{\mathbf{x}} \quad \text{and} \quad D_{Y_{\mathrm{O}}|X_{\mathrm{O}}}(K+1|\mathbf{x}) = 1, \ \text{ if } \mathbf{x} \in C_{\mathrm{O}}.$$

Since $h' \notin \mathcal{H}$ and $|\mathcal{X}| < +\infty$, then $\arg\min_{h \in \mathcal{H}} R_D(h) \neq \emptyset$, and $\inf_{h \in \mathcal{H}} R_D(h) > 0$. Additionally, $R_D^{\mathrm{in}}(h^{\mathrm{in}}) = 0$ (here $h^{\mathrm{in}} = 1$), hence, $\inf_{h \in \mathcal{H}} R_D^{\mathrm{in}}(h) = 0$.

Since OOD detection is learnable in $\mathscr{D}_{XY}^s$ for $\mathcal{H}$, Lemma 1 implies that

$$\inf_{h \in \mathcal{H}} R_D(h) = (1 - \pi^{\mathrm{out}}) \inf_{h \in \mathcal{H}} R_D^{\mathrm{in}}(h) + \pi^{\mathrm{out}} \inf_{h \in \mathcal{H}} R_D^{\mathrm{out}}(h),$$

where $\pi^{\mathrm{out}} = D_Y(Y = K + 1) = 1$ or $0.5$. Since $\inf_{h \in \mathcal{H}} R_D^{\mathrm{in}}(h) = 0$ and $\inf_{h \in \mathcal{H}} R_D(h) > 0$, we obtain that $\inf_{h \in \mathcal{H}} R_D^{\mathrm{out}}(h) > 0$.

Until now, we have constructed a special domain $D_{XY} \in \mathscr{D}_{XY}^s$ satisfying that $\inf_{h \in \mathcal{H}} R_D^{\mathrm{out}}(h) > 0$. Using Lemma 4, we know that OOD detection in $\mathscr{D}_{XY}^s$ is not learnable for $\mathcal{H}$, which is **inconsistent** with the condition that OOD detection is learnable in $\mathscr{D}_{XY}^s$ for $\mathcal{H}$. Therefore, the assumption (there exists $h' \in \mathcal{H}_{\mathrm{all}}$ such that $h' \neq h^{\mathrm{out}}$ and $h \notin \mathcal{H}$) doesn't hold, which implies that $\mathcal{H}_{\mathrm{all}} - \{h^{\mathrm{out}}\} \subset \mathcal{H}$.

**Second**, we prove that if $\mathcal{H}_{\mathrm{all}} - \{h^{\mathrm{out}}\} \subset \mathcal{H}$, then OOD detection is learnable in $\mathscr{D}_{XY}^s$ for $\mathcal{H}$.

To prove this result, we need to design a special algorithm. Let $d_0 = \min_{\mathbf{x}, \mathbf{x}' \in \mathcal{X} \text{ and } \mathbf{x} \neq \mathbf{x}'} \mathrm{dist}(\mathbf{x}, \mathbf{x}')$, where $\mathrm{dist}$ is the Euclidean distance. It is clear that $d_0 > 0$. Let

$$\mathbf{A}(S)(\mathbf{x}) = \begin{cases} 1, & \text{if } \mathrm{dist}(\mathbf{x}, \pi_1(\mathbf{x}, S)) < 0.5 * d_0; \\ 2, & \text{if } \mathrm{dist}(\mathbf{x}, \pi_1(\mathbf{x}, S)) \geq 0.5 * d_0, \end{cases}$$

where $\pi_1(\mathbf{x}, S) = \arg\min_{\tilde{\mathbf{x}} \in S_X} \mathrm{dist}(\mathbf{x}, \tilde{\mathbf{x}})$, here $S_X$ is the feature part of $S$, *i.e.*, $S_X = \{\mathbf{x}^1, ..., \mathbf{x}^n\}$, if $S = \{(\mathbf{x}^1, y^1), ..., (\mathbf{x}^n, y^n)\}$.

For any $\mathbf{x} \in \mathrm{supp} D_{X_{\mathrm{I}}}$, it is easy to check that for almost all $S \sim D_{X_{\mathrm{I}} Y_{\mathrm{I}}}^n$,

$$\mathrm{dist}(\mathbf{x}, \pi_1(\mathbf{x}, S)) > 0.5 * d_0,$$

which implies that

$$\mathbf{A}(S)(\mathbf{x}) = 2,$$

hence,

$$\mathbb{E}_{S \sim D_{X_{\mathrm{I}} Y_{\mathrm{I}}}^n} R_D^{\mathrm{out}}(\mathbf{A}(S)) = 0. \tag{33}$$

Using Lemma 7, for any $\mathbf{x} \in \mathrm{supp} D_{X_{\mathrm{I}}}$, we have

$$\mathbb{E}_{\mathbf{x} \sim D_{X_{\mathrm{I}}}, S \sim D_{X_{\mathrm{I}} Y_{\mathrm{I}}}^n} \mathrm{dist}(\mathbf{x}, \pi_1(\mathbf{x}, S)) < \epsilon_{\mathrm{cons}}(n),$$

where $\epsilon_{\mathrm{cons}}(n) \to 0$, as $n \to 0$ and $\epsilon_{\mathrm{cons}}(n)$ is a monotonically decreasing sequence.

Hence, we have that

$$D_{X_\mathrm{I}} \times D_{X_\mathrm{I}Y_\mathrm{I}}^n(\{(\mathbf{x}, S) : \mathrm{dist}(\mathbf{x}, \pi_1(\mathbf{x}, S)) \geq 0.5 * d_0\}) \leq 2\epsilon_{\mathrm{cons}}(n)/d_0,$$

where $D_{X_\mathrm{I}} \times D_{X_\mathrm{I}Y_\mathrm{I}}^n$ is the product measure of $D_{X_\mathrm{I}}$ and $D_{X_\mathrm{I}Y_\mathrm{I}}^n$ [36]. Therefore,

$$D_{X_\mathrm{I}} \times D_{X_\mathrm{I}Y_\mathrm{I}}^n(\{(\mathbf{x}, S) : \mathbf{A}(S)(\mathbf{x}) = 1\}) > 1 - 2\epsilon_{\mathrm{cons}}(n)/d_0,$$

which implies that

$$\mathbb{E}_{S \sim D_{X_\mathrm{I}Y_\mathrm{I}}^n} R_D^{\mathrm{in}}(\mathbf{A}(S)) \leq 2B\epsilon_{\mathrm{cons}}(n)/d_0, \tag{34}$$

where $B = \max\{\ell(1,2), \ell(2,1)\}$. Using Eq. (33) and Eq. (34), we have proved that

$$\mathbb{E}_{S \sim D_{X_\mathrm{I}Y_\mathrm{I}}^n} R_D(\mathbf{A}(S)) \leq 0 + 2B\epsilon_{\mathrm{cons}}(m)/d_0 \leq \inf_{h \in \mathcal{H}} R_D(h) + 2B\epsilon_{\mathrm{cons}}(m)/d_0. \tag{35}$$

It is easy to check that $\mathbf{A}(S) \in \mathcal{H}_{\mathrm{all}} - \{h^{\mathrm{out}}\}$. Therefore, we have constructed a consistent algorithm $\mathbf{A}$ for $\mathcal{H}$. We have completed this proof. □

## I.2  Proof of Theorem 7

**Theorem 7.** *Let $|\mathcal{X}| < +\infty$ and $\mathcal{H} = \mathcal{H}^{\mathrm{in}} \bullet \mathcal{H}^{\mathrm{b}}$. If $\mathcal{H}_{\mathrm{all}} - \{h^{\mathrm{out}}\} \subset \mathcal{H}^{\mathrm{b}}$ and Condition 2 holds, then OOD detection is learnable in $\mathscr{D}_{XY}^s$ for $\mathcal{H}$, where $\mathcal{H}_{\mathrm{all}}$ and $h^{\mathrm{out}}$ are defined in Theorem 6.*

*Proof of Theorem 7.* Since $|\mathcal{X}| < +\infty$, we know that $|\mathcal{H}| < +\infty$, which implies that $\mathcal{H}^{\mathrm{in}}$ is agnostic PAC learnable for supervised learning in classification. Therefore, there exist an algorithm $\mathbf{A}^{\mathrm{in}} : \cup_{n=1}^{+\infty}(\mathcal{X} \times \mathcal{Y})^n \to \mathcal{H}^{\mathrm{in}}$ and a monotonically decreasing sequence $\epsilon(n)$, such that $\epsilon(n) \to 0$, as $n \to +\infty$, and for any $D_{XY} \in \mathscr{D}_{XY}^s$,

$$\mathbb{E}_{S \sim D_{X_\mathrm{I}Y_\mathrm{I}}^n} R_D^{\mathrm{in}}(\mathbf{A}^{\mathrm{in}}(S)) \leq \inf_{h \in \mathcal{H}^{\mathrm{in}}} R_D^{\mathrm{in}}(h) + \epsilon(n).$$

Since $|\mathcal{X}| < +\infty$ and $\mathcal{H}^{\mathrm{b}}$ almost contains all binary classifiers, then using Theorem 6 and Theorem 1, we obtain that there exist an algorithm $\mathbf{A}^{\mathrm{b}} : \cup_{n=1}^{+\infty}(\mathcal{X} \times \{1, 2\})^n \to \mathcal{H}^{\mathrm{b}}$ and a monotonically decreasing sequence $\epsilon'(n)$, such that $\epsilon'(n) \to 0$, as $n \to +\infty$, and for any $D_{XY} \in \mathscr{D}_{XY}^s$,

$$\mathbb{E}_{S \sim D_{X_\mathrm{I}Y_\mathrm{I}}^n} R_{\phi(D)}^{\mathrm{in}}(\mathbf{A}^{\mathrm{b}}(\phi(S))) \leq \inf_{h \in \mathcal{H}^{\mathrm{b}}} R_{\phi(D)}^{\mathrm{in}}(h) + \epsilon'(n),$$

$$\mathbb{E}_{S \sim D_{X_\mathrm{I}Y_\mathrm{I}}^n} R_{\phi(D)}^{\mathrm{out}}(\mathbf{A}^{\mathrm{b}}(\phi(S))) \leq \inf_{h \in \mathcal{H}^{\mathrm{b}}} R_{\phi(D)}^{\mathrm{out}}(h) + \epsilon'(n),$$

where $\phi$ maps ID's labels to 1 and OOD's label to 2,

$$R_{\phi(D)}^{\mathrm{in}}(\mathbf{A}^{\mathrm{b}}(\phi(S))) = \int_{\mathcal{X} \times \mathcal{Y}} \ell(\mathbf{A}^{\mathrm{b}}(\phi(S))(\mathbf{x}), \phi(y))\mathrm{d}D_{X_\mathrm{I}Y_\mathrm{I}}(\mathbf{x}, y), \tag{36}$$

$$R_{\phi(D)}^{\mathrm{in}}(h) = \int_{\mathcal{X} \times \mathcal{Y}} \ell(h(\mathbf{x}), \phi(y))\mathrm{d}D_{X_\mathrm{I}Y_\mathrm{I}}(\mathbf{x}, y), \tag{37}$$

$$R_{\phi(D)}^{\mathrm{out}}(\mathbf{A}^{\mathrm{b}}(\phi(S))) = \int_{\mathcal{X} \times \{K+1\}} \ell(\mathbf{A}^{\mathrm{b}}(\phi(S))(\mathbf{x}), \phi(y))\mathrm{d}D_{X_\mathrm{O}Y_\mathrm{O}}(\mathbf{x}, y), \tag{38}$$

and

$$R_{\phi(D)}^{\mathrm{out}}(h) = \int_{\mathcal{X} \times \{K+1\}} \ell(h(\mathbf{x}), \phi(y))\mathrm{d}D_{X_\mathrm{O}Y_\mathrm{O}}(\mathbf{x}, y), \tag{39}$$

here $\phi(S) = \{(\mathbf{x}^1, \phi(y^1)), ..., (\mathbf{x}^n, \phi(y^n))\}$, if $S = \{(\mathbf{x}^1, y^1), ..., (\mathbf{x}^n, y^n)\}$.

Note that $\mathcal{H}^b$ almost contains all classifiers, and $\mathscr{D}_{XY}^s$ is the separate space. Hence,

$$\mathbb{E}_{S \sim D_{X_\mathrm{I}Y_\mathrm{I}}^n} R_{\phi(D)}^{\mathrm{in}}(\mathbf{A}^{\mathrm{b}}(\phi(S))) \leq \epsilon'(n), \quad \mathbb{E}_{S \sim D_{X_\mathrm{I}Y_\mathrm{I}}^n} R_{\phi(D)}^{\mathrm{out}}(\mathbf{A}^{\mathrm{b}}(\phi(S))) \leq \epsilon'(n).$$

**Next**, we construct an algorithm $\mathbf{A}$ using $\mathbf{A}^{\mathrm{in}}$ and $\mathbf{A}^{\mathrm{out}}$.

$$\mathbf{A}(S)(\mathbf{x}) = \begin{cases} K+1, & \text{if } \mathbf{A}^{\mathrm{b}}(\phi(S))(\mathbf{x}) = 2; \\ \mathbf{A}^{\mathrm{in}}(S)(\mathbf{x}), & \text{if } \mathbf{A}^{\mathrm{b}}(\phi(S))(\mathbf{x}) = 1. \end{cases}$$

Since $\inf_{h \in \mathcal{H}} R^{\mathrm{in}}_{\phi(D)}(\phi \circ h) = 0$, $\inf_{h \in \mathcal{H}} R^{\mathrm{out}}_D(h) = 0$, then by Condition 2, it is easy to check that

$$\inf_{h \in \mathcal{H}^{\mathrm{in}}} R^{\mathrm{in}}_D(h) = \inf_{h \in \mathcal{H}} R^{\mathrm{in}}_D(h).$$

Additionally, the risk $R^{\mathrm{in}}_D(\mathbf{A}(S))$ is from two parts: 1) ID data are detected as OOD data; 2) ID data are detected as ID data, but are classified as incorrect ID classes. Therefore, we have the inequality:

$$\begin{aligned} \mathbb{E}_{S \sim D^n_{X_I Y_I}} R^{\mathrm{in}}_D(\mathbf{A}(S)) &\le \mathbb{E}_{S \sim D^n_{X_I Y_I}} R^{\mathrm{in}}_D(\mathbf{A}^{\mathrm{in}}(S)) + c \mathbb{E}_{S \sim D^n_{X_I Y_I}} R^{\mathrm{in}}_{\phi(D)}(\mathbf{A}^{\mathrm{b}}(\phi(S))) \\ &\le \inf_{h \in \mathcal{H}^{\mathrm{in}}} R^{\mathrm{in}}_D(h) + \epsilon(n) + c\epsilon'(n) = \inf_{h \in \mathcal{H}} R^{\mathrm{in}}_D(h) + \epsilon(n) + c\epsilon'(n), \end{aligned} \tag{40}$$

where $c = \max_{y_1, y_2 \in \mathcal{Y}} \ell(y_1, y_2) / \min\{\ell(1,2), \ell(2,1)\}$.

Note that the risk $R^{\mathrm{out}}_D(\mathbf{A}(S))$ is from the case that OOD data are detected as ID data. Therefore,

$$\begin{aligned} \mathbb{E}_{S \sim D^n_{X_I Y_I}} R^{\mathrm{out}}_D(\mathbf{A}(S)) &\le c \mathbb{E}_{S \sim D^n_{X_{|rmI} Y_I}} R^{\mathrm{out}}_{\phi(D)}(\mathbf{A}^{\mathrm{b}}(\phi(S))) \\ &\le c\epsilon'(n) \le \inf_{h \in \mathcal{H}} R^{\mathrm{out}}_D(h) + c\epsilon'(n). \end{aligned} \tag{41}$$

Note that $(1-\alpha)\inf_{h \in \mathcal{H}} R^{\mathrm{in}}_D(h) + \alpha \inf_{h \in \mathcal{H}} R^{\mathrm{out}}_D(h) \le \inf_{h \in \mathcal{H}} R^{\alpha}_D(h)$. Then, using Eq. (40) and Eq. (41), we obtain that for any $\alpha \in [0, 1]$,

$$\mathbb{E}_{S \sim D^n_{X_I Y_I}} R^{\alpha}_D(\mathbf{A}(S)) \le \inf_{h \in \mathcal{H}} R^{\alpha}_D(h) + \epsilon(n) + c\epsilon'(n).$$

According to Theorem 1 (the second result), we complete the proof. $\square$

# J  Proofs of Theorems 8 and 9

## J.1  Proof of Theorem 8

**Lemma 8.** *Given a prior-unknown space $\mathscr{D}_{XY}$ and a hypothesis space $\mathcal{H}$, if Condition 3 holds, then for any equivalence class $[D'_{XY}]$ with respect to $\mathscr{D}_{XY}$, OOD detection is learnable in the equivalence class $[D'_{XY}]$ for $\mathcal{H}$. Furthermore, the learning rate can attain $O(1/n)$.*

*Proof.* Let $\mathscr{F}$ be a set consisting of all infinite sequences, whose coordinates are hypothesis functions, *i.e.*,

$$\mathscr{F} = \{\boldsymbol{h} = (h_1, ..., h_n, ...) : \forall h_n \in \mathcal{H}, n = 1, ...., +\infty\}.$$

For each $\boldsymbol{h} \in \mathscr{F}$, there is a corresponding algorithm $\mathbf{A}_{\boldsymbol{h}}$: $\mathbf{A}_{\boldsymbol{h}}(S) = h_n$, if $|S| = n$. $\mathscr{F}$ generates an algorithm class $\mathscr{A} = \{\mathbf{A}_{\boldsymbol{h}} : \forall \boldsymbol{h} \in \mathscr{F}\}$. We select a consistent algorithm from the algorithm class $\mathscr{A}$.

We construct a special infinite sequence $\tilde{\boldsymbol{h}} = (\tilde{h}_1, ..., \tilde{h}_n, ...) \in \mathscr{F}$. For each positive integer $n$, we select $\tilde{h}_n$ from

$$\bigcap_{\forall D_{XY} \in [D'_{XY}]} \{h' \in \mathcal{H} : R^{\mathrm{out}}_D(h') \le \inf_{h \in \mathcal{H}} R^{\mathrm{out}}_D(h) + 2/n\} \bigcap \{h' \in \mathcal{H} : R^{\mathrm{in}}_D(h') \le \inf_{h \in \mathcal{H}} R^{\mathrm{in}}_D(h) + 2/n\}.$$

The existence of $\tilde{h}_n$ is based on Condition 3. It is easy to check that for any $D_{XY} \in [D'_{XY}]$,

$$\mathbb{E}_{S \sim D^n_{X_I Y_I}} R^{\mathrm{in}}_D(\mathbf{A}_{\tilde{\boldsymbol{h}}}(S)) \le \inf_{h \in \mathcal{H}} R^{\mathrm{in}}_D(h) + 2/n.$$

$$\mathbb{E}_{S \sim D^n_{X_I Y_I}} R^{\mathrm{out}}_D(\mathbf{A}_{\tilde{\boldsymbol{h}}}(S)) \le \inf_{h \in \mathcal{H}} R^{\mathrm{out}}_D(h) + 2/n.$$

Since $(1-\alpha)\inf_{h \in \mathcal{H}} R^{\mathrm{in}}_D(h) + \alpha \inf_{h \in \mathcal{H}} R^{\mathrm{out}}_D(h) \le \inf_{h \in \mathcal{H}} R^{\alpha}_D(h)$, we obtain that for any $\alpha \in [0, 1]$,

$$\mathbb{E}_{S \sim D^n_{X_I Y_I}} R^{\alpha}_D(\mathbf{A}_{\tilde{\boldsymbol{h}}}(S)) \le \inf_{h \in \mathcal{H}} R^{\alpha}_D(h) + 2/n.$$

Using Theorem 1 (the second result), we have completed this proof. $\square$

**Theorem 8.** *Suppose that $\mathcal{X}$ is a bounded set. OOD detection is learnable in the finite-ID-distribution space $\mathscr{D}_{XY}^F$ for $\mathcal{H}$ **if and only if** the compatibility condition (i.e., Condition 3) holds. Furthermore, the learning rate $\epsilon_{\mathrm{cons}}(n)$ can attain $O(1/\sqrt{n^{1-\theta}})$, for any $\theta \in (0,1)$.*

*Proof of Theorem 8.*

**First**, we prove that if OOD detection is learnable in $\mathscr{D}_{XY}^F$ for $\mathcal{H}$, then Condition 3 holds.

Since $\mathscr{D}_{XY}^F$ is the prior-unknown space, by Theorem 1, there exist an algorithm $\mathbf{A} : \cup_{n=1}^{+\infty} (\mathcal{X} \times \mathcal{Y})^n \to \mathcal{H}$ and a monotonically decreasing sequence $\epsilon_{\mathrm{cons}}(n)$, such that $\epsilon_{\mathrm{cons}}(n) \to 0$, as $n \to +\infty$, and for any $D_{XY} \in \mathscr{D}_{XY}^F$,

$$\mathbb{E}_{S \sim D_{X_I Y_I}^n} \big[ R_D^{\mathrm{in}}(\mathbf{A}(S)) - \inf_{h \in \mathcal{H}} R_D^{\mathrm{in}}(h) \big] \le \epsilon_{\mathrm{cons}}(n),$$

$$\mathbb{E}_{S \sim D_{X_I Y_I}^n} \big[ R_D^{\mathrm{out}}(\mathbf{A}(S)) - \inf_{h \in \mathcal{H}} R_D^{\mathrm{out}}(h) \big] \le \epsilon_{\mathrm{cons}}(n).$$

Then, for any $\epsilon > 0$, we can find $n_\epsilon$ such that $\epsilon \ge \epsilon_{\mathrm{cons}}(n_\epsilon)$, therefore, if $n = n_\epsilon$, we have

$$\mathbb{E}_{S \sim D_{X_I Y_I}^{n_\epsilon}} \big[ R_D^{\mathrm{in}}(\mathbf{A}(S)) - \inf_{h \in \mathcal{H}} R_D^{\mathrm{in}}(h) \big] \le \epsilon,$$

$$\mathbb{E}_{S \sim D_{X_I Y_I}^{n_\epsilon}} \big[ R_D^{\mathrm{out}}(\mathbf{A}(S)) - \inf_{h \in \mathcal{H}} R_D^{\mathrm{out}}(h) \big] \le \epsilon,$$

which implies that there exists $S_\epsilon \sim D_{X_I Y_I}^{n_\epsilon}$ such that

$$R_D^{\mathrm{in}}(\mathbf{A}(S_\epsilon)) - \inf_{h \in \mathcal{H}} R_D^{\mathrm{in}}(h) \le \epsilon,$$

$$R_D^{\mathrm{out}}(\mathbf{A}(S_\epsilon)) - \inf_{h \in \mathcal{H}} R_D^{\mathrm{out}}(h) \le \epsilon.$$

Therefore, for any equivalence class $[D'_{XY}]$ with respect to $\mathscr{D}_{XY}^F$ and any $\epsilon > 0$, there exists a hypothesis function $\mathbf{A}(S_\epsilon) \in \mathcal{H}$ such that for any domain $D_{XY} \in [D'_{XY}]$,

$$\mathbf{A}(S_\epsilon) \in \{h' \in \mathcal{H} : R_D^{\mathrm{out}}(h') \le \inf_{h \in \mathcal{H}} R_D^{\mathrm{out}}(h) + \epsilon\} \cap \{h' \in \mathcal{H} : R_D^{\mathrm{in}}(h') \le \inf_{h \in \mathcal{H}} R_D^{\mathrm{in}}(h) + \epsilon\},$$

which implies that Condition 3 holds.

**Second**, we prove Condition 3 implies the learnability of OOD detection in $\mathscr{D}_{XY}^F$ for $\mathcal{H}$.

For convenience, we assume that all equivalence classes are $[D_{XY}^1], ..., [D_{XY}^m]$. By Lemma 8, for every equivalence class $[D_{XY}^i]$, we can find a corresponding algorithm $\mathbf{A}_{D^i}$ such that OOD detection is learnable in $[D_{XY}^i]$ for $\mathcal{H}$. Additionally, we also set the learning rate for $\mathbf{A}_{D^i}$ is $\epsilon^i(n)$. By Lemma 8, we know that $\epsilon^i(n)$ can attain $O(1/n)$.

Let $\mathcal{Z}$ be $\mathcal{X} \times \mathcal{Y}$. Then, we consider a bounded universal kernel $K(\cdot, \cdot)$ defined over $\mathcal{Z} \times \mathcal{Z}$. Consider the *maximum mean discrepancy* (MMD) [83], which is a metric between distributions: for any distributions $P$ and $Q$ defined over $\mathcal{Z}$, we use $\mathrm{MMD}_K(Q, P)$ to represent the distance.

Let $\mathscr{F}$ be a set consisting of all finite sequences, whose coordinates are labeled data, *i.e.*,

$$\mathscr{F} = \{\mathbf{S} = (S_1, ..., S_i, ..., S_m) : \forall i = 1, ..., m \text{ and } \forall \text{ labeled data } S_i\}.$$

Then, we define an algorithm space as follows:

$$\mathscr{A} = \{\mathbf{A_S}^7 : \forall \mathbf{S} \in \mathscr{F}\},$$

where

$$\mathbf{A_S}(S) = \mathbf{A}_{D^i}(S), \text{ if } i = \underset{i \in \{1, ...m\}}{\arg\min} \mathrm{MMD}_K(P_{S_i}, P_S),$$

here

$$P_S = \frac{1}{n} \sum_{(\mathbf{x}, y) \in S} \delta_{(\mathbf{x}, y)}, \quad P_{S_i} = \frac{1}{n} \sum_{(\mathbf{x}, y) \in S_i}, \delta_{(\mathbf{x}, y)}$$

---

[7]In this paper, we regard an algorithm as a mapping from $\cup_{n=1}^{+\infty} (\mathcal{X} \times \mathcal{Y})^n$ to $\mathcal{H}$. So we can design an algorithm like this.

and $\delta_{(\mathbf{x},y)}$ is the Dirac measure. Next, we prove that we can find an algorithm $\mathbf{A}$ from the algorithm space $\mathscr{A}$ such that $\mathbf{A}$ is the consistent algorithm.

Since the number of different equivalence classes is finite, we know that there exists a constant $c > 0$ such that for any different equivalence classes $[D_{XY}^i]$ and $[D_{XY}^j]$ $(i \neq j)$,

$$\mathrm{MMD}_K(D_{X_I Y_I}^i, D_{X_I Y_I}^j) > c.$$

Additionally, according to [83] and the property of $\mathscr{D}_{XY}^F$ (the number of different equivalence classes is finite), there exists a monotonically decreasing $\epsilon(n) \to 0$, as $n \to +\infty$ such that for any $D_{XY} \in \mathscr{D}$,

$$\mathbb{E}_{S \sim D_{X_I Y_I}^n} \mathrm{MMD}_K(D_{X_I Y_I}, P_S) \leq \epsilon(n), \text{ where } \epsilon(n) = O(\frac{1}{\sqrt{n^{1-\theta}}}). \tag{42}$$

Therefore, for every equivalence class $[D_{XY}^i]$, we can find data points $S_{D^i}$ such that

$$\mathrm{MMD}_K(D_{X_I Y_I}^i, P_{S_{D^i}}) < \frac{c}{100}.$$

Let $\mathbf{S}' = \{S_{D^1}, ..., S_{D^i}, ..., S_{D^m}\}$. Then, we prove that $\mathbf{A}_{\mathbf{S}'}$ is a consistent algorithm. By Eq. (42), it is easy to check that for any $i \in \{1, ..., m\}$ and any $0 < \delta < 1$,

$$\mathbb{P}_{S \sim D_{X_I Y_I}^{i,n}} \left[ \mathrm{MMD}_K(D_{X_I Y_I}^i, P_S) \leq \frac{\epsilon(n)}{\delta} \right] > 1 - \delta,$$

which implies that

$$\mathbb{P}_{S \sim D_{X_I Y_I}^{i,n}} \left[ \mathrm{MMD}_K(P_{S_{D^i}}, P_S) \leq \frac{\epsilon(n)}{\delta} + \frac{c}{100} \right] > 1 - \delta.$$

Therefore, (here we set $\delta = 200\epsilon(n)/c$)

$$\mathbb{P}_{S \sim D_{X_I Y_I}^{i,n}} \left[ \mathbf{A}_{\mathbf{S}'}(S) \neq \mathbf{A}_{D^i}(S) \right] \leq \frac{200\epsilon(n)}{c}.$$

Because $\mathbf{A}_{D^i}$ is a consistent algorithm for $[D_{XY}^i]$, we conclude that for all $\alpha \in [0, 1]$,

$$\mathbb{E}_{S \sim D_{X_I Y_I}^{i,n}} \left[ R_D^\alpha(\mathbf{A}_{\mathbf{S}'}(S)) - \inf_{h \in \mathcal{H}} R_D^\alpha(h) \right] \leq \epsilon^i(n) + \frac{200 B \epsilon(n)}{c},$$

where $\epsilon^i(n) = O(1/n)$ is the learning rate of $\mathbf{A}_{D^i}$ and $B$ is the upper bound of the loss $\ell$.

Let $\epsilon^{\max}(n) = \max\{\epsilon^1(n), ..., \epsilon^m(n)\} + \frac{200 B \epsilon(n)}{c}$.

Then, we obtain that for any $D_{XY} \in \mathscr{D}_{XY}^F$ and all $\alpha \in [0, 1]$,

$$\mathbb{E}_{S \sim D_{X_I Y_I}^n} \left[ R_D^\alpha(\mathbf{A}_{\mathbf{S}'}(S)) - \inf_{h \in \mathcal{H}} R_D^\alpha(h) \right] \leq \epsilon^{\max}(n) = O(\frac{1}{\sqrt{n^{1-\theta}}}).$$

According to Theorem 1 (the second result), $\mathbf{A}_{\mathbf{S}'}$ is the consistent algorithm. This proof is completed. $\square$

## J.2  Proof of Theorem 9

**Theorem 9.** *Given a density-based space $\mathscr{D}_{XY}^{\mu,b}$, if $\mu(\mathcal{X}) < +\infty$, the Realizability Assumption holds, then when $\mathcal{H}$ has finite Natarajan dimension [21], OOD detection is learnable in $\mathscr{D}_{XY}^{\mu,b}$ for $\mathcal{H}$. Furthermore, the learning rate $\epsilon_{\mathrm{cons}}(n)$ can attain $O(1/\sqrt{n^{1-\theta}})$, for any $\theta \in (0, 1)$.*

*Proof of Theorem 9.* **First**, we consider the case that the loss $\ell$ is the zero-one loss.

Since $\mu(\mathcal{X}) < +\infty$, without loss of generality, we assume that $\mu(\mathcal{X}) = 1$. We also assume that $f_I$ is $D_{X_I}$'s density function and $f_O$ is $D_{X_O}$'s density function. Let $f$ be the density function for $0.5 * D_{X_I} + 0.5 * D_{X_O}$. It is easy to check that $f = 0.5 * f_I + 0.5 * f_O$. Additionally, due to

Realizability Assumption, it is obvious that for any samples $S = \{(\mathbf{x}_1, y_1), ..., (\mathbf{x}_n, y_n)\} \sim D^n_{X_I Y_I}$, i.i.d., we have that there exists $h^* \in \mathcal{H}$ such that

$$\frac{1}{n} \sum_{i=1}^{n} \ell(h^*(\mathbf{x}_i), y_i) = 0.$$

Given $m$ data points $S_m = \{\mathbf{x}'_1, ..., \mathbf{x}'_m\} \subset \mathcal{X}^m$. We consider the following learning rule:

$$\min_{h \in \mathcal{H}} \frac{1}{m} \sum_{j=1}^{m} \ell(h(\mathbf{x}'_j), K+1), \ \text{ subject to } \frac{1}{n} \sum_{i=1}^{n} \ell(h(\mathbf{x}_i), y_i) = 0.$$

We denote the algorithm, which solves the above rule, as $\mathbf{A}_{S_m}$[8]. For different data points $S_m$, we have different algorithm $\mathbf{A}_{S_m}$. Let $\mathcal{S}$ be the infinite sequence set that consists of all infinite sequences, whose coordinates are data points, *i.e.*,

$$\mathcal{S} := \{\mathbf{S} := (S_1, S_2, ..., S_m, ...) : S_m \text{ are any } m \text{ data points, } m = 1, ..., +\infty\}. \tag{43}$$

Using $\mathcal{S}$, we construct an algorithm space as follows:

$$\mathscr{A} := \{\mathbf{A}_{\mathbf{S}} : \forall \, \mathbf{S} \in \mathcal{S}\}, \text{ where } \mathbf{A}_{\mathbf{S}}(S) = \mathbf{A}_{S_n}(S), \text{ if } |S| = n.$$

Next, we prove that there exists an algorithm $\mathbf{A}_{\mathbf{S}} \in \mathscr{A}$, which is a consistent algorithm. Given data points $S_n \sim \mu^n$, i.i.d., using the Natarajan dimension theory and Empirical risk minimization principle [21], it is easy to obtain that there exists a uniform constant $C_\theta$ such that (we mainly use the uniform bounds to obtain the following bounds)

$$\mathbb{E}_{S \sim D^n_{X_I Y_I}} \sup_{h \in \mathcal{H}_S} R^{in}_D(h) \leq \inf_{h \in \mathcal{H}} R^{in}_D(h) + \frac{C_\theta}{\sqrt{n^{1-\theta}}},$$

and

$$\mathbb{E}_{S_n \sim \mu^n} R_\mu(\mathbf{A}_{S_n}(S), K+1) \leq \inf_{h \in \mathcal{H}_S} R_\mu(h, K+1) + \frac{C_\theta}{\sqrt{n^{1-\theta}}}, \tag{44}$$

where

$$\mathcal{H}_S = \{h \in \mathcal{H} : \sum_{i=1}^{n} \ell(h(\mathbf{x}_i), y_i) = 0\}, \text{ here } S = \{(\mathbf{x}_1, y_1), ..., (\mathbf{x}_n, y_n)\} \sim D^n_{X_I Y_I},$$

and

$$R_\mu(h, K+1) = \mathbb{E}_{\mathbf{x} \sim \mu} \ell(h(\mathbf{x}), K+1) = \int_{\mathcal{X}} \ell(h(\mathbf{x}), K+1) \mathrm{d}\mu(\mathbf{x}).$$

Due to Realizability Assumption, we obtain that $\inf_{h \in \mathcal{H}} R^{in}_D(h) = 0$. Therefore,

$$\mathbb{E}_{S \sim D^n_{X_I Y_I}} \sup_{h \in \mathcal{H}_S} R^{in}_D(h) \leq \frac{C_\theta}{\sqrt{n^{1-\theta}}}, \tag{45}$$

which implies that (in following inequalities, $g$ is the groundtruth labeling function, *i.e.*, $R_D(g) = 0$)

$$\frac{C_\theta}{\sqrt{n}} \geq \mathbb{E}_{S \sim D^n_{X_I Y_I}} \sup_{h \in \mathcal{H}_S} R^{in}_D(h) = \mathbb{E}_{S \sim D^n_{X_I Y_I}} \sup_{h \in \mathcal{H}_S} \int_{g < K+1} \ell(h(\mathbf{x}), g(\mathbf{x})) f_I(\mathbf{x}) \mathrm{d}\mu(\mathbf{x})$$

$$\geq \frac{2}{b} \mathbb{E}_{S \sim D^n_{X_I Y_I}} \sup_{h \in \mathcal{H}_S} \int_{g < K+1} \ell(h(\mathbf{x}), g(\mathbf{x})) \mathrm{d}\mu(\mathbf{x}).$$

This implies that (here we have used the property of zero-one loss)

$$\mathbb{E}_{S \sim D^n_{X_I Y_I}} \inf_{h \in \mathcal{H}_S} \int_{g < K+1} \ell(h(\mathbf{x}), K+1) \mathrm{d}\mu(\mathbf{x}) \geq \mu(\mathbf{x} \in \mathcal{X} : g(\mathbf{x}) < K+1) - \frac{C_\theta b}{2\sqrt{n^{1-\theta}}}.$$

---

[8]In this paper, we regard an algorithm as a mapping from $\cup_{n=1}^{+\infty} (\mathcal{X} \times \mathcal{Y})^n$ to $\mathcal{H}$. So we can design an algorithm like this.

Therefore,

$$\mathbb{E}_{S \sim D^n_{X_I Y_I}} \inf_{h \in \mathcal{H}_S} R_\mu(h, K+1) \geq \mu(\mathbf{x} \in \mathcal{X} : g(\mathbf{x}) < K+1) - \frac{C_\theta b}{2\sqrt{n^{1-\theta}}}. \tag{46}$$

Additionally, $R_\mu(g, K+1) = \mu(\mathbf{x} \in \mathcal{X} : g(\mathbf{x}) < K+1)$ and $g \in \mathcal{H}_S$, which implies that

$$\inf_{h \in \mathcal{H}_S} R_\mu(h, K+1) \leq \mu(\mathbf{x} \in \mathcal{X} : g(\mathbf{x}) < K+1). \tag{47}$$

Combining inequalities (46) and (47), we obtain that

$$\left| \mathbb{E}_{S \sim D^n_{X_I Y_I}} \inf_{h \in \mathcal{H}_S} R_\mu(h, K+1) - \mu(\mathbf{x} \in \mathcal{X} : g(\mathbf{x}) < K+1) \right| \leq \frac{C_\theta b}{2\sqrt{n^{1-\theta}}}. \tag{48}$$

Using inequalities (44) and (48), we obtain that

$$\left| \mathbb{E}_{S \sim D^n_{X_I Y_I}} \mathbb{E}_{S_n \sim \mu^n} R_\mu(\mathbf{A}_{S_n}(S), K+1) - \mu(\mathbf{x} \in \mathcal{X} : g(\mathbf{x}) < K+1) \right| \leq \frac{C_\theta (b+1)}{\sqrt{n^{1-\theta}}}.$$

By Fubini theorem, we have that

$$\left| \mathbb{E}_{S_n \sim \mu^n} \mathbb{E}_{S \sim D^n_{X_I Y_I}} R_\mu(\mathbf{A}_{S_n}(S), K+1) - \mu(\mathbf{x} \in \mathcal{X} : g(\mathbf{x}) < K+1) \right| \leq \frac{C_\theta (b+1)}{\sqrt{n^{1-\theta}}}. \tag{49}$$

Using inequality (45), we have

$$\mathbb{E}_{S_n \sim \mu^n} \mathbb{E}_{S \sim D^n_{X_I Y_I}} R_D^{\text{in}}(\mathbf{A}_{S_n}(S)) \leq \frac{C_\theta}{\sqrt{n^{1-\theta}}}, \tag{50}$$

which implies that (here we use the property of zero-one loss)

$$\left| \mathbb{E}_{S_n \sim \mu^n} \mathbb{E}_{S \sim D^n_{X_I Y_I}} \int_{g < K+1} \ell(\mathbf{A}_{S_n}(S)(\mathbf{x}), K+1) \mathrm{d}\mu(\mathbf{x}) - \mu(\mathbf{x} \in \mathcal{X} : g(\mathbf{x}) < K+1) \right| \leq \frac{2bC_\theta}{\sqrt{n^{1-\theta}}}. \tag{51}$$

Combining inequalities (49) and (51), we have

$$\left| \mathbb{E}_{S_n \sim \mu^n} \mathbb{E}_{S \sim D^n_{X_I Y_I}} \int_{g = K+1} \ell(\mathbf{A}_{S_n}(S)(\mathbf{x}), K+1) \mathrm{d}\mu(\mathbf{x}) \right| \leq \frac{2bC_\theta}{\sqrt{n^{1-\theta}}} + \frac{C_\theta (b+1)}{\sqrt{n^{1-\theta}}}.$$

Therefore, there exist data points $S'_n$ such that

$$\begin{aligned} &\mathbb{E}_{S \sim D^n_{X_I Y_I}} R_D^{\text{out}}(\mathbf{A}_{S'_n}) \\ =& \mathbb{E}_{S \sim D^n_{X_I Y_I}} \int_{g = K+1} \ell(\mathbf{A}_{S_n}(S)(\mathbf{x}), K+1) f_O(\mathbf{x}) \mathrm{d}\mu(\mathbf{x}) \\ \leq& 2b \mathbb{E}_{S \sim D^n_{X_I Y_I}} \int_{g = K+1} \ell(\mathbf{A}_{S_n}(S)(\mathbf{x}), K+1) \mathrm{d}\mu(\mathbf{x}) \leq \frac{4b^2 C_\theta}{\sqrt{n^{1-\theta}}} + \frac{2C_\theta (b^2 + b)}{\sqrt{n^{1-\theta}}}. \end{aligned} \tag{52}$$

Combining inequalities (45) and (52), we obtain that for any $n$, there exists data points $S'_n$ such that

$$\mathbb{E}_{S \sim D^n_{X_I Y_I}} R_D^\alpha(\mathbf{A}_{S'_n}) \leq \max \left\{ \frac{4b^2 C_\theta}{\sqrt{n^{1-\theta}}} + \frac{2C_\theta (b^2 + b)}{\sqrt{n^{1-\theta}}}, \frac{C_\theta}{\sqrt{n^{1-\theta}}} \right\}.$$

We set data point sequences $\mathbf{S}' = (S'_1, S'_2, ..., S'_n, ...)$. Then, $\mathbf{A}_{\mathbf{S}'} \in \mathscr{A}$ is the universally consistent algorithm, i.e., for any $\alpha \in [0, 1]$

$$\mathbb{E}_{S \sim D^n_{X_I Y_I}} R_D^\alpha(\mathbf{A}_{\mathbf{S}}) \leq \max \left\{ \frac{4b^2 C_\theta}{\sqrt{n^{1-\theta}}} + \frac{2C_\theta (b^2 + b)}{\sqrt{n^{1-\theta}}}, \frac{C_\theta}{\sqrt{n^{1-\theta}}} \right\}.$$

We have completed this proof when $\ell$ is the zero-one loss.

**Second**, we prove the case that $\ell$ is not the zero-one loss. We use the notation $\ell_{0-1}$ as the zero-one loss. According the definition of loss introduced in Section 2, we know that there exists a constant $M > 0$ such that for any $y_1, y_2 \in \mathcal{Y}_{\text{all}}$,

$$\frac{1}{M}\ell_{0-1}(y_1, y_2) \leq \ell(y_1, y_2) \leq M\ell_{0-1}(y_1, y_2).$$

Hence,

$$\frac{1}{M}R_D^{\alpha,\ell_{0-1}}(h) \leq R_D^{\alpha,\ell}(h) \leq MR_D^{\alpha,\ell_{0-1}}(h),$$

where $R_D^{\alpha,\ell_{0-1}}$ is the $\alpha$-risk with zero-one loss, and $R_D^{\alpha,\ell}$ is the $\alpha$-risk for loss $\ell$.

Above inequality tells us that Realizability Assumption holds with zero-one loss if and only if Realizability Assumption holds with the loss $\ell$. Therefore, we use the result proven in first step. We can find a consistent algorithm **A** such that for any $\alpha \in [0, 1]$,

$$\mathbb{E}_{S \sim D_{X_I Y_I}^n} R_D^{\alpha,\ell_{0-1}}(\mathbf{A}) \leq O(\frac{1}{\sqrt{n^{1-\theta}}}),$$

which implies that for any $\alpha \in [0, 1]$,

$$\frac{1}{M}\mathbb{E}_{S \sim D_{X_I Y_I}^n} R_D^{\alpha,\ell}(\mathbf{A}) \leq O(\frac{1}{\sqrt{n^{1-\theta}}}).$$

We have completed this proof. $\qquad\qquad\qquad\qquad\qquad\qquad\qquad\qquad\qquad\qquad\qquad$ □

# K   Proof of Proposition 1 and Proof of Proposition 2

To better understand the contents in Appendices K-M, we introduce the important notations for FCNN-based hypothesis space and score-based hypothesis space detaily.

**FCNN-based Hypothesis Space.** Given a sequence $\mathbf{q} = (l_1, l_2, ..., l_g)$, where $l_i$ and $g$ are positive integers and $g > 2$, we use $g$ to represent the depth of neural network and use $l_i$ to represent the width of the $i$-th layer. After the activation function $\sigma$ is selected, we can obtain the architecture of FCNN according to the sequence $\mathbf{q}$. Given any weights $\mathbf{w}_i \in \mathbb{R}^{l_i \times l_{i-1}}$ and bias $\mathbf{b}_i \in \mathbb{R}^{l_i \times 1}$, the output of the $i$-layer can be written as follows: for any $\mathbf{x} \in \mathbb{R}^{l_1}$,

$$\mathbf{f}_i(\mathbf{x}) = \sigma(\mathbf{w}_i \mathbf{f}_{i-1}(\mathbf{x}) + \mathbf{b}_i), \; \forall i = 2, ..., g-1,$$

where $\mathbf{f}_{i-1}(\mathbf{x})$ is the $i$-th layer output and $\mathbf{f}_1(\mathbf{x}) = \mathbf{x}$. Then, the output of FCNN is $\mathbf{f}_{\mathbf{w},\mathbf{b}}(\mathbf{x}) = \mathbf{w}_g \mathbf{f}_{g-1}(\mathbf{x}) + \mathbf{b}_g$, where $\mathbf{w} = \{\mathbf{w}_2, ..., \mathbf{w}_g\}$ and $\mathbf{b} = \{\mathbf{b}_2, ..., \mathbf{b}_g\}$.

An FCNN-based scoring function space is defined as:

$$\mathcal{F}_{\mathbf{q}}^\sigma := \{\mathbf{f}_{\mathbf{w},\mathbf{b}} : \forall \mathbf{w}_i \in \mathbb{R}^{l_i \times l_{i-1}}, \; \forall \mathbf{b}_i \in \mathbb{R}^{l_i \times 1}, \; i = 2, ..., g\}.$$

Additionally, given two sequences $\mathbf{q} = (l_1, ..., l_g)$ and $\mathbf{q}' = (l_1', ..., l_{g'}')$, we use the notation $\mathbf{q} \lesssim \mathbf{q}'$ to represent the following equations and inequalities:

$$g \leq g', \quad l_1 = l_1', \quad l_g = l_{g'}',$$
$$l_i \leq l_i', \quad \forall i = 1, ..., g-1,$$
$$l_{g-1} \leq l_i', \quad \forall i = g, ..., g'-1.$$

Given a sequence $\mathbf{q} = (l_1, ...l_g)$ satisfying that $l_1 = d$ and $l_g = K + 1$, the FCNN-based scoring function space $\mathcal{F}_{\mathbf{q}}^\sigma$ can induce an FCNN-based hypothesis space. Before defining the FCNN-based hypothesis space, we define the induced hypothesis function. For any $\mathbf{f}_{\mathbf{w},\mathbf{b}} \in \mathcal{F}_{\mathbf{q}}^\sigma$, the induced hypothesis function is:

$$h_{\mathbf{w},\mathbf{b}}(\mathbf{x}) := \underset{k \in \{1, ..., K+1\}}{\arg\max} f_{\mathbf{w},\mathbf{b}}^k(\mathbf{x}), \; \forall \mathbf{x} \in \mathcal{X},$$

where $f_{\mathbf{w},\mathbf{b}}^k(\mathbf{x})$ is the $k$-th coordinate of $\mathbf{f}_{\mathbf{w},\mathbf{b}}(\mathbf{x})$. Then, we define the FCNN-based hypothesis space as follows:

$$\mathcal{H}_{\mathbf{q}}^\sigma := \{h_{\mathbf{w},\mathbf{b}} : \forall \mathbf{w}_i \in \mathbb{R}^{l_i \times l_{i-1}}, \; \forall \mathbf{b}_i \in \mathbb{R}^{l_i \times 1}, \; i = 2, ..., g\}.$$

**Score-based Hypothesis Space.** Many OOD algorithms detect OOD data using a score-based strategy. That is, given a threshold $\lambda$, a scoring function space $\mathcal{F}_l \subset \{\mathbf{f} : \mathcal{X} \to \mathbb{R}^l\}$ and a scoring function $E : \mathcal{F}_l \to \mathbb{R}$, then $\mathbf{x}$ is regarded as ID, if $E(\mathbf{f}(\mathbf{x})) \geq \lambda$; otherwise, $\mathbf{x}$ is regarded as OOD.

Using $E$, $\lambda$ and $\mathbf{f} \in \mathcal{F}_\mathbf{q}^\sigma$, we can generate a binary classifier $h_{\mathbf{f},E}^\lambda$:

$$h_{\mathbf{f},E}^\lambda(\mathbf{x}) := \left\{ \begin{array}{ll} 1, & \text{if } E(\mathbf{f}(\mathbf{x})) \geq \lambda; \\ 2, & \text{if } E(\mathbf{f}(\mathbf{x})) < \lambda, \end{array} \right.$$

where 1 represents ID data, and 2 represents OOD data. Hence, a binary classification hypothesis space $\mathcal{H}^b$, which consists of all $h_{\mathbf{f},E}^\lambda$, is generated. We define the score-based hypothesis space $\mathcal{H}_{\mathbf{q},E}^{\sigma,\lambda} := \{h_{\mathbf{f},E}^\lambda : \forall \mathbf{f} \in \mathcal{F}_\mathbf{q}^\sigma\}$.

Next, we introduce two important propositions.

**Proposition 1.** *Given a sequence $\mathbf{q} = (l_1, ... l_g)$ satisfying that $l_1 = d$ and $l_g = K + 1$ (note that $d$ is the dimension of input data and $K + 1$ is the dimension of output), then the constant functions $h_1$, $h_2,...,h_{K+1}$ belong to $\mathcal{H}_\mathbf{q}^\sigma$, where $h_i(\mathbf{x}) = i$, for any $\mathbf{x} \in \mathcal{X}$. Therefore, Assumption 1 holds for $\mathcal{H}_\mathbf{q}^\sigma$.*

*Proof of Proposition 1.* Note that the output of FCNN can be written as

$$\mathbf{f}_{\mathbf{w},\mathbf{b}}(\mathbf{x}) = \mathbf{w}_g \mathbf{f}_{g-1}(\mathbf{x}) + \mathbf{b}_g,$$

where $\mathbf{w}_g \in \mathbb{R}^{(K+1)\times l_{g-1}}$, $\mathbf{b}_g \in \mathbb{R}^{(K+1)\times 1}$ and $\mathbf{f}_{g-1}(\mathbf{x})$ is the output of the $l_{g-1}$-th layer. If we set $\mathbf{w}_g = \mathbf{0}$, and set $\mathbf{b}_g = \mathbf{y}_i$, where $\mathbf{y}_i$ is the one-hot vector corresponding to label $i$. Then $\mathbf{f}_{\mathbf{w},\mathbf{b}}(\mathbf{x}) = \mathbf{y}_i$, for any $\mathbf{x} \in \mathcal{X}$. Therefore, $h_i(\mathbf{x}) \in \mathcal{H}_\mathbf{q}^\sigma$, for any $i = 1, ..., K, K + 1$. $\quad\square$

Note that in some works [84], $\mathbf{b}_g$ is fixed to $\mathbf{0}$. In fact, it is easy to check that when $g > 2$ and activation function $\sigma$ is not a constant, Proposition 1 still holds, even if $\mathbf{b}_g = \mathbf{0}$.

**Proposition 2.** *For any sequence $\mathbf{q} = (l_1, ..., l_g)$ satisfying that $l_1 = d$ and $l_g = l$ (note that $d$ is the dimension of input data and $l$ is the dimension of output), if $\{\mathbf{v} \in \mathbb{R}^l : E(\mathbf{v}) \geq \lambda\} \neq \emptyset$ and $\{\mathbf{v} \in \mathbb{R}^l : E(\mathbf{v}) < \lambda\} \neq \emptyset$, then the functions $h_1$ and $h_2$ belong to $\mathcal{H}_{\mathbf{q},E}^{\sigma,\lambda}$, where $h_1(\mathbf{x}) = 1$ and $h_2(\mathbf{x}) = 2$, for any $\mathbf{x} \in \mathcal{X}$, where 1 represents the ID labels, and 2 represents the OOD labels. Therefore, Assumption 1 holds.*

*Proof of Proposition 2.* Since $\{\mathbf{v} \in \mathbb{R}^l : E(\mathbf{v}) \geq \lambda\} \neq \emptyset$ and $\{\mathbf{v} \in \mathbb{R}^l : E(\mathbf{v}) < \lambda\} \neq \emptyset$, we can find $\mathbf{v}_1 \in \{\mathbf{v} \in \mathbb{R}^l : E(\mathbf{v}) \geq \lambda\}$ and $\mathbf{v}_2 \in \{\mathbf{v} \in \mathbb{R}^l : E(\mathbf{v}) < \lambda\}$.

For any $\mathbf{f}_{\mathbf{w},\mathbf{b}} \in \mathcal{F}_\mathbf{q}^\sigma$, we have

$$\mathbf{f}_{\mathbf{w},\mathbf{b}}(\mathbf{x}) = \mathbf{w}_g \mathbf{f}_{g-1}(\mathbf{x}) + \mathbf{b}_g,$$

where $\mathbf{w}_g \in \mathbb{R}^{l\times l_{g-1}}$, $\mathbf{b}_g \in \mathbb{R}^{l\times 1}$ and $\mathbf{f}_{g-1}(\mathbf{x})$ is the output of the $l_{g-1}$-th layer.

If we set $\mathbf{w}_g = \mathbf{0}_{l\times l_{g-1}}$ and $\mathbf{b}_g = \mathbf{v}_1$, then $\mathbf{f}_{\mathbf{w},\mathbf{b}}(\mathbf{x}) = \mathbf{v}_1$ for any $\mathbf{x} \in \mathcal{X}$, where $\mathbf{0}_{l\times l_{g-1}}$ is $l \times l_{g-1}$ zero matrix. Hence, $h_1$ can be induced by $\mathbf{f}_{\mathbf{w},\mathbf{b}}$. Therefore, $h_1 \in \mathcal{H}_{\mathbf{q},E}^{\sigma,\lambda}$.

Similarly, if we set $\mathbf{w}_g = \mathbf{0}_{l\times l_{g-1}}$ and $\mathbf{b}_g = \mathbf{v}_2$, then $\mathbf{f}_{\mathbf{w},\mathbf{b}}(\mathbf{x}) = \mathbf{v}_2$ for any $\mathbf{x} \in \mathcal{X}$, where $\mathbf{0}_{l\times l_{g-1}}$ is $l \times l_{g-1}$ zero matrix. Hence, $h_2$ can be induced by $\mathbf{f}_{\mathbf{w},\mathbf{b}}$. Therefore, $h_2 \in \mathcal{H}_{\mathbf{q},E}^{\sigma,\lambda}$. $\quad\square$

It is easy to check that when $g > 2$ and activation function $\sigma$ is not a constant, Proposition 2 still holds, even if $\mathbf{b}_g = \mathbf{0}$.

## L   Proof of Theorem 10

Before proving Theorem 10, we need several lemmas.

**Lemma 9.** *Let $\sigma$ be ReLU function: $\max\{x, 0\}$. Given $\mathbf{q} = (l_1, ..., l_g)$ and $\mathbf{q}' = (l_1', ..., l_g')$ such that $l_g = l_g'$ and $l_1 = l_1'$, and $l_i \leq l_i'$ ($i = 1, ..., g - 1$), then $\mathcal{F}_\mathbf{q}^\sigma \subset \mathcal{F}_{\mathbf{q}'}^\sigma$ and $\mathcal{H}_\mathbf{q}^\sigma \subset \mathcal{H}_{\mathbf{q}'}^\sigma$.*

*Proof of Lemma 9.* Given any weights $\mathbf{w}_i \in \mathbb{R}^{l_i \times l_{i-1}}$ and bias $\mathbf{b}_i \in \mathbb{R}^{l_i \times 1}$, the $i$-layer output of FCNN with architecture $\mathbf{q}$ can be written as

$$\mathbf{f}_i(\mathbf{x}) = \sigma(\mathbf{w}_i \mathbf{f}_{i-1}(\mathbf{x}) + \mathbf{b}_i), \ \ \forall \mathbf{x} \in \mathbb{R}^{l_1}, \forall i = 2, ..., g-1,$$

where $\mathbf{f}_{i-1}(\mathbf{x})$ is the $i$-th layer output and $\mathbf{f}_1(\mathbf{x}) = \mathbf{x}$. Then, the output of last layer is

$$\mathbf{f}_{\mathbf{w},\mathbf{b}}(\mathbf{x}) = \mathbf{w}_g \mathbf{f}_{g-1}(\mathbf{x}) + \mathbf{b}_g.$$

We will show that $\mathbf{f}_{\mathbf{w},\mathbf{b}} \in \mathcal{F}^\sigma_{\mathbf{q}'}$. We construct $\mathbf{f}_{\mathbf{w}',\mathbf{b}'}$ as follows: for every $\mathbf{w}'_i \in \mathbb{R}^{l'_i \times l'_{i-1}}$, if $l'_i - l_i > 0$ and $l'_{i-1} - l_{i-1} > 0$, we set

$$\mathbf{w}'_i = \begin{bmatrix} \mathbf{w}_i & \mathbf{0}_{l_i \times (l'_{i-1} - l_{i-1})} \\ \mathbf{0}_{(l'_i - l_i) \times l'_{i-1}} & \mathbf{0}_{(l'_i - l_i) \times (l'_{i-1} - l_{i-1})} \end{bmatrix}, \ \ \mathbf{b}'_i = \begin{bmatrix} \mathbf{b}_i \\ \mathbf{0}_{(l'_i - l_i) \times 1} \end{bmatrix}$$

where $\mathbf{0}_{pq}$ means the $p \times q$ zero matrix. If $l'_i - l_i = 0$ and $l'_{i-1} - l_{i-1} > 0$, we set

$$\mathbf{w}'_i = \begin{bmatrix} \mathbf{w}_i & \mathbf{0}_{l_i \times (l'_{i-1} - l_{i-1})} \end{bmatrix}, \ \ \mathbf{b}'_i = \mathbf{b}_i.$$

If $l'_{i-1} - l_{i-1} = 0$ and $l'_i - l_i > 0$, we set

$$\mathbf{w}'_i = \begin{bmatrix} \mathbf{w}_i \\ \mathbf{0}_{(l'_i - l_i) \times l'_{i-1}} \end{bmatrix}, \ \ \mathbf{b}'_i = \begin{bmatrix} \mathbf{b}_i \\ \mathbf{0}_{(l'_i - l_i) \times 1} \end{bmatrix}.$$

If $l'_{i-1} - l_{i-1} = 0$ and $l'_i - l_i = 0$, we set

$$\mathbf{w}'_i = \mathbf{w}_i, \ \ \mathbf{b}'_i = \mathbf{b}_i.$$

It is easy to check that if $l'_i - l_i > 0$

$$\mathbf{f}'_i = \begin{bmatrix} \mathbf{f}_i \\ \mathbf{0}_{(l'_i - l_i) \times 1} \end{bmatrix}.$$

If $l'_i - l_i = 0$,

$$\mathbf{f}'_i = \mathbf{f}_i.$$

Since $l'_g - l_g = 0$,

$$\mathbf{f}'_g = \mathbf{f}_g, \ i.e., \ \mathbf{f}_{\mathbf{w}',\mathbf{b}'} = \mathbf{f}_{\mathbf{w},\mathbf{b}}.$$

Therefore, $f_{\mathbf{w},\mathbf{b}} \in \mathcal{F}^\sigma_{\mathbf{q}'}$, which implies that $\mathcal{F}^\sigma_{\mathbf{q}} \subset \mathcal{F}^\sigma_{\mathbf{q}'}$. Therefore, $\mathcal{H}^\sigma_{\mathbf{q}} \subset \mathcal{H}^\sigma_{\mathbf{q}'}$. $\qquad\square$

**Lemma 10.** *Let $\sigma$ be the ReLU function: $\sigma(x) = \max\{x, 0\}$. Then, $\mathbf{q} \lesssim \mathbf{q}'$ implies that $\mathcal{F}^\sigma_{\mathbf{q}} \subset \mathcal{F}^\sigma_{\mathbf{q}'}$, $\mathcal{H}^\sigma_{\mathbf{q}} \subset \mathcal{H}^\sigma_{\mathbf{q}'}$, where $\mathbf{q} = (l_1, ..., l_g)$ and $\mathbf{q}' = (l'_1, ..., l'_{g'})$.*

*Proof of Lemma 10.* Given $l'' = (l''_1, ..., l''_{g''})$ satisfying that $g \leq g''$, $l''_i = l_i$ for $i = 1, ..., g-1$, $l''_i = l_{g-1}$ for $i = g, ..., g''-1$, and $l''_{g''} = l_g$, we first prove that $\mathcal{F}^\sigma_{\mathbf{q}} \subset \mathcal{F}^\sigma_{\mathbf{q}''}$ and $\mathcal{H}^\sigma_{\mathbf{q}} \subset \mathcal{H}^\sigma_{\mathbf{q}''}$.

Given any weights $\mathbf{w}_i \in \mathbb{R}^{l_i \times l_{i-1}}$ and bias $\mathbf{b}_i \in \mathbb{R}^{l_i \times 1}$, the $i$-th layer output of FCNN with architecture $\mathbf{q}$ can be written as

$$\mathbf{f}_i(\mathbf{x}) = \sigma(\mathbf{w}_i \mathbf{f}_{i-1}(\mathbf{x}) + \mathbf{b}_i), \ \ \forall \mathbf{x} \in \mathbb{R}^{l_1}, \forall i = 2, ..., g-1,$$

where $\mathbf{f}_{i-1}(\mathbf{x})$ is the $i$-th layer output and $\mathbf{f}_1(\mathbf{x}) = \mathbf{x}$. Then, the output of the last layer is

$$\mathbf{f}_{\mathbf{w},\mathbf{b}}(\mathbf{x}) = \mathbf{w}_g \mathbf{f}_{g-1}(\mathbf{x}) + \mathbf{b}_g.$$

We will show that $\mathbf{f}_{\mathbf{w},\mathbf{b}} \in \mathcal{F}^\sigma_{\mathbf{q}''}$. We construct $\mathbf{f}_{\mathbf{w}'',\mathbf{b}''}$ as follows: if $i = 2, ..., g-1$, then $\mathbf{w}''_i = \mathbf{w}$ and $\mathbf{b}''_i = \mathbf{b}_i$; if $i = g, ..., g''-1$, then $\mathbf{w}''_i = \mathbf{I}_{l_{g-1} \times l_{g-1}}$ and $\mathbf{b}''_i = \mathbf{0}_{l_{g-1} \times 1}$, where $\mathbf{I}_{l_{g-1} \times l_{g-1}}$ is the $l_{g-1} \times l_{g-1}$ identity matrix, and $\mathbf{0}_{l_{g-1} \times 1}$ is the $l_{g-1} \times 1$ zero matrix; and if $i = g''$, then $\mathbf{w}''_{g''} = \mathbf{w}_g$, $\mathbf{b}''_{g''} = \mathbf{b}_g$. Then it is easy to check that the output of the $i$-th layer is

$$\mathbf{f}''_i = \mathbf{f}_{g-1}, \forall i = g-1, g, ..., g''-1.$$

Therefore, $\mathbf{f}_{\mathbf{w}'',\mathbf{b}''} = \mathbf{f}_{\mathbf{w},\mathbf{b}}$, which implies that $\mathcal{F}^\sigma_{\mathbf{q}} \subset \mathcal{F}^\sigma_{\mathbf{q}''}$. Hence, $\mathcal{H}^\sigma_{\mathbf{q}} \subset \mathcal{H}^\sigma_{\mathbf{q}''}$.

When $g'' = g'$, we use Lemma 9 ($\mathbf{q}''$ and $\mathbf{q}$ satisfy the condition in Lemma 9), which implies that $\mathcal{F}^\sigma_{\mathbf{q}''} \subset \mathcal{F}^\sigma_{\mathbf{q}'}$, $\mathcal{H}^\sigma_{\mathbf{q}''} \subset \mathcal{H}^\sigma_{\mathbf{q}'}$. Therefore, $\mathcal{F}^\sigma_{\mathbf{q}} \subset \mathcal{F}^\sigma_{\mathbf{q}'}$, $\mathcal{H}^\sigma_{\mathbf{q}} \subset \mathcal{H}^\sigma_{\mathbf{q}'}$. $\qquad\square$

**Lemma 11.** *[85] If the activation function $\sigma$ is not a polynomial, then for any continuous function $f$ defined in $\mathbb{R}^d$, and any compact set $C \subset \mathbb{R}^d$, there exists a fully-connected neural network with architecture $\mathbf{q}$ ($l_1 = d, l_g = 1$) such that*

$$\inf_{f_{\mathbf{w},\mathbf{b}} \in \mathcal{F}_{\mathbf{q}}^{\sigma}} \max_{\mathbf{x} \in C} |f_{\mathbf{w},\mathbf{b}}(\mathbf{x}) - f(\mathbf{x})| < \epsilon.$$

*Proof of Lemma 11.* The proof of Lemma 11 can be found in Theorem 3.1 in [85]. $\qquad\square$

**Lemma 12.** *If the activation function $\sigma$ is the ReLU function, then for any continuous vector-valued function $\mathbf{f} \in C(\mathbb{R}^d; \mathbb{R}^l)$, and any compact set $C \subset \mathbb{R}^d$, there exists a fully-connected neural network with architecture $\mathbf{q}$ ($l_1 = d, l_g = l$) such that*

$$\inf_{\mathbf{f}_{\mathbf{w},\mathbf{b}} \in \mathcal{F}_{\mathbf{q}}^{\sigma}} \max_{\mathbf{x} \in C} \|\mathbf{f}_{\mathbf{w},\mathbf{b}}(\mathbf{x}) - \mathbf{f}(\mathbf{x})\|_2 < \epsilon,$$

*where $\|\cdot\|_2$ is the $\ell_2$ norm. (Note that we can also prove the same result, if $\sigma$ is not a polynomial.)*

*Proof of Lemma 12.* Let $\mathbf{f} = [f_1, ..., f_l]^\top$, where $f_i$ is the $i$-th coordinate of $\mathbf{f}$. Based on Lemma 11, we obtain $l$ sequences $\mathbf{q}^1, \mathbf{q}^2,...,\mathbf{q}^l$ such that

$$\inf_{g_1 \in \mathcal{F}_{\mathbf{q}^1}^{\sigma}} \max_{\mathbf{x} \in C} |g_1(\mathbf{x}) - f_1(\mathbf{x})| < \epsilon/\sqrt{l},$$

$$\inf_{g_2 \in \mathcal{F}_{\mathbf{q}^2}^{\sigma}} \max_{\mathbf{x} \in C} |g_2(\mathbf{x}) - f_2(\mathbf{x})| < \epsilon/\sqrt{l},$$

$$...$$
$$...$$

$$\inf_{g_l \in \mathcal{F}_{\mathbf{q}^l}^{\sigma}} \max_{\mathbf{x} \in C} |g_l(\mathbf{x}) - f_l(\mathbf{x})| < \epsilon/\sqrt{l}.$$

It is easy to find a sequence $\mathbf{q} = (l_1, ..., l_g)$ ($l_g = 1$) such that $\mathbf{q}^i \lesssim \mathbf{q}$, for all $i = 1, ..., l$. Using Lemma 10, we obtain that $\mathcal{F}_{\mathbf{q}^i}^{\sigma} \subset \mathcal{F}_{\mathbf{q}}^{\sigma}$. Therefore,

$$\inf_{g \in \mathcal{F}_{\mathbf{q}}^{\sigma}} \max_{\mathbf{x} \in C} |g(\mathbf{x}) - f_1(\mathbf{x})| < \epsilon/\sqrt{l},$$

$$\inf_{g \in \mathcal{F}_{\mathbf{q}}^{\sigma}} \max_{\mathbf{x} \in C} |g(\mathbf{x}) - f_2(\mathbf{x})| < \epsilon/\sqrt{l},$$

$$...$$
$$...$$

$$\inf_{g \in \mathcal{F}_{\mathbf{q}}^{\sigma}} \max_{\mathbf{x} \in C} |g(\mathbf{x}) - f_l(\mathbf{x})| < \epsilon/\sqrt{l}.$$

Therefore, for each $i$, we can find $g_{\mathbf{w}^i,\mathbf{b}^i}$ from $\mathcal{F}_{\mathbf{q}}^{\sigma}$ such that

$$\max_{\mathbf{x} \in C} |g_{\mathbf{w}^i,\mathbf{b}^i}(\mathbf{x}) - f_i(\mathbf{x})| < \epsilon/\sqrt{l},$$

where $\mathbf{w}^i$ represents weights and $\mathbf{b}^i$ represents bias.

We construct a larger FCNN with $\mathbf{q}' = (l_1', l_2', ..., l_g')$ satisfying that $l_1' = d$, $l_i' = l * l_i$, for $i = 2, ..., g$. We can regard this larger FCNN as a combinations of $l$ FCNNs with architecture $\mathbf{q}$, that is: there are $m$ disjoint sub-FCNNs with architecture $\mathbf{q}$ in the larger FCNN with architecture $\mathbf{q}'$. For $i$-th sub-FCNN, we use weights $\mathbf{w}^i$ and bias $\mathbf{b}^i$. For weights and bias which connect different sub-FCNNs, we set these weights and bias to $\mathbf{0}$. Finally, we can obtain that $\mathbf{g}_{\mathbf{w},\mathbf{b}} = [g_{\mathbf{w}^1,\mathbf{b}^1}, g_{\mathbf{w}^2,\mathbf{b}^2}, ..., g_{\mathbf{w}^l,\mathbf{b}^l}]^\top \in \mathcal{F}_{\mathbf{q}'}^{\sigma}$, which implies that

$$\max_{\mathbf{x} \in C} \|\mathbf{g}_{\mathbf{w},\mathbf{b}}(\mathbf{x}) - \mathbf{f}(\mathbf{x})\|_2 < \epsilon.$$

We have completed this proof. $\qquad\square$

Given a sequence $\mathbf{q} = (l_1, ..., l_g)$, we are interested in following function space $\mathcal{F}^\sigma_{\mathbf{q},\mathbf{M}}$:

$$\mathcal{F}^\sigma_{\mathbf{q},\mathbf{M}} := \{\mathbf{M} \cdot (\sigma \circ \mathbf{f}) : \forall \mathbf{f} \in \mathcal{F}^\sigma_{\mathbf{q}}\},$$

where $\circ$ means the composition of two functions, $\cdot$ means the product of two matrices, and

$$\mathbf{M} = \begin{bmatrix} \mathbf{1}_{1 \times (l_g - 1)} & 0 \\ \mathbf{0}_{1 \times (l_g - 1)} & 1 \end{bmatrix},$$

here $\mathbf{1}_{1 \times (l_g - 1)}$ is the $1 \times (l_g - 1)$ matrix whose all elements are 1, and $\mathbf{0}_{1 \times (l_g - 1)}$ is the $1 \times (l_g - 1)$ zero matrix. Using $\mathcal{F}^\sigma_{\mathbf{q},\mathbf{M}}$, we can construct a binary classification space $\mathcal{H}^\sigma_{\mathbf{q},\mathbf{M}}$, which consists of all classifiers satisfying the following condition:

$$h(\mathbf{x}) = \underset{k = \{1,2\}}{\arg\min} \; f^k_{\mathbf{M}}(\mathbf{x}),$$

where $f^k_{\mathbf{M}}(\mathbf{x})$ is the $k$-th coordinate of $\mathbf{M} \cdot (\sigma \circ \mathbf{f})$.

**Lemma 13.** *Suppose that $\sigma$ is the ReLU function: $\max\{x, 0\}$. Given a sequence $\mathbf{q} = (l_1, ..., l_g)$ satisfying that $l_1 = d$ and $l_g = K + 1$, then the space $\mathcal{H}^\sigma_{\mathbf{q},\mathbf{M}}$ contains $\phi \circ \mathcal{H}^\sigma_{\mathbf{q}}$, and $\mathcal{H}^\sigma_{\mathbf{q},\mathbf{M}}$ has finite VC dimension (Vapnik–Chervonenkis dimension), where $\phi$ maps ID data to $1$ and OOD data to $2$. Furthermore, if given $\mathbf{q}' = (l'_1, ..., l'_g)$ satisfying that $l'_g = K$ and $l'_i = l_i$, for $i = 1, ..., g - 1$, then $\mathcal{H}^\sigma_{\mathbf{q}} \subset \mathcal{H}^\sigma_{\mathbf{q}'} \bullet \mathcal{H}^\sigma_{\mathbf{q},\mathbf{M}}$.*

*Proof of Lemma 13.* For any $h_{\mathbf{w},\mathbf{b}} \in \mathcal{H}^\sigma_{\mathbf{q}}$, then there exists $\mathbf{f}_{\mathbf{w},\mathbf{b}} \in \mathcal{F}^\sigma_{\mathbf{q}}$ such that $h_{\mathbf{w},\mathbf{b}}$ is induced by $\mathbf{f}_{\mathbf{w},\mathbf{b}}$. We can write $\mathbf{f}_{\mathbf{w},\mathbf{b}}$ as follows:

$$\mathbf{f}_{\mathbf{w},\mathbf{b}}(\mathbf{x}) = \mathbf{w}_g \mathbf{f}_{g-1}(\mathbf{x}) + \mathbf{b}_g,$$

where $\mathbf{w}_g \in \mathbb{R}^{(K+1) \times l_{g-1}}$, $\mathbf{b}_g \in \mathbb{R}^{(K+1) \times 1}$ and $\mathbf{f}_{g-1}(\mathbf{x})$ is the output of the $l_{g-1}$-th layer.

Suppose that

$$\mathbf{w}_g = \begin{bmatrix} \mathbf{v}_1 \\ \mathbf{v}_2 \\ ... \\ \mathbf{v}_K \\ \mathbf{v}_{K+1} \end{bmatrix}, \quad \mathbf{b}_g = \begin{bmatrix} b_1 \\ b_2 \\ ... \\ b_K \\ b_{K+1} \end{bmatrix},$$

where $\mathbf{v}_i \in \mathbb{R}^{1 \times l_{g-1}}$ and $b_i \in \mathbb{R}$.

We set

$$\mathbf{f}_{\mathbf{w}',\mathbf{b}'}(\mathbf{x}) = \mathbf{w}'_g \mathbf{f}_{g-1}(\mathbf{x}) + \mathbf{b}'_g,$$

where

$$\mathbf{w}'_g = \begin{bmatrix} \mathbf{v}_1 \\ \mathbf{v}_2 \\ ... \\ \mathbf{v}_K \end{bmatrix}, \quad \mathbf{b}'_g = \begin{bmatrix} b_1 \\ b_2 \\ ... \\ b_K \end{bmatrix},$$

It is obvious that $\mathbf{f}_{\mathbf{w}',\mathbf{b}'} \in \mathcal{F}^\sigma_{\mathbf{q}'}$. Using $\mathbf{f}_{\mathbf{w}',\mathbf{b}'} \in \mathcal{F}^\sigma_{\mathbf{q}'}$, we construct a classifier $h_{\mathbf{w}',\mathbf{b}'} \in \mathcal{H}^\sigma_{\mathbf{q}'}$:

$$h_{\mathbf{w}',\mathbf{b}'} = \underset{k \in \{1,...,K\}}{\arg\max} \; f^k_{\mathbf{w}',\mathbf{b}'},$$

where $f^k_{\mathbf{w}',\mathbf{b}'}$ is the $k$-th coordinate of $\mathbf{f}_{\mathbf{w}',\mathbf{b}'}$.

Additionally, we consider

$$\mathbf{f}_{\mathbf{w},\mathbf{b},\mathbf{B}} = \mathbf{M} \cdot \sigma(\mathbf{B} \cdot \mathbf{f}_{\mathbf{w},\mathbf{b}}) \in \mathcal{F}^\sigma_{\mathbf{q},\mathbf{M}},$$

where

$$\mathbf{B} = \begin{bmatrix} \mathbf{I}_{(l_g - 1) \times (l_g - 1)} & -\mathbf{1}_{(l_g - 1) \times 1} \\ \mathbf{0}_{1 \times (l_g - 1)} & 0 \end{bmatrix},$$

here $\mathbf{I}_{(l_g - 1) \times (l_g - 1)}$ is the $(l_g - 1) \times (l_g - 1)$ identity matrix, $\mathbf{0}_{1 \times (l_g - 1)}$ is the $1 \times (l_g - 1)$ zero matrix, and $\mathbf{1}_{(l_g - 1) \times 1}$ is the $(l_g - 1) \times 1$ matrix, whose all elements are 1.

Then, we define that for any $\mathbf{x} \in \mathcal{X}$,

$$h_{\mathbf{w},\mathbf{b},\mathbf{B}}(\mathbf{x}) := \arg\max_{k \in \{1,2\}} f^k_{\mathbf{w},\mathbf{b},\mathbf{B}}(\mathbf{x}),$$

where $f^k_{\mathbf{w},\mathbf{b},\mathbf{B}}(\mathbf{x})$ is the $k$-th coordinate of $\mathbf{f}_{\mathbf{w},\mathbf{b},\mathbf{B}}(\mathbf{x})$. Furthermore, we can check that $h_{\mathbf{w},\mathbf{b},\mathbf{B}}$ can be written as follows: for any $\mathbf{x} \in \mathcal{X}$,

$$h_{\mathbf{w},\mathbf{b},\mathbf{B}}(\mathbf{x}) = \begin{cases} 1, & \text{if } f^1_{\mathbf{w},\mathbf{b},\mathbf{B}}(\mathbf{x}) > 0; \\ 2, & \text{if } f^1_{\mathbf{w},\mathbf{b},\mathbf{B}}(\mathbf{x}) \leq 0. \end{cases}$$

It is easy to check that

$$h_{\mathbf{w},\mathbf{b},\mathbf{B}} = \phi \circ h_{\mathbf{w},\mathbf{b}},$$

where $\phi$ maps ID labels to 1 and OOD labels to 2.

Therefore, $h_{\mathbf{w},\mathbf{b}}(\mathbf{x}) = K + 1$ if and only if $h_{\mathbf{w},\mathbf{b},\mathbf{B}} = 2$; and $h_{\mathbf{w},\mathbf{b}}(\mathbf{x}) = k$ ($k \neq K + 1$) if and only if $h_{\mathbf{w},\mathbf{b},\mathbf{B}} = 1$ and $h_{\mathbf{w}',\mathbf{b}'}(\mathbf{x}) = k$. This implies that $\mathcal{H}^\sigma_{\mathbf{q}} \subset \mathcal{H}^\sigma_{\mathbf{q}'} \bullet \mathcal{H}^\sigma_{\mathbf{q},\mathbf{M}}$ and $\phi \circ \mathcal{H}^\sigma_{\mathbf{q}} \subset \mathcal{H}^\sigma_{\mathbf{q},\mathbf{M}}$.

Let $\tilde{\mathbf{q}}$ be $(l_1, ..., l_g, 2)$. Then $\mathcal{F}^\sigma_{\mathbf{q},\mathbf{M}} \subset \mathcal{F}^\sigma_{\tilde{\mathbf{q}}}$. Hence, $\mathcal{H}^\sigma_{\mathbf{q},\mathbf{M}} \subset \mathcal{H}^\sigma_{\tilde{\mathbf{q}}}$. According to the VC dimension theory [37] for feed-forward neural networks, $\mathcal{H}^\sigma_{\tilde{\mathbf{q}}}$ has finite VC dimension. Hence, $\mathcal{H}^\sigma_{\mathbf{q},\mathbf{M}}$ has finite VC-dimension. We have completed the proof. $\qquad\square$

**Lemma 14.** *Let $|\mathcal{X}| < +\infty$ and $\sigma$ be the ReLU function: $\max\{x, 0\}$. Given $r$ hypothesis functions $h_1, h_2, ..., h_r \in \{h : \mathcal{X} \to \{1, ..., l\}\}$, then there exists a sequence $\mathbf{q} = (l_1, ..., l_g)$ with $l_1 = d$ and $l_g = l$, such that $h_1, ..., h_r \in \mathcal{H}^\sigma_{\mathbf{q}}$.*

*Proof of Lemma 14.* For each $h_i$ ($i = 1, ..., r$), we introduce a corresponding $\mathbf{f}_i$ (defined over $\mathcal{X}$) satisfying that for any $\mathbf{x} \in \mathcal{X}$, $\mathbf{f}_i(\mathbf{x}) = \mathbf{y}_k$ if and only if $h_i(\mathbf{x}) = k$, where $\mathbf{y}_k \in \mathbb{R}^l$ is the one-hot vector corresponding to the label $k$. Clearly, $\mathbf{f}_i$ is a continuous function in $\mathcal{X}$, because $\mathcal{X}$ is a discrete set. Tietze Extension Theorem implies that $\mathbf{f}_i$ can be extended to a continuous function in $\mathbb{R}^d$.

Since $\mathcal{X}$ is a compact set, then Lemma 12 implies that there exist a sequence $\mathbf{q}^i = (l^i_1, ..., l^i_{g^i})$ ($l^i_1 = d$ and $l^i_{g^i} = l$) and $\mathbf{f}_{\mathbf{w},\mathbf{b}} \in \mathcal{F}^\sigma_{\mathbf{q}^i}$ such that

$$\max_{\mathbf{x} \in \mathcal{X}} \|\mathbf{f}_{\mathbf{w},\mathbf{b}}(\mathbf{x}) - \mathbf{f}_i(\mathbf{x})\|_{\ell_2} < \frac{1}{10 \cdot l},$$

where $\|\cdot\|_{\ell_2}$ is the $\ell_2$ norm in $\mathbb{R}^l$. Therefore, for any $\mathbf{x} \in \mathcal{X}$, it easy to check that

$$\arg\max_{k \in \{1,...,l\}} f^k_{\mathbf{w},\mathbf{b}}(\mathbf{x}) = h_i(\mathbf{x}),$$

where $f^k_{\mathbf{w},\mathbf{b}}(\mathbf{x})$ is the $k$-th coordinate of $\mathbf{f}_{\mathbf{w},\mathbf{b}}(\mathbf{x})$. Therefore, $h_i(\mathbf{x}) \in \mathcal{H}^\sigma_{\mathbf{q}^i}$.

Let $\mathbf{q}$ be $(l_1, ..., l_g)$ ($l_1 = d$ and $l_g = l$) satisfying that $\mathbf{q}^i \lesssim \mathbf{q}$. Using Lemma 10, we obtain that $\mathcal{H}^\sigma_{\mathbf{q}^i} \subset \mathcal{H}^\sigma_{\mathbf{q}}$, for each $i = 1, ..., r$. Therefore, $h_1, ..., h_r \in \mathcal{H}^\sigma_{\mathbf{q}}$. $\qquad\square$

**Lemma 15.** *Let the activation function $\sigma$ be the ReLU function. Suppose that $|\mathcal{X}| < +\infty$. If $\{\mathbf{v} \in \mathbb{R}^l : E(\mathbf{v}) \geq \lambda\}$ and $\{\mathbf{v} \in \mathbb{R}^l : E(\mathbf{v}) < \lambda\}$ both contain nonempty open sets of $\mathbb{R}^l$ (here, open set is a topological terminology). There exists a sequence $\mathbf{q} = (l_1, ..., l_g)$ ($l_1 = d$ and $l_g = l$) such that $\mathcal{H}^{\sigma,\lambda}_{\mathbf{q},E}$ consists of all binary classifiers.*

*Proof of Lemma 15.* Since $\{\mathbf{v} \in \mathbb{R}^l : E(\mathbf{v}) \geq \lambda\}$, $\{\mathbf{v} \in \mathbb{R}^l : E(\mathbf{v}) < \lambda\}$ both contain nonempty open sets, we can find $\mathbf{v}_1 \in \{\mathbf{v} \in \mathbb{R}^l : E(\mathbf{v}) \geq \lambda\}$, $\mathbf{v}_2 \in \{\mathbf{v} \in \mathbb{R}^l : E(\mathbf{v}) < \lambda\}$ and a constant $r > 0$ such that $B_r(\mathbf{v}_1) \subset \{\mathbf{v} \in \mathbb{R}^l : E(\mathbf{v}) \geq \lambda\}$ and $B_r(\mathbf{v}_2) \subset \{\mathbf{v} \in \mathbb{R}^l : E(\mathbf{v}) < \lambda\}$, where $B_r(\mathbf{v}_1) = \{\mathbf{v} : \|\mathbf{v} - \mathbf{v}_1\|_{\ell_2} < r\}$ and $B_r(\mathbf{v}_2) = \{\mathbf{v} : \|\mathbf{v} - \mathbf{v}_2\|_{\ell_2} < r\}$, here $\|\cdot\|_{\ell_2}$ is the $\ell_2$ norm.

For any binary classifier $h$ over $\mathcal{X}$, we can induce a vector-valued function as follows: for any $\mathbf{x} \in \mathcal{X}$,

$$\mathbf{f}(\mathbf{x}) = \begin{cases} \mathbf{v}_1, & \text{if } h(\mathbf{x}) = 1; \\ \mathbf{v}_2, & \text{if } h(\mathbf{x}) = 2. \end{cases}$$

Since $\mathcal{X}$ is a finite set, then Tietze Extension Theorem implies that $\mathbf{f}$ can be extended to a continuous function in $\mathbb{R}^d$. Since $\mathcal{X}$ is a compact set, Lemma 12 implies that there exists a sequence $\mathbf{q}^h = (l_1^h, ..., l_{g^h}^h)$ ($l_1^h = d$ and $l_{g^h}^h = l$) and $\mathbf{f}_{\mathbf{w},\mathbf{b}} \in \mathcal{F}_{\mathbf{q}^h}^\sigma$ such that

$$\max_{\mathbf{x} \in \mathcal{X}} \|\mathbf{f}_{\mathbf{w},\mathbf{b}}(\mathbf{x}) - \mathbf{f}(\mathbf{x})\|_{\ell_2} < \frac{r}{2},$$

where $\| \cdot \|_{\ell_2}$ is the $\ell_2$ norm in $\mathbb{R}^l$. Therefore, for any $\mathbf{x} \in \mathcal{X}$, it is easy to check that $E(\mathbf{f}_{\mathbf{w},\mathbf{b}}(\mathbf{x})) \geq \lambda$ if and only if $h(\mathbf{x}) = 1$, and $E(\mathbf{f}_{\mathbf{w},\mathbf{b}}(\mathbf{x})) < \lambda$ if and only if $h(\mathbf{x}) = 2$.

For each $h$, we have found a sequence $\mathbf{q}^h$ such that $h$ is induced by $\mathbf{f}_{\mathbf{w},\mathbf{b}} \in \mathcal{F}_{\mathbf{q}^h}^\sigma$, $E$ and $\lambda$. Since $|\mathcal{X}| < +\infty$, only finite binary classifiers are defined over $\mathcal{X}$. Using Lemma 14, we can find a sequence $\mathbf{q}$ such that $\mathcal{H}_{\text{all}}^b = \mathcal{H}_{\mathbf{q},E}^{\sigma,\lambda}$, where $\mathcal{H}_{\text{all}}^b$ consists of all binary classifiers. $\square$

**Lemma 16.** *Suppose the hypothesis space is score-based. Let $|\mathcal{X}| < +\infty$. If $\{\mathbf{v} \in \mathbb{R}^l : E(\mathbf{v}) \geq \lambda\}$ and $\{\mathbf{v} \in \mathbb{R}^l : E(\mathbf{v}) < \lambda\}$ both contain nonempty open sets, and Condition 2 holds, then there exists a sequence $\mathbf{q} = (l_1, ..., l_g)$ ($l_1 = d$ and $l_g = l$) such that for any sequence $\mathbf{q}'$ satisfying $\mathbf{q} \lesssim \mathbf{q}'$ and any ID hypothesis space $\mathcal{H}^{\text{in}}$, OOD detection is learnable in the separate space $\mathscr{D}_{XY}^s$ for $\mathcal{H}^{\text{in}} \bullet \mathcal{H}^b$, where $\mathcal{H}^b = \mathcal{H}_{\mathbf{q}',E}^{\sigma,\lambda}$ and $\mathcal{H}^{\text{in}} \bullet \mathcal{H}^b$ is defined below Eq. (4).*

*Proof of Lemma 16.* Note that we use the ReLU function as the activation function in this lemma. Using Lemma 10, Lemma 15 and Theorem 7, we can prove this result. $\square$

**Theorem 10.** *Suppose that Condition 2 holds and the hypothesis space $\mathcal{H}$ is FCNN-based or score-based, i.e., $\mathcal{H} = \mathcal{H}_{\mathbf{q}}^\sigma$ or $\mathcal{H} = \mathcal{H}^{\text{in}} \bullet \mathcal{H}^b$, where $\mathcal{H}^{\text{in}}$ is an ID hypothesis space, $\mathcal{H}^b = \mathcal{H}_{\mathbf{q},E}^{\sigma,\lambda}$ and $\mathcal{H} = \mathcal{H}^{\text{in}} \bullet \mathcal{H}^b$ is introduced below Eq. (4), here $E$ is introduced in Eqs. (5) or (6). Then*

> *There is a sequence $\mathbf{q} = (l_1, ..., l_g)$ such that OOD detection is learnable in the separate space $\mathscr{D}_{XY}^s$ for $\mathcal{H}$ if and only if $|\mathcal{X}| < +\infty$.*

*Furthermore, if $|\mathcal{X}| < +\infty$, then there exists a sequence $\mathbf{q} = (l_1, ..., l_g)$ such that for any sequence $\mathbf{q}'$ satisfying that $\mathbf{q} \lesssim \mathbf{q}'$, OOD detection is learnable in $\mathscr{D}_{XY}^s$ for $\mathcal{H}$.*

*Proof of Theorem 10.* Note that we use the ReLU function as the activation function in this theorem.

**• The Case that $\mathcal{H}$ is FCNN-based.**

**First**, we prove that if $|\mathcal{X}| = +\infty$, then OOD detection is not learnable in $\mathscr{D}_{XY}^s$ for $\mathcal{H}_{\mathbf{q}}^\sigma$, for any sequence $\mathbf{q} = (l_1, ..., l_g)$ ($l_1 = d$ and $l_g = K + 1$).

By Lemma 13, Theorems 5 and 8 in [86], we know that $\text{VCdim}(\phi \circ \mathcal{H}_{\mathbf{q}}^\sigma) < +\infty$, where $\phi$ maps ID data to 1 and maps OOD data to 2. Additionally, Proposition 1 implies that Assumption 1 holds and $\sup_{h \in \mathcal{H}_{\mathbf{q}}^\sigma} |\{\mathbf{x} \in \mathcal{X} : h(\mathbf{x}) \in \mathcal{Y}\}| = +\infty$, when $|\mathcal{X}| = +\infty$. Therefore, Theorem 5 implies that OOD detection is not learnable in $\mathscr{D}_{XY}^s$ for $\mathcal{H}_{\mathbf{q}}^\sigma$, when $|\mathcal{X}| = +\infty$.

**Second**, we prove that if $|\mathcal{X}| < +\infty$, there exists a sequence $\mathbf{q} = (l_1, ..., l_g)$ ($l_1 = d$ and $l_g = K + 1$) such that OOD detection is learnable in $\mathscr{D}_{XY}^s$ for $\mathcal{H}_{\mathbf{q}}^\sigma$.

Since $|\mathcal{X}| < +\infty$, it is clear that $|\mathcal{H}_{\text{all}}| < +\infty$, where $\mathcal{H}_{\text{all}}$ consists of all hypothesis functions from $\mathcal{X}$ to $\mathcal{Y}_{\text{all}}$. According to Lemma 14, there exists a sequence $\mathbf{q}$ such that $\mathcal{H}_{\text{all}} \subset \mathcal{H}_{\mathbf{q}}^\sigma$. Additionally, Lemma 13 implies that there exist $\mathcal{H}^{\text{in}}$ and $\mathcal{H}^b$ such that $\mathcal{H}_{\mathbf{q}}^\sigma \subset \mathcal{H}^{\text{in}} \bullet \mathcal{H}^b$. Since $\mathcal{H}_{\text{all}}$ consists all hypothesis space, $\mathcal{H}_{\text{all}} = \mathcal{H}_{\mathbf{q}}^\sigma = \mathcal{H}^{\text{in}} \bullet \mathcal{H}^b$. Therefore, $\mathcal{H}^b$ contains all binary classifiers from $\mathcal{X}$ to $\{1, 2\}$. Theorem 7 implies that OOD detection is learnable in $\mathscr{D}_{XY}^s$ for $\mathcal{H}_{\mathbf{q}}^\sigma$.

**Third**, we prove that if $|\mathcal{X}| < +\infty$, then there exists a sequence $\mathbf{q} = (l_1, ..., l_g)$ ($l_1 = d$ and $l_g = K + 1$) such that for any sequence $\mathbf{q}' = (l'_1, ..., l'_{g'})$ satisfying that $\mathbf{q} \lesssim \mathbf{q}'$, OOD detection is learnable in $\mathscr{D}^s_{XY}$ for $\mathcal{H}^\sigma_{\mathbf{q}'}$.

We can use the sequence $\mathbf{q}$ constructed in the second step of the proof. Therefore, $\mathcal{H}^\sigma_{\mathbf{q}} = \mathcal{H}_{\mathrm{all}}$. Lemma 10 implies that $\mathcal{H}^\sigma_{\mathbf{q}} \subset \mathcal{H}^\sigma_{\mathbf{q}'}$. Therefore, $\mathcal{H}^\sigma_{\mathbf{q}'} = \mathcal{H}_{\mathrm{all}} = \mathcal{H}^\sigma_{\mathbf{q}}$. The proving process (second step of the proof) has shown that if $|\mathcal{X}| < +\infty$, Condition 2 holds and hypothesis space $\mathcal{H}$ consists of all hypothesis functions, then OOD detection is learnable in $\mathscr{D}^s_{XY}$ for $\mathcal{H}$. Therefore, OOD detection is learnable in $\mathscr{D}^s_{XY}$ for $\mathcal{H}^\sigma_{\mathbf{q}'}$. We complete the proof when the hypothesis space $\mathcal{H}$ is FCNN-based.

**• The Case that $\mathcal{H}$ is score-based**

**Fourth**, we prove that if $|\mathcal{X}| = +\infty$, then OOD detection is not learnable in $\mathscr{D}^s_{XY}$ for $\mathcal{H}^{\mathrm{in}} \bullet \mathcal{H}^{\mathrm{b}}$, where $\mathcal{H}^{\mathrm{b}} = \mathcal{H}^{\sigma,\lambda}_{\mathbf{q},E}$ for any sequence $\mathbf{q} = (l_1, ..., l_g)$ ($l_1 = d, l_g = l$), where $E$ is in Eqs. (5) or (6).

By Theorems 5 and 8 in [86], we know that $\mathrm{VCdim}(\mathcal{H}^{\sigma,\lambda}_{\mathbf{q},E}) < +\infty$. Additionally, Proposition 2 implies that Assumption 1 holds and $\sup_{h \in \mathcal{H}^\sigma_{\mathbf{q}}} |\{\mathbf{x} \in \mathcal{X} : h(\mathbf{x}) \in \mathcal{Y}\}| = +\infty$, when $|\mathcal{X}| = +\infty$. Hence, Theorem 5 implies that OOD detection is not learnable in $\mathscr{D}^s_{XY}$ for $\mathcal{H}^\sigma_{\mathbf{q}}$, when $|\mathcal{X}| = +\infty$.

**Fifth**, we prove that if $|\mathcal{X}| < +\infty$, there exists a sequence $\mathbf{q} = (l_1, ..., l_g)$ ($l_1 = d$ and $l_g = l$) such that OOD detection is learnable in $\mathscr{D}^s_{XY}$ for for $\mathcal{H}^{\mathrm{in}} \bullet \mathcal{H}^{\mathrm{b}}$, where $\mathcal{H}^{\mathrm{b}} = \mathcal{H}^{\sigma,\lambda}_{\mathbf{q},E}$ for any sequence $\mathbf{q} = (l_1, ..., l_g)$ ($l_1 = d, l_g = l$), where $E$ is in Eq. (5) or Eq. (6).

Based on Lemma 16, we only need to show that $\{\mathbf{v} \in \mathbb{R}^l : E(\mathbf{v}) \geq \lambda\}$ and $\{\mathbf{v} \in \mathbb{R}^l : E(\mathbf{v}) < \lambda\}$ both contain nonempty open sets for different scoring functions $E$.

Since $\max_{k \in \{1, ..., l\}} \frac{\exp(v^k)}{\sum_{c=1}^l \exp(v^c)}$, $\max_{k \in \{1, ..., l\}} \frac{\exp(v^k/T)}{\sum_{c=1}^K \exp(v^c/T)}$ and $T \log \sum_{c=1}^l \exp(v^c/T)$ are continuous functions, whose ranges contain $(\frac{1}{l}, 1)$, $(\frac{1}{l}, 1)$, $(0, +\infty)$ and $(0, +\infty)$, respectively.

Based on the property of continuous function ($E^{-1}(A)$ is an open set, if $A$ is an open set), we obtain that $\{\mathbf{v} \in \mathbb{R}^l : E(\mathbf{v}) \geq \lambda\}$ and $\{\mathbf{v} \in \mathbb{R}^l : E(\mathbf{v}) < \lambda\}$ both contain nonempty open sets. Using Lemma 16, we complete the fifth step.

**Sixth**, we prove that if $|\mathcal{X}| < +\infty$, then there exists a sequence $\mathbf{q} = (l_1, ..., l_g)$ ($l_1 = d$ and $l_g = l$) such that for any sequence $\mathbf{q}' = (l'_1, ..., l'_{g'})$ satisfying that $\mathbf{q} \lesssim \mathbf{q}'$, OOD detection is learnable in $\mathscr{D}^s_{XY}$ for for $\mathcal{H}^{\mathrm{in}} \bullet \mathcal{H}^{\mathrm{b}}$, where $\mathcal{H}^{\mathrm{b}} = \mathcal{H}^{\sigma,\lambda}_{\mathbf{q}',E}$, where $E$ is in Eq. (5) or Eq. (6).

In the fifth step, we have proven that Eq. (5) and Eq. (6) meet the condition in Lemma 16. Therefore, Lemma 16 implies this result. We complete the proof when the hypothesis space $\mathcal{H}$ is score-based. $\quad\square$

# M   Proofs of Theorem 11 and Theorem 12

## M.1   Proof of Theorem 11

**Theorem 11.** *Suppose that each domain $D_{XY}$ in $\mathscr{D}^{\mu,b}_{XY}$ is attainable, i.e., $\arg\min_{h \in \mathcal{H}} R_D(h) \neq \emptyset$ (the finite discrete domains satisfy this). Let $K = 1$ and the hypothesis space $\mathcal{H}$ be score-based ($\mathcal{H} = \mathcal{H}^{\sigma,\lambda}_{\mathbf{q},E}$, where $E$ is in Eqs. (5) or (6)) or FCNN-based ($\mathcal{H} = \mathcal{H}^\sigma_{\mathbf{q}}$). If $\mu(\mathcal{X}) < +\infty$, then the following four conditions are **equivalent**:*

> *Learnability in $\mathscr{D}^{\mu,b}_{XY}$ for $\mathcal{H}$ $\iff$ Condition 1 $\iff$ Realizability Assumption $\iff$ Condition 3*

*Proof of Theorem 11.*

1) By Lemma 1, we conclude that Learnability in $\mathscr{D}^{\mu,b}_{XY}$ for $\mathcal{H} \Rightarrow$ Condition 1.

2) By Proposition 1 and Proposition 2, we know that when $K = 1$, there exist $h_1, h_2 \in \mathcal{H}$, where $h_1 = 1$ and $h_2 = 2$, here 1 represents ID, and 2 represent OOD. Therefore, we know that when $K = 1$, $\inf_{h \in \mathcal{H}} R^{\mathrm{in}}_D(h) = 0$ and $\inf_{h \in \mathcal{H}} R^{\mathrm{out}}_D(h) = 0$, for any $D_{XY} \in \mathscr{D}^{\mu,b}_{XY}$.

By Condition 1, we obtain that $\inf_{h \in \mathcal{H}} R_D(h) = 0$, for any $D_{XY} \in \mathscr{D}^{\mu,b}_{XY}$. Because each domain $D_{XY}$ in $\mathscr{D}^{\mu,b}_{XY}$ is attainable, we conclude that Realizability Assumption holds.

We have proven that Condition 1⇒ Realizability Assumption.

3) By Theorems 5 and 8 in [86], we know that $\text{VCdim}(\phi \circ \mathcal{H}_{\mathbf{q}}^{\sigma}) < +\infty$ and $\text{VCdim}(\mathcal{H}_{\mathbf{q},E}^{\sigma,\lambda}) < +\infty$. Then, using Theorem 9, we conclude that Realizability Assumption⇒ Learnability in $\mathscr{D}_{XY}^{\mu,b}$ for $\mathcal{H}$.

4) According to the results in 1), 2) and 3), we have proven that

$$\text{Learnability in } \mathscr{D}_{XY}^{\mu,b} \text{ for } \mathcal{H} \Leftrightarrow \text{Condition 1} \Leftrightarrow \text{Realizability Assumption.}$$

5) By Lemma 2, we conclude that Condition 3⇒Condition 1.

6) **Here we prove that Learnability in $\mathscr{D}_{XY}^{\mu,b}$ for $\mathcal{H}$ ⇒Condition 3.** Since $\mathscr{D}_{XY}^{\mu,b}$ is the prior-unknown space, by Theorem 1, there exist an algorithm $\mathbf{A} : \cup_{n=1}^{+\infty}(\mathcal{X} \times \mathcal{Y})^n \to \mathcal{H}$ and a monotonically decreasing sequence $\epsilon_{\text{cons}}(n)$, such that $\epsilon_{\text{cons}}(n) \to 0$, as $n \to +\infty$, and for any $D_{XY} \in \mathscr{D}_{XY}^{\mu,b}$,

$$\mathbb{E}_{S \sim D_{X_I Y_I}^n}\left[R_D^{\text{in}}(\mathbf{A}(S)) - \inf_{h \in \mathcal{H}} R_D^{\text{in}}(h)\right] \le \epsilon_{\text{cons}}(n),$$

$$\mathbb{E}_{S \sim D_{X_I Y_I}^n}\left[R_D^{\text{out}}(\mathbf{A}(S)) - \inf_{h \in \mathcal{H}} R_D^{\text{out}}(h)\right] \le \epsilon_{\text{cons}}(n).$$

Then, for any $\epsilon > 0$, we can find $n_\epsilon$ such that $\epsilon \ge \epsilon_{\text{cons}}(n_\epsilon)$, therefore, if $n = n_\epsilon$, we have

$$\mathbb{E}_{S \sim D_{X_I Y_I}^{n_\epsilon}}\left[R_D^{\text{in}}(\mathbf{A}(S)) - \inf_{h \in \mathcal{H}} R_D^{\text{in}}(h)\right] \le \epsilon,$$

$$\mathbb{E}_{S \sim D_{X_I Y_I}^{n_\epsilon}}\left[R_D^{\text{out}}(\mathbf{A}(S)) - \inf_{h \in \mathcal{H}} R_D^{\text{out}}(h)\right] \le \epsilon,$$

which implies that there exists $S_\epsilon \sim D_{X_I Y_I}^{n_\epsilon}$ such that

$$R_D^{\text{in}}(\mathbf{A}(S_\epsilon)) - \inf_{h \in \mathcal{H}} R_D^{\text{in}}(h) \le \epsilon,$$

$$R_D^{\text{out}}(\mathbf{A}(S_\epsilon)) - \inf_{h \in \mathcal{H}} R_D^{\text{out}}(h) \le \epsilon.$$

Therefore, for any equivalence class $[D'_{XY}]$ with respect to $\mathscr{D}_{XY}^{\mu,b}$ and any $\epsilon > 0$, there exists a hypothesis function $\mathbf{A}(S_\epsilon) \in \mathcal{H}$ such that for any domain $D_{XY} \in [D'_{XY}]$,

$$\mathbf{A}(S_\epsilon) \in \{h' \in \mathcal{H} : R_D^{\text{out}}(h') \le \inf_{h \in \mathcal{H}} R_D^{\text{out}}(h) + \epsilon\} \cap \{h' \in \mathcal{H} : R_D^{\text{in}}(h') \le \inf_{h \in \mathcal{H}} R_D^{\text{in}}(h) + \epsilon\},$$

which implies that Condition 3 holds. Therefore, Learnability in $\mathscr{D}_{XY}^{\mu,b}$ for $\mathcal{H}$ ⇒Condition 3.

7) Note that in 4), 5) and 6), we have proven that

Learnability in $\mathscr{D}_{XY}^{\mu,b}$ for $\mathcal{H}$ ⇒Condition 3⇒Condition 1, and Learnability in $\mathscr{D}_{XY}^{\mu,b}$ for $\mathcal{H}$ ⇔Condition 1, thus, we conclude that Learnability in $\mathscr{D}_{XY}^{\mu,b}$ for $\mathcal{H}$ ⇔Condition 3⇔Condition 1.

8) Combining 4) and 7), we have completed the proof. □

## M.2    Proof of Theorem 12

**Theorem 12.** *Let $K = 1$ and the hypothesis space $\mathcal{H}$ be score-based ($\mathcal{H} = \mathcal{H}_{\mathbf{q},E}^{\sigma,\lambda}$, where $E$ is in Eqs. (5) or (6)) or FCNN-based ($\mathcal{H} = \mathcal{H}_{\mathbf{q}}^{\sigma}$). Given a prior-unknown space $\mathscr{D}_{XY}$, if there exists a domain $D_{XY} \in \mathscr{D}_{XY}$, which has an overlap between ID and OOD distributions (see Definition 4), then OOD detection is not learnable in the domain space $\mathscr{D}_{XY}$ for $\mathcal{H}$.*

*Proof of Theorem 12.* Using Proposition 1 and Proposition 2, we obtain that $\inf_{h \in \mathcal{H}} R_D^{\text{in}}(h) = 0$ and $\inf_{h \in \mathcal{H}} R_D^{\text{out}}(h) = 0$. Then, Theorem 3 implies this result. □

Note that if we replace the activation function $\sigma$ (ReLU function) in Theorem 12 with any other activation functions, Theorem 12 still hold.