# OpenReview forum: "Is Out-of-Distribution Detection Learnable?"
_NeurIPS.cc/2022/Conference — NeurIPS 2022 Accept_

### Official Review · Reviewer_iXvy · 2022-06-27

**Rating:** 8
**Confidence:** 5
**Soundness:** 4 excellent
**Presentation:** 4 excellent
**Contribution:** 4 excellent

**Summary:**

Recently, reliable AI plays important role in designing an intelligent machine learning system. How to let AI system tell “do not know” is critical for reliable AI systems, which is the focus of this paper. In this paper, the authors consider a practical scenario where out-of-distribution data (the system should not know) is unseen during the training process. In this scenario, the authors want to investigate if the OOD detection is learnable.

The theoretical part is easy to follow. I find that the theoretical contributions are completed and interesting. At first, this paper shows that OOD detection is not learnable in the most general case, which does make sense due to the unavailability of OOD data. Then, this paper points out a necessary condition (sometimes as a necessary and sufficient condition) of the learnability of OOD detection, which directly induces a lot of necessary and sufficient conditions of learnability of OOD detection. In my opinion, this is a significant contribution to the field. Finding necessary and sufficient conditions is always a core and the most important part when studying a problem.

From the practical part, several theorems are considered using networks or finite in-distribution domains, making the whole paper also fit the taste of practitioners. In many practical scenarios, we cannot expect OOD data is the ones we have already seen, which is exactly the problem this paper studies. Besides, the theorem regarding finite ID distributions is also practical. If I understand correctly, in this practical scenario, this paper gives a better result, which is very interesting to me and significant to the field (we often only have finite ID distributions in practice).


**Questions:**

Please answer/revise your paper according to the questions proposed in weaknesses 1,2,3,4,7,8.

**Limitations:**

It is a pure theoretical paper. So I think there is no negative social impacts.



**Strengths And Weaknesses:**

Pros:

1. This paper is the first to characterize the learnability of OOD detection, which makes a significant contribution to the field. There are many OOD detection papers targeting the problem this paper considers. The problem is very difficult yet very important in practice. Previously, no theoretical works are proposed for this problem. In this paper, a completed theory is proposed for this problem, including when OOD detection will fail and when OOD detection will succeed. A lot of necessary and sufficient conditions of learnability of OOD detection are exciting to this field.

2. For practitioners, this paper relieves some big concerns regarding existing OOD detection methods. Before this work, one could intuitively think that OOD detection is not learnable (which is true in the most general case, yet our common datasets are not such general). However, this paper gives a theoretical boundary between learnability and unlearnability of OOD detection by proving some necessary and sufficient conditions. Thus, we can know, on what kind of datasets, OOD detection is learnable.
This contribution is significant and meaningful.

3. Fig. 1 is very helpful in understanding the key necessary condition of OOD detection, which seems that it can motivate a bunch of papers in this research direction.

4. I can see that there are three research topics regarding that “let AI say don’t know”: 1) classification with reject option; 2) PQ learning; and 3) OOD detection. The first two have already had some theories but the last one does not have. This paper fills up this gap, making OOD detection method (which might be more practical than the other two) possible in theory.

5. Although the proofs of this paper are not easy to follow, the logic and organizations of proofs are clear. I have read most proofs and have not found unrecoverable errors for important results. The proofs are soundness.

Cons:

1. I have read some papers regarding PQ learning and feel that PQ learning is totally different from OOD detection. PQ learning focuses on scenarios where OOD data are somehow available, yet OOD detection focuses on the opposite scenarios. However, it is better to demonstrate their difference deeply. Does PQ learning have limitations when meeting different OOD data in the future? I am interested to see some discussions regarding this part.

2. Similar to PQ learning, classification with reject option could be deeply compared to OOD detection instead of just comparing both using plain words. I know they are very different and OOD detection theory is more difficult. But giving more detailed comparation is better for this paper.

3. I have some questions regarding Figure 1, which I hope that the authors can confirm with me. In my opinion, the solid line is the ground-truth line. Do we expect that the estimated lines (dash lines) get closer to the solid line? If so, when overlap exists, why is the solid line not straight? Can you bring me to the specific part regarding this? It seems that the solid line will be straight if there are no overlaps, which makes OOD detection learnable. Is that correct?

4. More explanation, like Figure 1, could be added for understanding the theorems better. Brief proofs might be also useful.

5. In line 26, there are too many separate citations. In my opinion, it is not necessary.

6. Line 148 should not be a new paragraph.

7. The density-based space is very important and interesting. Especially, the theorem 11 is one of the spotlights. Can you give more explanations or applications regarding density-based space (theorems 9 and 11)?

8. The mathematic expression in Definition 1 about PAC learnability is different with the normal expression of PAC learnability. Although line 118 has told us that they are equivalent and I also realize that they are equivalent by paper [21,30] (exercise 4.5 in [21] can prove it?) , the paper will be improved and more clear if a brief proof for the equivalent descriptions is given in the final version.

---

> ### Author Response · Authors · 2022-07-31
> **Response to Reviewer iXvy**
>
> Thanks for your comments! We will answer them as follows:
>
> $\bf{Q1.}$  I have read some papers regarding PQ learning and feel that PQ learning is totally different from OOD detection. PQ learning focuses on scenarios where OOD data are somehow available, yet OOD detection focuses on the opposite scenarios. However, it is better to demonstrate their difference deeply. Does PQ learning have limitations when meeting different OOD data in the future? I am interested to see some discussions regarding this part.
>
> $\bf{A1.}$ Thank you for your comment. This is a good comment. [49, 50] focus on PQ learning theory. In PQ learning, $P$ corresponds to ID distribution $D_{X_{\rm I}}$ in OOD detection. $Q$ corresponds to marginal distribution $D_{X}$. $f$ is the labeling function. Using the same notations in [50], PQ learning aims to achieve the following estimation: for algorithm $\mathbf{A}$,
>
> $~~~~~~~~~~~~~~~~~~~~~~~~~~~~~~~~~~~~~~~~~~~~~~~~~~~~~~~~~~~~$$ \mathbb{E}_S~[{\rm rej}_P(\mathbf{A}(S)) + {\rm err}_Q(\mathbf{A}(S);f)]<\epsilon(n).$
>
> In the separate space and $f$ is only defined over ${\rm supp}~P$, our task aims to achieve the following estimation:  for any $h\in \mathcal{H}$,
>
> $~~~~~~~~~~~~~~~~$$\mathbb{E}_S~[(1-\pi^{\rm out}){\rm rej}_P(\mathbf{A}(S)) $$+ {\rm err}_Q(\mathbf{A}(S);f)]< $$[\{(1-\pi^{\rm out}){\rm rej}_P(h) + {\rm err}_Q(h;f) \} ]+ \epsilon(n).$
>
> PQ learning aims to give PAC estimation or estimation under the realizability assumption. But, we study the agnostic PAC. Under some conditions, PQ learning can be regarded as the PAC theory for OOD detection in the semi-supervised (SS) or transductive learning (TL) cases.  When the OOD distribution is different with Q, PQ learning has limitations when meeting different OOD data.
>
>
>
>
> $\bf{Q2}.$ Similar to PQ learning, classification with reject option could be deeply compared to OOD detection instead of just
> comparing both using plain words. I know they are very different and OOD detection theory is more difficult. But giving more
> detailed comparison is better for this paper.
>
> $\bf{A2.}$ Thank you for your comment. This is a good comment. Many papers [42, 43, 44, 45, 46, 47, 48] have discussed Classification with Reject Option (CwRO). There are two main differences between CwRO and OOD detection.
>
> The first difference is that CwRO only focuses on the ID risk estimation: for any $h\in \mathcal{H}$,
>
> $~~~~~~~~~~~~~~~~~~~~~~~~~~~~~~~~~~~~~~~~~~~~~~~~~~~~~~~~~~~~~~~~~~~~~~~$$\mathbb{E}_S~R_D^{\rm in}(\mathbf{A}(S))<R_D^{\rm in}(h)$
>
> However, OOD detection theory not only focuses on the ID risk estimation, but also focuses on the OOD risk estimation: : for any $h\in \mathcal{H}$,
>
> $~~~~~~~~~~~~~~~~~~~~~~~~~~~~~~~~~~~~~~~~~~~~~~~~~~~~~~~~~~~~~~~~~~~~~~~$$\mathbb{E}_S~R_D^{\rm out}(\mathbf{A}(S))<R_D^{\rm out}(h)$
>
> The second difference is that CwRO focuses on constructing special hypothesis spaces to reject the outlier, however, the hypothesis spaces used in our paper are more general and practical. For example, our hypothesis spaces are FCNN-based, Score-based and kernel-based.
>
> $\bf{Q3.1}.$  In Figure.1, do we expect that the estimated lines (dash lines) get closer to the solid line?
>
> $\bf{A3.1}.$ If we hope that OOD detection is learnable, then we except that the estimated lines (dash lines) get closer to the solid line.
>
> $\bf{Q3.2}.$ If so, when overlap exists, why is the solid line not straight? Can you bring me to the specific part regarding this?
>
> $\bf{A3.2}.$ When the condition $\inf_{h\in \mathcal{H}} R_D^{\rm in}(h)=0$ and $\inf_{h\in \mathcal{H}} R_D^{\rm out}(h)=0$ holds, then we can ensure that when overlap exists, the solid line is not straight. This has been proven in Theorem 3. You can find he detailed proof of Theorem 3 in Appendix.
>
> $\bf{Q3.3}.$ It seems that the solid line will be straight if there are no overlaps, which makes OOD detection learnable. Is that correct?
>
> $\bf{A3.3}.$ For some special OOD domain space, this is correct. Theorems 2, 8 and 11 imply that under some mild conditions, if the solid line is a line, then OOD detection is learnable for the single-distribution space, finite-ID-distribution space and density-based space.
>
> $\bf{Q4}.$ More explanation, like Figure 1, could be added for understanding the theorems better. Brief proofs might be also useful.
>
> $\bf{Q4}.$ Thank you for your comments. This is a very good suggestion. We will add the proof sketch for main theorems (e.g.,
> Theorems 5, 8, 9 and 10) in the final version.

---

> ### Author Response · Authors · 2022-07-31
> **Response to Reviewer iXvy**
>
> ${\bf Q5}.$ The density-based space is very important and interesting. Especially, the theorem 11 is one of the spotlights. Can you
> give more explanations or applications regarding density-based space (theorems 9 and 11)?
>
> ${\bf A5}.$ Thank you for your comments. The density-based space can be widely used. I give two practical examples to explain how to use the density-based space. Example 1, if ID distribution and OOD distribution are mixture truncated normal distributions, then we can check that the generated domains belong to some density-based spaces. Example 2, for any domain space $\mathscr{D}_{XY}$, which contains a density-based space such that the equivalence classes between the domain space and  the density-based space are same, then we can check that Theorems 9 and 11 still hold for this domain space.
>
> ${\bf Q6}.$  The mathematical expression in Definition 1 about PAC learnability is different with the normal expression of PAC learnability. Although line 118 has told us that they are equivalent and I also realize that they are equivalent by paper [21,30] (exercise 4.5 in [21] can prove it?) , the paper will be improved and more clear if a brief proof for the equivalent descriptions is given in the final version.
>
> ${\bf A6}.$ Thanks for your comments and suggestions. You are right. The exercise 4.5 in [21] implies the answer. Reviewer FqYr also proposed a similar question. In the response for Q1 of Reviewer FqYr, we give a proof to show that the standard form of PAC-learnability implies the learnability. In the revision, we also provide a  proof in the Appendix D.3 to show why Definition 1 about PAC learnability is equal to the normal expression of PAC learnability.

---

### Official Review · Reviewer_KK6q · 2022-07-10

**Rating:** 8
**Confidence:** 1
**Soundness:** 4 excellent
**Presentation:** 4 excellent
**Contribution:** 4 excellent

**Summary:**

This paper explores the theoretical foundation of learnability of out-of-distribution detection. Based on the PAC learning theory, the paper proved several impossibility theorems for the learnability of OOD detection under some scenarios, and finds some conditions that OOD detection is PAC-learnable. Also, the paper demonstrate the theory in real practice using FCNN and OOD scores as examples. Recently there are loads of papers proposed empirical methods for OOD detection, but the theory is rarely explored.This paper is the first to investigate the theory of OOD detection so thoroughly, which is meaningful to this field.

**Questions:**

- What is the "+" means in $D_X := (1-\pi^{out}) D_{X_1} + \pi^{out} D_{X_O}$ ? (line 82)

**Limitations:**

Yes.

**Strengths And Weaknesses:**

Strengths:
- The paper is clear and well-written. And the proofs are generally correct.
- This paper is one of the few theoretical works focusing on OOD detection, which plays a significant role in this field.
- The theory is intuitive and have some practical impacts. It can somewhat guide the design of OOD detection algorithms.

Weakness:
- Some notations and expressions can be refined in Section 2. For example, $S$ or $D_{XY}^n $ in eq.2 can be explained (minor).
- Some typos. In section 2 Definition 1. "if there exist an algorithm" -> "if there exists an algorithm".
- Some experiments can be added to show the correctness of the theorems.
- The practical impacts may not be large enough.

---

> ### Author Response · Authors · 2022-07-31
> **Response to Reviewer KK6q**
>
> Thanks for your comments! We will answer them as follows:
>
> ${\bf Q1.}$ Some notations and expressions can be refined in Section 2. For example, $S$ and $D_{X_{\rm I}Y_{\rm I}}^n$ in eq.2 can be explained (minor).
>
> ${\bf A1.}$ Thank you for your helpful comments. In the revision (Appendix D.4), we add explanations to refine some notations and expressions in Section 2.
>
>  In the example,
>
> $S=\{(\mathbf{x}^1,{y}^1),...,(\mathbf{x}^n,{y}^n)\}$ is training data drawn independent and identically distributed  from $D_{X_{\rm I}Y_{\rm I}}$.
>
> $D_{X_{\rm I}Y_{\rm I}}^n$ denotes the probability over $n$-tuples induced
> by applying $D_{X_{\rm I}Y_{\rm I}}$ to pick each element of the tuple independently of the other
> members of the tuple.
>
> Because these samples are i.i.d. drawn $n$ times, researchers often use ''$S\sim D_{X_{\rm I}Y_{\rm I}}^n$" to represent a sample set $S$ (of size $n$) whose each element is drawn i.i.d. from $D_{X_{\rm I}Y_{\rm I}}$.
>
>  The notation $S\sim D_{XY}^n$ is common used in learning theory and can be found in page 38 in machine learning book [21].
>
> ${\bf Q2.}$ Some typos. In section 2 Definition 1. "if there exist an algorithm" $->$ "if there exists an algorithm".
>
> ${\bf A2.}$  Thank you for your helpful suggestions. We will revise these typos in the revision.
>
> ${\bf Q3.}$ Some experiments can be added to show the correctness of the theorems.
>
> ${\bf A3.}$ Thank you for your comments. The mathematical proofs are logical experiments. Compared to empirical experiments, the proofs are more rigorous and comprehensive. In general, when we cannot get mathematical proofs, we often verify our results via empirical experiments. However, when we already have rigorous mathematical proofs, empirical experiments are not necessary.
> Other reviewers also check our proofs and believe that our proofs are solid and correct. Since we have provided rigorous mathematical proofs, it is unnecessary to conduct empirical proofs (i.e., experiments) for our theoretical results.
>
> ${\bf Q4.}$ The practical impacts may not be large enough.
>
> ${\bf A4.}$ Thank you for your comments.
>
> We still argue that our study is not of purely theoretical interest; it has also practical impacts.
>
> In this paper, we are the first to provide the agnostic PAC theory for OOD detection.
>
> First, when we design OOD detection algorithms, we normally only have finite ID datasets,
> corresponding to
> the finite-ID-distribution space. In this case, Theorem 8 provides necessary and sufficient conditions to the success of OOD detection. This theorem is very useful and can give practice guidances.
>
>
> Second, our theory also provides theoretical support (Theorems 10 and 11) for several representative OOD detection works [7,8,23].
>
> Third, our theory shows that OOD detection can be addressed in image-based distributions as long as ID images have clearly different semantic meanings from OOD images.
>
> Fourth, we should not expect a universally working algorithm.  It is necessary to design different algorithms in different scenarios.
>
> Fifth, our theory reveals many necessary and sufficient condition for the learnability of OOD detection, hence opening a door to studying the learnability of OOD detection.
>
> Additionally, the other reviewers also agree with us and think that our theory has large practical impacts.
>
> Reviewer FqYr:  The scenarios that the authors consider are not too technical but highly relevant to practical OOD detection methods. Hence, it gives useful insights for practitioners as well.
>
> Reviewer KYDH: These assumptions are practical and mild, and can be satisfied by many practical cases, for example, FCNNs, CNNs and kernel space. Therefore, the theory can be tightly connected with practical applications.
>
> Reviewer KYDH: I think the contribution are significantly important and this work can give a good guidance for the development of OOD detection. This paper has the potential to achieve a long term impact to OOD learning field.
>
> Reviewer iXvy: From the practical part, several theorems are considered using networks or finite in-distribution domains, making the whole paper also fit the taste of practitioners. In many practical scenarios, we cannot expect OOD data is the ones we have already seen, which is exactly the problem this paper studies. Besides, the theorem regarding finite ID distributions is also practical. If I understand correctly, in this practical scenario, this paper gives a better result, which is very interesting to me and significant to the field (we often only have finite ID distributions in practice).

---

> > ### Comment · Reviewer_KK6q · 2022-08-05
> > **Response**
> >
> > Thanks for the comprehensive response. I have to say sorry for giving an unfair score at first. Since I am not familiar in this area and time is quite limited in reviewing period, I wasn't able to go through all the proofs and give more constructive suggestions. After carefully reading the full paper together with the supplemental materials and other reviewers' comments, I would love to see this paper published and I increased my score.
> > Thanks again for your responses to my trivial questions.

---

> > > ### Author Response · Authors · 2022-08-05
> > > **Response to Reviewer KK6q**
> > >
> > > Dear Reviewer KK6q
> > >
> > > Many thanks for your kind support!
> > >
> > > Do you have more suggestions to improve the quality of our paper? We are glad to discuss our paper with you.
> > >
> > > Best,
> > >
> > > Authors of Paper485

---

> ### Author Response · Authors · 2022-07-31
> **Response to Reviewer KK6q**
>
> Thanks for your comments! We will answer them as follows:
>
> ${\bf Q5.}$ What is the ''+" means in $D_{X}:=(1-\pi^{\rm out}) D_{X_I}+\pi^{\rm out} D_{X_O}$?
>
> ${\bf A5.}$ Thank you for your question.
>
> For convenience, let $P=(1-\pi^{\rm out}) D_{X_{\rm I}}$ and $Q=\pi^{\rm out} D_{X_{\rm O}} $.
>
> It is clear that $P$ and $Q$ are measures. Then $P+Q$ is also a measure, which is defined as follows: for any measurable set $A\subset \mathcal{X}$, we have
> $
>     (P+Q)(A)=P(A)+Q(A).
> $
>
> For example, when $P$ and $Q$ are discrete measures, then $P+Q$ is also discrete measure: for any $\mathbf{x}\in \mathcal{X}$,
> $
>     (P+Q)(\mathbf{x})=P(\mathbf{x})+Q(\mathbf{x}).
> $
>
> When $P$ and $Q$ are continuous measures with density functions $f$ and $g$, then $P+Q$ is also continuous measure with density function $f+g$: for any measurable $A\subset \mathcal{X}$,
> \begin{equation*}
>     P(A) = \int_A f(\mathbf{x}) {\rm d} \mathbf{x},~~~Q(A) = \int_A g(\mathbf{x}) {\rm d} \mathbf{x},
> \end{equation*}
> then
> \begin{equation*}
>     (P+Q)(A) = \int_A f(\mathbf{x})+ g(\mathbf{x}) {\rm d} \mathbf{x}.
> \end{equation*}

---

### Official Review · Reviewer_KYDH · 2022-07-11

**Rating:** 8
**Confidence:** 5
**Soundness:** 4 excellent
**Presentation:** 3 good
**Contribution:** 4 excellent

**Summary:**

The out-of-distribution detection problem is defined as follows: after training on an ID joint distribution $D_{X_{ I}Y_{ I}}$ with random variables from $\mathcal{X}$ and labels in $\mathcal{Y}$, we need to learn a classifier which can detect a test sample as OOD if the sample is drawn from outside of $D_{X_{ I}Y_{ I}}$, while predicting the correct label if the test sample is drawn from ID distribution.

This paper mainly answers the agnostic PAC learnability of out-of-distribution detection in different scenarios, which is known as an open problem in out-of-distribution learning theory.

This paper firstly defines the basic concepts of agnostic PAC learnability of OOD detection, which are natural extensions of agnostic PAC learnability of supervised learning. Then, considering the imbalance issue of OOD detection, the author proposes the prior-unknown spaces and indicates that researchers should focus on agnostic PAC learnability of OOD detection in the prior-unknown spaces.

By discovering a necessary condition (Condition 1), the author shows that the condition cannot hold in the total space and separate space. Based on this observation, the paper proves that in most general setting (total space and separate space), OOD detection is not agnostic PAC learnable.

Next, the author proves the necessary and sufficient conditions to show that the separate space can be learnable if and only if the hypothesis space contains almost all classifiers, while the paper proves that in the finite-ID-distribution space, Condition 3 is the necessary and sufficient condition for the learnability of OOD detection. The paper also proves that in the realizability assumption case, OOD detection is learnable in density-based space.

Lastly, the author considers OOD detection in some practical hypothesis space—FCNN-based and score-based. The paper shows that in the separate space, OOD is learnable in FCNN-based spaces or score-based spaces iff the feature space is finite. In Theorem 11, the paper shows that Condition 1, condition 3 and realizability assumption and learnability are equivalent. In Theorem 12, the author also reveals that overlap will lead to the failure of OOD detection.

This paper is important to understand when and how OOD can work in real applications, as this also gives insight and guidance to OOD detection algorithm designing.



**Questions:**

See the weakness 1,2,3,4.

**Limitations:**

The paper focuses on theory for OOD detection and gives the first theoretical support to understand when OOD detection can work. There is no any potential negative social impact.

**Strengths And Weaknesses:**

Strengths:
1.	The issue is definitely relevant to the NeurIPS as well as ICML, ALT and COLT. When OOD detection can be learnable is an open issue in OOD learning. Due to missing necessary information from the OOD data, the learnability of OOD detection is very difficult. Despite plenty of applied work, there is still few theory to be established for this issue. To address this issue, it requires the author to dig and discovery unknown necessary conditions from scratch. This paper does make an effort to address this problem and make great progress.
2.	This paper is sound. I am interested in this topic, but the paper is long. So I spend several days to check the proofs carefully. All of the results in this paper are supported by proofs.  From what I have checked, all proofs are correct.
3.	The paper answers negatively and positively the question of agnostic PAC learnability of OOD, and introduces sufficient assumptions to recover it (such as assumption 1). These assumptions are practical and mild, and can be satisfied by many practical cases, for example, FCNNs, CNNs and kernel space. Therefore, the theory can be tightly connected with practical applications.
4.	Plenty applied work has been proposed to address this OOD, but theoretical works discussing when OOD detection work is lacking. The paper theoretical shows when OOD can work in practical cases. I think the contribution are significantly important and this work can give a good guidance for the development of OOD detection. This paper has the potential to achieve a long term impact to OOD learning field.
5.	The paper is written well enough to understand.


Weaknesses:

1.	The appendix is long and the proofs are complicated. Although I have check almost all important proofs and believe they are correct, I still spend three days to check them. It is better for the author to provide proof sketch and intuitions for important theorems.
2.	It seems that the description of Theorem 4 in main text is slightly different from the description of Theorem 4 in appendix. I have checked it and found that the description Theorem 4 in appendix is more rigorous. Although you have explained why they are different (because of the space limitation) in appendix G.2, I still suggest that the author should use the description of Theorem 4 in appendix to replace Theorem 4 in main text, because the description in appendix is correct.
3.	Typos/grammar:
1) In line 305, $K$ should be $\lambda$.
2) In line 340, $D_{XY|Y}^{ in}$ should be $D_{X_{I}Y_{I}}$.
3) In line 171, $D_{X_{I}}$ should be $D_{X_{I}Y_{I}}$?
4.  After checking your proof, I think Condition 2 can be removed from Theorems 7 and 10. Although Condition 2 is weak and meaningful, I still think it is better to remove Condition 2. The idea about how to remove Condition 2 can be motivate from the proof of Theorem 9 (the second part).

---

> ### Author Response · Authors · 2022-07-31
> **Response to Reviewer KYDH**
>
> Thanks for your comments! We will answer them as follows:
>
> ${\bf Q1.}$ It is better for the author to provide proof sketch and intuitions for important theorems.
>
> ${\bf A1.}$ This is a very good suggestion. We will add the proof sketch for main theorems (e.g., Theorems 5, 8, 9 and 10) in the final version.
>
> ${\bf Q2.}$ I suggest that the author should use the description of Theorem 4 in appendix G.2 to replace Theorem 4 in main text.
>
>
> ${\bf A2.}$ This is a very good suggestion. Your suggestion is correct. In the revision, we revise Theorem 4 according to your suggestions.
>
> ${\bf A3.}$ Typos/grammar:
>
> 1) In line 305, $K$ should be $\lambda$ ?
>
> 2) In line 340, $D_{XY|Y}^{\rm in}$ should be $D_{X_IY_I}$ ?
>
> 3) In line 171. $D_{X_I}$ should be $D_{X_IY_I}$ ?
>
>
> ${\bf A3.}$ Thank you for your detailed checking. In the revision, we revise all typos according to your suggestions.
>
> ${\bf Q4.}$ After checking your proof, I think Condition 2 can be removed from Theorems 7 and 10. Although Condition 2 is weak and meaningful, I still think it is better to remove Condition 2. The idea about how to remove Condition 2 can be motivate from the proof of Theorem 9 (the second part).
>
> ${\bf A4.}$ Thank you for your constructive comments! Your idea is correct, when $K=1$. However, when $K>1$, Condition 2 can not be removed. Because the techniques used in Theorem 9 require that $\inf_{h\in \mathcal{H}} R_D(h)=0$. When $K=1$,  we can ensure that the approximate error $\inf_{h\in \mathcal{H}} R_D^{\alpha}(h)=0$ in Theorems 7 and can also find FCNN to ensure that  $\inf_{h\in \mathcal{H}} R_D^{\alpha}(h)=0$ in Theorems 10. However, when $K>1$, we cannot guarantee this. Therefore, the techniques developed in Theorem 9 can only be used to remove the Condition 2 when $K=1$.

---

### Official Review · Reviewer_FqYr · 2022-07-11

**Rating:** 8
**Confidence:** 3
**Soundness:** 4 excellent
**Presentation:** 4 excellent
**Contribution:** 3 good

**Summary:**

This paper provides theory on PAC-learnability of out-of-distribution (OOD) detection.
OOD detection is classification task but test data may come from unknown classes.
If test data come from classes known during training, we want to classify them into those classes, but otherwise, we need to detect they belong to unknown classes.
The authors provide a series of theorems about conditions for OOD detection in several interesting setups.
Their results imply that we should not hope for finding an OOD detection algorithm that works in general cases, but we can still design algorithms for special cases.

**Questions:**

- l.118, "they are equivalent by Markov's inequality": I can see that Eq. (2) implies the standard form of PAC-learnability by Markov's inequality, but I cannot see how we can confirm the converse. Could the authors provide a proof or a reference for that?
- ll.260-261, "Since researchers can only collect finite ID datasets as the training data in the process of algorithm design, it is worthy to study the learnability of OOD detection in the finite-ID-distribution space": I am not sure how to relate the fact to finite-ID-distribution space. Does "finite ID datasets" here mean datasets of finite samples or a finite variety of datasets? If it's for the latter sense, do the authors assume $\vert \mathcal{X} \vert < \infty$ here?
- Could the authors define the realizability assumption explicitly?
- The conditions of Theorem 3, $\inf_{h\in\mathcal{H}} R_D^{\mathrm{in}}(h) = 0$ and $\inf_{h\in\mathcal{H}} R_D^{\mathrm{out}}(h) = 0$, look similar to the realizability assumption and the compatibility condition. However, the conditions of Theorem 3 seem to be prohibiting the learnability in Theorem 3 while the realizability and the compatibility conditions are making the learning possible in Theorem 8 and Theorem 9. Should we consider such conditions as good ones or bad ones?
- The paper refer to distributions as "domains." Is this a common way of saying it in the literature? It is a little confusing to me, and I do not see good motivation for the choice of the word.

**Limitations:**

The theory only handles a few special combinations of distributions and hypothesis spaces although I do not consider this as a very strong limitation because they cover many common practical situations.


**Strengths And Weaknesses:**

# Strengths
- The paper provides rigorous theory on an important machine learning task.
- The paper is excellently-written and easy to follow despite its technical content although all the proofs are in the supplemental material.
- The scenarios that the authors consider are not too technical but highly relevant to practical OOD detection methods. Hence, it gives useful insights for practitioners as well.

# Weaknesses
- Most results are negative ones showing impossibility of OOD detection in general cases, and the paper does not provide concrete algorithms.

---

> ### Author Response · Authors · 2022-07-31
> **Response to Reviewer FqYr**
>
> ${\bf Q1.}$ l.118, "they are equivalent by Markov's inequality": I can see that Eq. (2) implies the standard form of PAC-learnability by Markov's inequality, but I cannot see how we can confirm the converse. Could the authors provide a proof or a reference for that?
>
> ${\bf A1.}$ Thanks for your helpful comments. You are right. Eq. (2) implies the standard form of PAC-learnability by Markov's inequality, but the converse is proven by another techniques or inequalities. We give a proof in Appendix D.3. Additionally, Reviewer iXvy also proposed a similar question and believes that exercise 4.5 in [21] implies the answer. We give a brief proof as follows:
>
> We need to prove the standard form of PAC-learnability implies the Eq. (2).
>
> PAC-learnability: for any $\epsilon>0$ and $0<\delta<1$, there exists a function $m(\epsilon,\delta)>0$ such that when the sample size $n>m(\epsilon,\delta)$, we have that with the probability at least $1-\delta>0$,
> $
> R_D(\mathbf{A}(S))-\inf_{h\in \mathcal{H}}  R_D(h) \leq \epsilon.
> $
>
> Note that the loss $\ell$ defined in line 104-105 is bounded (because $\mathcal{Y}_{\rm all}$ is a finite set). We assume the bound of $\ell$ is $M$, i.e., $|\ell|\leq M.$ Hence, according to the definition of PAC-learnability, when the sample size $n>m(\epsilon,\delta)$, we have that
>
> $
>  E_S [ R_D(\mathbf{A}(S))-\inf_{h\in \mathcal{H}}  R_D(h)] \leq \epsilon(1-\delta)+2M\delta < \epsilon+2M\delta.
> $
>
> If we set $\delta = \epsilon$, then when the sample size $n>m(\epsilon,\epsilon)$, we have that
>
> $
>  E_S [ R_D(\mathbf{A}(S))-\inf_{h\in \mathcal{H}}  R_D(h)]  < (2M+1)\epsilon.
> $
>
> this implies that
>
> $
> \lim_{n \rightarrow +\infty} E_S [ R_D(\mathbf{A}(S))-\inf_{h\in \mathcal{H}}  R_D(h)]  =0.
> $
>
> which implies the Eq. (2). We have completed this proof. The key of this proof is that the loss $\ell$ is bounded.
>
> ${\bf Q2.}$ ll.260-261, ''Since researchers can only collect finite ID datasets as the training data in the process of algorithm design, it is worthy to study the learnability of OOD detection in the finite-ID-distribution space": I am not sure how to relate the fact to finite-ID-distribution space. Does "finite ID datasets" here mean datasets of finite samples or a finite variety of datasets? If it's for the latter sense, do the authors assume $|\mathcal{X}|<+\infty$ here?
>
> ${\bf A2.}$ Thank you for your comments. Here the "finite ID datasets" means the number of ID datasets is finite. For examples, in the classical OOD detection paper [23], the authors use the SVHN, CIFAR-10 and CIFAR-100 datasets as the ID-distribution data to conduct experiments. There are three ID datasets, so the domain space, which only contains  SVHN, CIFAR-10 and CIFAR-100 as ID-distribution, can be regarded as the finite-ID-distribution space. Additionally, we needn't to assume that $|\mathcal{X}|<+\infty$. Theorem 8 is the key theorem related to finite-ID-distribution space, but we don't assume $|\mathcal{X}|<+\infty$ in Theorem 8. In Theorem 8, we only assume that $\mathcal{X}$ is a bounded set, which means that there exists a constant $M$ such that for any $\mathbf{x}\in \mathcal{X}$, $||\mathbf{x}||<M$. The assumption that $\mathcal{X}$ is a bounded set is very weak and can be satisfied in most cases.
>
> Note that
> $|\mathcal{X}|<+\infty$ means the number of elements in set $\mathcal{X}$ is finite. However, the assumption that $\mathcal{X}$ is a bounded set means that there exists a constant $M$ such that for any $\mathbf{x}\in \mathcal{X}$, $||\mathbf{x}||<M$. Hence, the two assumptions that $|\mathcal{X}|<+\infty$ and $\mathcal{X}$ is a bounded set are very different.
>
> ${\bf Q3.}$ Could the authors define the realizability assumption explicitly?
>
> ${\bf A3.}$ Thank you for your helpful comments. We will demonstrate realizability assumption in the revision (line 278-279) and give the strict definition in Appendix D.2. The definition of realizability assumption is from definition 2.1 in [21].
>
> Realizability Assumption: if for any domain $D_{XY}\in \mathscr{D}_{XY}$,  there is a  hypothesis function $h^*\in \mathcal{H}$ such that
>
> $~~~~~~~~~~~~~~~~~~~~~~~~~~~~~~~~~~~~~~~~~~~~~~~~~~~~~~~~~~~~~~~~~~~~~~~~~~~~$  $R_{D}(h^*)=0.$

---

> ### Author Response · Authors · 2022-07-31
> **Response to Reviewer FqYr**
>
> Thanks for your comments! We will answer them as follows:
>
> ${\bf Q4.}$ The conditions of Theorem 3, $\inf_{h\in \mathcal{H}} R_D^{\rm in}(h)=0$
> and $\inf_{h\in \mathcal{H}} R_D^{\rm out}(h)=0$, look similar to Realizability assumption and compatibility condition. However, the conditions of Theorem 3 seem to be prohibiting the learnability in Theorem 3 while Realizability and compatibility conditions are making the learning possible in Theorem 8 and Theorem 9. Should we consider such conditions as good ones or bad ones?
>
> ${\bf A4.}$ This is a very good question.
>
> First, the condition $\inf_{h\in \mathcal{H}} R_D^{\rm in}(h)=0$ and $\inf_{h\in \mathcal{H}} R_D^{\rm out}(h)=0$ is very different from  Realizability assumption.
>
> 1) Realizability assumption requires that there is $h^* \in \mathcal{H}$ such that $R_{D}(h^*)=0.$
>  Hence, when the unknown class-prior probability $\pi^{\rm out}>0$, Realizability assumption implies the condition $\inf_{h\in \mathcal{H}} R_D^{\rm in}(h)=0$
> and $\inf_{h\in \mathcal{H}} R_D^{\rm out}(h)=0$.
>
> 2) But the condition $\inf_{h\in \mathcal{H}} R_D^{\rm in}(h)=0$
> and $\inf_{h\in \mathcal{H}} R_D^{\rm out}(h)=0$ doesn't imply Realizability assumption. The proof of Theorem 3 has shown that when ID and OOD distributions have overlap, then when  $\pi^{\rm out}>0$, we have $\inf_{h\in \mathcal{H}}R_D(h)>0$. Hence, we cannot find hypothesis function $h^* \in \mathcal{H}$ such that $
>     R_{D}(h^*)=0.$
>
> 3) The difference between condition $\inf_{h\in \mathcal{H}} R_D^{\rm in}(h)=0$
> and $\inf_{h\in \mathcal{H}} R_D^{\rm out}(h)=0$ and Realizability assumption is that Realizability assumption requires that we can find a ${\bf common}$ hypothesis function $h^*$ such that $R_D^{\rm in}(h^*)=0$
> and $R_D^{\rm out}(h^*)=0$, but the condition $\inf_{h\in \mathcal{H}} R_D^{\rm in}(h)=0$
> and $\inf_{h\in \mathcal{H}} R_D^{\rm out}(h)=0$ does not imply that we can find a ${\bf common}$ function $h^*$ such that $R_D^{\rm in}(h^*)=0$
> and $R_D^{\rm out}(h^*)=0$.
>
> Second, the condition $\inf_{h\in \mathcal{H}} R_D^{\rm in}(h)=0$
> and $\inf_{h\in \mathcal{H}} R_D^{\rm out}(h)=0$ is different from the compatibility condition.
>
> 1) The compatibility condition doesn't imply $\inf_{h\in \mathcal{H}} R_D^{\rm in}(h)=0$
> and $\inf_{h\in \mathcal{H}} R_D^{\rm out}(h)=0$.
>
> 2) The difference between condition $\inf_{h\in \mathcal{H}} R_D^{\rm in}(h)=0$, $\inf_{h\in \mathcal{H}} R_D^{\rm out}(h)=0$ and compatibility condition is that compatibility condition requires that we can find a ${\bf common}$ hypothesis function $h_{\epsilon}$ such that  $R_D^{\rm in}(h_{\epsilon})$
> and $R_D^{\rm out}(h_{\epsilon})$ can approximate $\inf_{h\in \mathcal{H}} R_D^{\rm in}(h)$
> and $\inf_{h\in \mathcal{H}} R_D^{\rm out}(h)$, respectively. But the condition $\inf_{h\in \mathcal{H}} R_D^{\rm in}(h)=0$
> and $\inf_{h\in \mathcal{H}} R_D^{\rm out}(h)=0$ does not imply that we can find a ${\bf common}$ hypothesis function $h_{\epsilon}$ such that  $R_D^{\rm in}(h_{\epsilon})$
> and $R_D^{\rm out}(h_{\epsilon})$ can approximate $\inf_{h\in \mathcal{H}} R_D^{\rm in}(h)$
> and $\inf_{h\in \mathcal{H}} R_D^{\rm out}(h)$, respectively.
>
> Third, the main aim to develop Theorem 3 is different from that of Theorems 8 and 9.
>
> 1) Theorem 3 discusses how the overlap between ID and OOD affects the learnability of OOD detection. According to Theorem 3, we know that if the condition $\inf_{h\in \mathcal{H}} R_D^{\rm in}(h)=0$
> and $\inf_{h\in \mathcal{H}} R_D^{\rm out}(h)=0$ holds, then overlap between ID and OOD results in the failure of OOD detection. In fact, Theorem 3 is deeply related to Theorems 4 and 12 and is the necessary lemma of Theorems 4 and 12.
>
> 2) Theorem 9 discusses when the Realizability assumption holds, OOD detection can be learnable in some cases. When the realizability assumption holds, the overlap will not happen. So there is no contradiction between Theorem 9 and Theorem 3.
>
> 3) Theorem 8 discusses that the compatibility condition is necessary and sufficient condition for the learnability of OOD detection in the finite-ID-distribution space. There is no any relation between Theorem 8 and the condition $\inf_{h\in \mathcal{H}} R_D^{\rm in}(h)=0$
> and $\inf_{h\in \mathcal{H}} R_D^{\rm out}(h)=0$.
>
> Fourth, it is difficult to say whether the condition $\inf_{h\in \mathcal{H}} R_D^{\rm in}(h)=0$
> and $\inf_{h\in \mathcal{H}} R_D^{\rm out}(h)=0$ is good or bad.
>
> 1) This condition is practical. When $K=1$, FCNN-based hypothesis space, score-based hypothesis space and kernel-based hypothesis space satisfy this condition. Thus, in the one-class novelty detection
> and semantic anomaly detection cases, it may be inevitable to meet this condition. So, from the practical perspective, this condition is good.
>
> 2) This condition implies that overlap between ID and OOD distributions can result in the failure of OOD detection. Thus, this condition restricts the scope of application of OOD detection. From this view, this condition is bad.

---

> ### Author Response · Authors · 2022-07-31
> **Response to Reviewer FqYr**
>
> Thanks for your comments! We will answer them as follows:
>
> ${\bf Q5.}$ The paper refer to distributions as "domains." Is this a common way of saying it in the literature?
>
> ${\bf A5.}$ In the classical statistical learning theory papers, researchers directly use "distribution" since there is only one used distribution. When there are more than one distribution used in papers, some researchers tend to use domain to represent the distribution, e.g., transfer learning field, domain adaptation field . Because there are two main distributions used in the OOD detection: ID distribution and OOD distribution, we use "domain" to represent the distributions $D_{XY}$ in our paper.  Additionally, we also note that paper [24] related to open set learning also use the word "domain" to represent the distributions $D_{XY}$.

---

### Meta-Review · Area_Chair_zdzR · 2022-08-24

**Recommendation:** Accept
**Confidence:** Certain

**Metareview:**

This paper studies generalization and learnability questions in the realm of out-of-distribution (OOD) detection. Specifically, it applies PAC learning to the theory of OOD detection. The contributions include new conceptual definitions of agnostic PAC learnability of OOD detection. Then, the authors argue for studying prior-unknown spaces under certain necessary conditions. This leads to a number of novel results, both in theory and in terms of possible practical impact (e.g., when OOD detection will succeed vs. fail). The reviewers found the paper sound, insightful, clearly-written, and novel. This paper benefits the community because it is one of the few theoretical studies of OOD detection.

For the final version, the reviewers have many comments regarding definitions, terminology, and some of the technical details. I encourage the authors to incorporate as much of this feedback as possible to make the paper easier to read for future audiences. For example, please
- add the full proof of how Eq. (2) relates to PAC-learnability,
- add and clarify the realizability assumption in the revision,
- use the description of Theorem 4 in appendix G.2 to replace Theorem 4 in main text.

The authors should also provide proof sketches for the main results (either in the main paper or the appendix). This paper contains many theoretical results, as well as ways to unpack them in the context of more practical scenarios. All of this would benefit from clear exposition. There are also a handful of typos to fix (in the notation/equations and in the exposition). Given the large number of small questions/issues, it is important to address these in the final version of the paper.

The reviewers all vote positively toward acceptance of this paper, and therefore, I also recommend acceptance.

**Award:**

Yes

---

### Decision · Program_Chairs · 2022-09-14

Accept